# OH-Initiated atmospheric degradation of hydroxyalkyl hydroperoxides: mechanism, kinetics, and structure-activity relationship

Long Chen,[1,2] Yu Huang, *[1,2] Yonggang Xue, [1,2] Zhihui Jia,[3] Wenliang Wang[4]

[1] *State Key Lab of Loess and Quaternary Geology (SKLLQG), Institute of Earth Environment, Chinese Academy of Sciences (CAS), Xi'an, 710061, China*

[2] *CAS Center for Excellence in Quaternary Science and Global Change, Xi'an, 710061, China*

[3] *School of Materials Science and Engineering, Shaanxi Normal University, Xi'an, Shaanxi, 710119, China*

[4] *School of Chemistry and Chemical Engineering, Key Laboratory for Macromolecular Science of Shaanxi Province, Shaanxi Normal University, Xi'an, Shaanxi, 710119, China*

Submitted to *Atmospheric Chemistry & Physics*

*Corresponding author:

Prof. Yu Huang, E-mail address: huangyu@ieecas.cn

**Abstract:**

Hydroxyalkyl hydroperoxides (HHPs), formed in the reactions of Criegee intermediates (CIs) with water vapour, play essential roles in the formation of secondary organic aerosol (SOA) under atmospheric conditions. However, the transformation mechanisms for OH-initiated oxidation of HHPs remain incompletely understood. Herein, the quantum chemical and kinetics modeling methods are applied to insight into the detailed mechanisms of OH-initiated oxidation of distinct HHPs ($HOCH_2OOH$, $HOCH(CH_3)OOH$ and $HOC(CH_3)_2OOH$) formed from the reactions of $CH_2OO$, *anti*-$CH_3CHOO$ and $(CH_3)_2COO$) with water vapor. The calculations show that the dominant pathway is the H-abstraction from the -OOH group in the initiation reactions of OH radical with $HOCH_2OOH$ and $HOC(CH_3)_2OOH$. H-abstraction from the -CH group is competitive with that from the -OOH group in the reaction of OH radical with $HOCH(CH_3)OOH$. The barrier of H-abstraction from the -OOH group is slightly increased as the number of methyl group is increased. In pristine environments, the self-reaction of $RO_2$ radical initially produces tetroxide intermediate via an oxygen-to-oxygen coupling, then it decomposes into propagation and termination products through the asymmetric two-step O-O bond scission, in which the rate-limiting step is the first O-O bond cleavage. The barrier height of distinct $RO_2$ radical reactions with $HO_2$ radical is independent on the number of methyl substitution. In urban environments, reaction with $O_2$ forming formic acid and $HO_2$ radical is the dominant removal pathway for $HOCH_2O$ radical formed from the reaction of $HOCH_2OO$ radical with NO. The $\beta$-site C-C bond scission is the dominant pathway in the dissociation of $HOCH(CH_3)O$ and $HOC(CH_3)_2O$ radicals formed from the reactions of NO with $HOCH(CH_3)OO$ and $HOC(CH_3)_2OO$ radicals. These new findings are expected to deepen our current understanding for the photochemical oxidation of hydroperoxides under realistic atmospheric conditions.

## 1. Introduction

Hydroxyalkyl hydroperoxides (HHPs), formed in the reactions of Criegee intermediates (CIs) with water vapour and in the initiation OH-addition with subsequent $HO_2$-termination reactions, play important roles in the formation of secondary organic aerosol (SOA) (Qiu et al., 2019; Kumar et al., 2014). The CIs formed from the ozonolysis of alkenes are characterized by high reactivity and excess energies, which can proceed either prompt unimolecular decay to OH radical or, after collisional stabilization, bimolecular reactions with various trance gases like $SO_2$, $NO_2$ and $H_2O$ to produce sulfate, nitrate and SOA, thereby influencing air quality and human health (Lester and Klippenstein, 2018; Chen et al., 2017, 2019; Liu et al., 2019; Chhantyal-Pun et al., 2017; Anglada and Solé, 2016; Gong and Chen, 2021). Among these reactions, the bimolecular reaction of CIs with water is thought to be the dominant chemical sink because its concentration (1.3-8.3 $\times 10^{17}$ molecules $cm^{-3}$) is several orders of magnitude greater than those of $SO_2$ and $NO_2$ (~ $10^{12}$ molecules $cm^{-3}$) in the atmosphere (Huang et al., 2015; Khan et al., 2018; Taatjes et al., 2013, 2017). The primary products of CIs reactivity toward water are highly oxygenated HHPs that are difficult to detect and identify by using the available analytical techniques due to their thermally instability (Qiu et al., 2019; Anglada and Solé, 2016; Chao et al., 2015; Chen et al., 2016a; Ryzhkov and Ariya, 2003).

HHPs, due to the presence of both hydroxyl and perhydroxy moieties, have relatively low volatility contributing substantially to the formation of SOA (Qiu et al., 2019). The atmospheric degradation of HHPs initiated by OH radical is expected to be one of the dominant loss processes because OH radical is the most powerful oxidizing agent (Gligorovski et al., 2015; Allen et al., 2018). Reaction with OH radical includes three possible H-abstraction channels: (a) the alkyl hydrogen, (b) the -OH hydrogen, and (c) the -OOH hydrogen, which is followed by further reactions to generate organic peroxy radicals ($RO_2$) as reactive intermediates (Allen et al., 2018). Based on our current mechanistic understanding, $RO_2$ radicals have three possible channels in pristine environments: (1) they can proceed self- and cross-reactions resulting in

formation of alkoxy radical RO, alcohol, carbonyl, accretion products (Berndt et al.,
2018; Zhang et al., 2012; Valiev et al., 2019); (2) they can react with $HO_2$ radical
leading to the formation of closed-shell hydroperoxide (ROOH), RO radical, OH
radical, etc.; (Dillon and Crowley, 2008; Iyer et al., 2018) (3) they can undergo
autoxidation via intramolecular H-shift and alternating $O_2$-addition steps producing
highly oxygenated organic molecules (HOMs), which have been identified as the low
volatility compounds that contribute to the formation of SOA (Crounse et al., 2013;
Jokinen et al., 2014; Wang et al., 2018; Ehn et al., 2014; Iyer et al., 2021). In urban
environments, $RO_2$ radicals can react with $NO_x$ generating peroxynitrate ($RO_2NO_2$),
organic nitrate ($RONO_2$), RO radical and other SOA precursors (Wang et al., 2017;
Xu et al., 2014, 2020; Ma et al., 2021). The relative importance of distinct pathways
depends strongly on the nature of $RO_2$ radicals and the concentrations of coreactants.
Hydroxymethyl hydroperoxide (HMHP, $HOCH_2OOH$), the simplest HHPs come
from the ozonolysis of all terminal alkenes in the presence of water, is observed in
significant abundance in the atmosphere (Allen et al., 2018). The measured
concentration of HMHP is varied considerably depending on the location, season and
altitude, and its concentration is measured to be up to 5 ppbv in forested regions
(Allen et al., 2018; Francisco and Eisfeld, 2009). Recently, the concentration of
HMHP was measured during the summer 2013 in the southeastern United States, and
found that the average mixing ratio of HMHP is 0.25 ppbv with a maximum of 4.0
ppbv in the boundary layer (Allen et al., 2018). Allen et al. (2018) conducted the
OH-initiated oxidation of HMHP in an environmental chamber and simulated the
impact of HMHP oxidation on the global formic acid concentration using the
chemical transport model GEOS-Chem. It was found that H-abstraction from the
methyl group of HMHP results in formic acid, and it contributes to the global formic
acid production about 1.7 Tg $yr^{-1}$. Francisco and Eisfeld (2009) by employing *ab*
*initio* CCSD(T)//MP2 methods, studied the atmospheric oxidation mechanism of
HMHP initiated by OH radical, arriving at the same conclusion that the degradation of
HMHP could be a new source of formic acid in the atmosphere. Additionally, the
unimolecular decomposition of HMHP is another important removal process in the
atmosphere. Chen et al. (2016b) found that the formation of $CH_2O$ and $H_2O_2$ is more
preferable than the production of HCOOH and $H_2O$. Kumar et al. (2014) obtained the
same conclusion that the aldehyde- or ketone-forming pathway is kinetically favored
over that the carboxylic acid-forming channel in the unimolecular decomposition of a
variety of HHPs. All the above milestone investigations offer very useful information
for understanding the decomposition of HHPs in the gas phase. However, to the best
of our knowledge, there are few studies on the subsequent transformations of the
resulting H-abstraction products formed from the OH-initiated oxidation of larger
HHPs. The effect of the size and number of substituents on the rates and outcomes of
SOA precursors (e.g. ROOR, HOMs) is uncertain up to now. Therefore, it is necessary
to assess the potential of larger HHPs and their oxidation products to substantial SOA
formation under different $NO_x$ conditions.
In this article, we mainly investigate the detailed mechanisms and kinetic
properties of distinct HHPs oxidation initiated by OH radical by employing quantum
chemical and kinetics modeling methods. For the resulting H-abstraction products
$RO_2$ radicals, the subsequent reactions involving self-reaction, isomerization and
reaction with $HO_2$ radical are taken into account in the absence of NO, while the
subsequent reactions including addition, decomposition and H-abstraction by $O_2$ are
considered in the presence of NO. The investigated HHPs in this work are generated
from the bimolecular reactions of distinct carbonyl oxides ($CH_2OO$, *anti*-$CH_3CHOO$
and $(CH_3)_2COO$) with water vapor.

## 2. Computational details

### 2.1 Electronic structure and energy calculations

The equilibrium geometries of all the open-shell species, including reactant (R),
pre-reactive complex (RC), transition state (TS), post-reactive complex (PC), and
product (P), are fully optimized at the unrestricted M06-2X/6-311+G(2df,2p) level of
theory (UM06-2X) (Zhao and Truhlar, 2006; Zheng and Truhlar, 2009), whereas all
the closed-shell species are optimized at the restricted M06-2X/6-311+G(2df,2p) level
of theory (RM06-2X). This is because the M06-2X functional has been proven to
produce reliable performance for describing thermochemistry, kinetics and
non-covalent interactions (Zhao and Truhlar, 2008). Harmonic vibrational frequencies
are performed at the same level to verify that each stationary point is either a true
minima (with no imaginary frequency) or a transition state (with one imaginary
frequency). Zero-point vibrational energy (ZPVE) and Gibbs free energy corrections
($G_{corr}$) from harmonic vibrational frequencies are scaled by a factor of 0.98 (Zhao and
Truhlar, 2006). The intrinsic reaction coordinate (IRC) calculations are performed to
verify the connection between the transition state and the designated reactant and
product (Fukui, 1981). The single-point energies are calculated at the
(U/R)M06-2X/ma-TZVP level of theory (Zheng, et al., 2011).

149        The tetroxide intermediate formed from the self-reaction of $RO_2$ radical proceeds

through the asymmetric two step O-O bond scission to produce a caged tetroxide
intermediate of overall singlet multiplicity comprising two same-spin alkoxyl radicals
(spin down) and triplet oxygen (spin up). This type of reaction mechanism can be
described by the broken symmetry unrestricted DFT (UDFT) and multi-reference
CASSCF methods (Lee, et al., 2016; Bach, et al., 2005). Previous studies have
demonstrated that the UDFT method is suitable to identify the minimum of metastable
singlet caged radical complex and the transition state of O-O bond homolysis, for
which the energies are comparable to the more accurate and expensive CASSCF
method (Lee, et al., 2016; Bach, et al., 2005). In the present study, the UDFT method
is selected to study the asymmetric O-O bond scission and represents a compromise
between the computational accuracy and efficiency. The broken symmetry
UM06-2X/6-311+G(2df,2p) method is applied to generate the initial guesses of the
tetroxide intermediate and transition state geometries with mixed HOMO and LUMO
($S^2 \approx 1$) by using the guess = mix keyword. The single-point energies are refined at
the UM06-2X/ma-TZVP level of theory.

165        In order to further evaluate the reliability of the employed method in predicting

reaction mechanism, the single-point energies for all the stationary points involved in
the initiation reactions of OH radical with distinct HHPs are recalculated at the
(U/R)CCSD(T)/6-311+G(2df,2p) level of theory based on the (U/R)M06-2X
optimized geometries. Furthermore, the basis set superposition error (BSSE) is also
performed to evaluate the stability of the pre-reactive complexes by employing the
counterpoise method (Boys and Bernardi, 1970). For simplicity, no prefix is adopted
throughout this article. Herein, the Gibbs free energy ($G$) for each species is obtained
by combining the single-point energy with the Gibbs correction ($G = G_{corr} + E$). The
electronic energy ($\Delta E_a^{\#}$) and free energy ($\Delta G_a^{\#}$) barriers are defined as the difference
in energy between transition state and pre-reactive complex ($\Delta E_a^{\#} = E_{TS} - E_{RC}$ and
$\Delta G_a^{\#} = G_{TS} - G_{RC}$). The reaction free energy ($\Delta G$) is referred to the difference in
energy between product and reactant ($\Delta G = G_P - G_R$). The calculated $\Delta E_a^{\#}$ and $\Delta G_a^{\#}$
for the initiation H-abstraction pathways are summarized in Table S1. As shown in
Table S1, the mean absolute deviations (MADs) of $\Delta E_a^{\#}$ and $\Delta G_a^{\#}$ between
CCSD(T)/6-311+G(2df,2p) and M06-2X/ma-TZVP approaches are 0.43 and 0.45
kcal·mol$^{-1}$, respectively; the largest deviations of $\Delta E_a^{\#}$ and $\Delta G_a^{\#}$ are 1.2 and 1.1
kcal·mol$^{-1}$, respectively. These results reveal that the energies obtained from the
M06-2X/ma-TZVP method are in very good accord with those from the gold-standard
coupled-cluster approach CCSD(T) within the uncertainties of systematic errors.
Therefore, the M06-2X/ma-TZVP method is selected to investigate the atmospheric
degradation of HHP initiated by OH radical under different conditions. In the
following sections, unless otherwise stated, the $\Delta G_a^{\#}$ is applied to construct the
reaction profiles.
For the H-shift reactions of RO$_2$ radicals, reactants, transition states and products
have multiple conformers. Previous literature has demonstrated that the reaction
kinetics of multiconformers involvement are more precisely than that of the single
conformer approximation (Møller, et al., 2016, 2020). Herein, the multiconformer
treatment is performed to investigate the H-shift reactions of RO$_2$ radicals. A
conformer search within the Molclus program is employed to generate a pool of
conformers for RO$_2$ radicals (Lu, 2020). The selected conformers are further
optimized at the M06-2X/6-311+G(2df,2p) level of theory, followed by single-point
energy calculations at the M06-2X/ma-TZVP level of theory. On the basis of the
calculated Gibbs free energies, the Boltzmann populations ($w_i$) of each RO$_2$
conformer is expressed as eqn 1.

$$w_i = \frac{e^{-\Delta G_i / k_B T}}{\sum_i e^{-\Delta G_i / k_B T}}$$

(1)

where $\Delta G_i$ is the relative Gibbs free energy of conformer i, $k_B$ is the Boltzmann's
constant, $T$ is temperature in Kelvin. All the quantum chemical calculations are
performed by using the Gaussian 09 program package (Frisch, et al., 2009).

## 2.2 Kinetics calculations

The rate coefficients of unimolecular reactions are calculated by using the
Rice-Ramsperger-Kassel-Marcus theory coupled with energy-grained master equation
(RRKM-ME) method (Holbrook, 1996), and the rate coefficients of bimolecular
reactions are determined by utilizing traditional transition state theory (TST)
(Fernández-Ramos, 2007). The RRKM-ME calculations are performed by
implementing the MESMER 6.0 program suite (Glowacki, et al., 2012). $N_2$ is used as
the buffer gas. The single exponential down model is employed to simulate the
collision energy transfer ($<\Delta E>_{down}$ = 200 cm$^{-1}$). The collisional Lennard-Jones
parameters are estimated by using an empirical formula described by Gilbert and
Smith (1990). For the H-shift reactions of $RO_2$ radicals, the rate coefficients are
determined by employing the multiconformer transition state theory (MC-TST)
approach (Møller, et al., 2016). The MC-TST rate coefficient $k_{MC-TST}$ is calculated by
the sum of the individual intrinsic reaction coordinate TST (IRC-TST) rate coefficient
$k_{IRC-TST}$, each weighted by Boltzmann population of corresponding $RO_2$ conformer
(Møller, et al., 2016).

$$k_{MC-TST} = \sum_i^{\text{all TS conf.}} w_i \times k_{IRC-TST i}$$

(2)

where $k_{IRC-TST i}$ represents the rate coefficient of conformer $i$, and $w_i$ is the relative
Boltzmann population of the corresponding reactant connected to TS$_i$. The
one-dimensional asymmetry Eckart model is employed to calculate the tunneling
correction (Eckart, 1930). Considering the uncertainty in barrier heights (~ 1.0
kcal mol$^{-1}$ by the M06-2X method) and in tunneling corrections, the uncertainty of the
calculated rate coefficient is about one order of magnitude in the present study.

## 3. Results and discussion

## 3.1 Initiation reaction of HHPs with OH radical

Previous literatures have proposed that the lifetime of CI with respect to the
reaction with water vapour exhibits strong dependence on the nature of CIs (Anglada
and Solé, 2016; Taatjes, et al., 2013; Anglada, et al., 2011), and the primary product is
HHPs in both gas phase and air/water interface (Chao, et al., 2015; Chen, et al., 2016a;
Smith et al., 2015; Zhu et al., 2016; Zhong et al., 2018). In the present study, we
mainly consider three kinds of HHPs originated from the addition of water to $CH_2OO$
and methyl-substituted CI (*anti*-$CH_3CHOO$ and $(CH_3)_2COO$). The lowest-energy
conformers HHP ($HOCH_2OOH$, $HOCH(CH_3)OOH$ and $HOC(CH_3)_2OOH$) are
obtained from the previous study as shown in Figure 1 (Chen et al., 2019), and they
are selected as model system to investigate the atmospheric degradation mechanism
of HHP initiated by OH radical. Letters and numbers are applied to mark carbon,
oxygen and hydrogen atoms in different reaction sites.

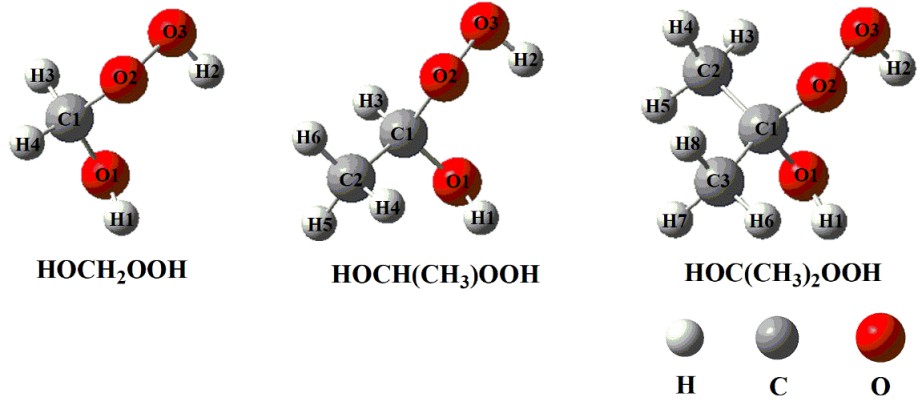

242                      **Figure 1.** The structures of distinct HHPs

The free-energy and electronic-energy potential energy surfaces (PESs) for the
initiation reactions of OH radical with $HOCH_2OOH$, $HOCH(CH_3)OOH$ and
$HOC(CH_3)_2OOH$ are presented in Figures 2-4 and S1-S3, respectively. The optimized
geometries of all the stationary points are displayed in Figures S6-S8, respectively. As
can be seen in Figure 2, the reaction for $HOCH_2OOH$ with OH radical proceeds via
four distinct pathways: H-abstraction from the -$O_1H_1$ (R1), -$C_1H_3$ (R2), -$C_1H_4$ (R3)
and -$O_2O_3H_2$ groups (R4). For each pathway, a pre-reactive complex with a six- or
seven-membered ring structure is formed in the entrance channel, which is stabilized
by hydrogen bond interactions between the oxygen atom of OH radical and the
abstraction hydrogen atom of $HOCH_2OOH$, and the remnant hydrogen atom of OH
radical and one of oxygen atoms of $HOCH_2OOH$ (Figure S6). Then, it surmounts
modest barrier that is higher in energy than the reactants to reaction. The reaction
barriers $\Delta G_a^{\#}$ are reduced in the order of 6.4 (R1) > 5.8 (R2) ≈ 5.4 (R3) > 1.5 (R4)
kcal mol$^{-1}$, indicating that H-abstraction from the -$O_2O_3H_2$ group (R4) is more
preferable than those from the -$O_1H_1$, -$C_1H_3$ and -$C_1H_4$ groups (R1-R3). The same
conclusion is also derived from the energy barriers $\Delta E_a^{\#}$ that R4 is the most favorable
H-abstraction pathway (Figure S1). The difference of barrier heights can be attributed
to the bond dissociation energy (BDE) of different types of bonds in $HOCH_2OOH$
molecule. The BDE decrease in the order of 103.7 ($O_1$-$H_1$) > 98.2 ($C_1$-$H_3$) ≈ 97.4
($C_1$-$H_4$) > 87.2 ($O_3$-$H_2$) kcal mol$^{-1}$, which are in good agreement with the order of
barrier heights of H-abstraction reactions. As indicated by their reaction free energy
values, it can be found that the exothermicity of R4 is the largest among these four
H-abstraction reactions. Based on the above discussions, it is concluded that
H-abstraction from the -$O_2O_3H_2$ group resulting in formation of $HOCH_2OO$ radical
(R4) is feasible on both thermodynamically and kinetically.
Considering the different reaction sites of hydrogen atoms, the atmospheric
transformation of $HOCH(CH_3)OOH$ from the *anti*-$CH_3CHOO$ + $H_2O$ reaction should
have six possible H-abstraction pathways as presented in Figure 3. As shown in
Figure 3, each H-abstraction reaction begins with the formation of a weakly bound
hydrogen bonded pre-reactive complex with a six- or seven-membered ring structure
in the entrance channel (Figure S7). Then it immediately transforms into the
respective product via the corresponding transition state. The $\Delta G_a^{\#}$ of H-abstraction
from the -$C_1H_3$ (R6) and -$O_2O_3H_2$ (R8) groups are 2.2 and 1.7 kcal mol$^{-1}$, respectively,
which are ~ 4-5 kcal mol$^{-1}$ lower than those from the -$O_1H_1$ (R5) and -$CH_3$ groups
(R7). This result shows that R6 and R8 have nearly identical importance in the
atmosphere. Compared with the barriers of H-abstraction at the $C_\alpha$ (R6) and $C_\beta$ (R7)

positions, it can be found that the former case is more favourable than the latter case. This conclusion is further supported by Jara-Toro's study for the reactions of OH radical with linear saturated alcohols (methanol, ethanol and n-propanol) that H-abstraction at the $C_\alpha$ position is predominant (Jara-Toro, et al., 2017, 2018).

For the OH-initiated oxidation of $HOCH(CH_3)OOH$ from the *syn*-$CH_3CHOO$ + $H_2O$ reaction, the corresponding free-energy and electronic-energy PESs are displayed in Figures S4 and S5, respectively. From Figure S4, it can be seen the H-abstraction by OH radical from $HOCH(CH_3)OOH$ has six possible pathways. For each pathway, a per-reactive complex is formed prior to the corresponding transition state, and then it overcomes modest barrier to reaction. The $\Delta G_a^{\#}$ of R6' and R8' are 2.3 and 1.8 kcal $mol^{-1}$, respectively, which are about 5 kcal $mol^{-1}$ lower than those of R5' and R7'. This result shows that H-abstraction from the -CH (R6') and -OOH (R8') groups are preferable kinetically. The same conclusion is also derived from the energy barriers $\Delta E_a^{\#}$ that the R6' and R8' are the most favourable H-abstraction pathways (Figure S5). It should be noted that although the barriers of R6' and R8' are comparable, the exoergicity of the former case is significantly lower than that of the latter case. The above-mentioned conclusions are consistent with the results derived from the OH-initiated oxidation of $HOCH(CH_3)OOH$ from the *anti*-$CH_3CHOO$ + $H_2O$ reaction. Zhou et al. has demonstrated that the bimolecular reaction of *syn*-$CH_3CHOO$ with water leading to the formation of $HOCH(CH_3)OOH$ is of less importance in the atmosphere, while the unimolecular decay to OH radical is the major loss process of *syn*-$CH_3CHOO$ (Zhou et al., 2019). Therefore, in the present study, we mainly focus on the subsequent mechanism of intermediate generated from OH-initiated oxidation of $HOCH(CH_3)OOH$ from the *anti*-$CH_3CHOO$ + $H_2O$ reaction.

From Figure 4, it can be seen that H-abstraction from $HOC(CH_3)_2OOH$ includes eight possible H-abstraction pathways. All the H-abstraction reactions are strongly exothermic and spontaneous, signifying that they are thermodynamically feasible under atmospheric conditions. It deserves mentioning that the release of energy of R12 is significantly greater than those of R9-R11. For each H-abstraction

pathway, a RC with a six- or seven-membered ring structure is formed prior to the corresponding TS, which is more stable than the separate reactants due to the hydrogen bond interactions between $HOC(CH_3)_2OOH$ and OH radical. Then, the RC overcomes modest barrier to reaction. The $\Delta G_a^{\#}$ of H-abstraction from the $-O_2O_3H_2$ group (R12) is 2.7 kcal $mol^{-1}$, which is the lowest among these eight H-abstraction reactions. This result again shows that the H-abstraction from the $-O_2O_3H_2$ group is the dominant pathway.

The rate coefficients of every H-abstraction pathway involved in the initiation reactions of distinct HHPs with OH radical are estimated over the temperature range from 273 to 400 K as summarized in Table S2-S4 and Figures S9-S11. As shown in Table S2, the total rate coefficients $k_{tot}$ of $HOCH_2OOH$ reaction with OH radical decrease slightly with increasing temperature. At ambient temperature, $k_{tot}$ is estimated to be $3.3 \times 10^{-11}$ $cm^3$ molecule$^{-1}$ s$^{-1}$, which is a factor of ~5 greater than the Allen's result $((7.1 \pm 1.5) \times 10^{-12}$ $cm^3$ molecule$^{-1}$ s$^{-1}$, at 295 K) derived from the reaction of HMHP with OH radical by using the $CF_3O^-$ chemical ionization mass spectrometry (CIMS) and laser-induced fluorescence (LIF) (Allen, et al., 2018). Such a discrepancy could be attributed to the uncertainty in barrier height and tunneling correction. $k_{R4(O3-H2)}$ is 1-2 orders of magnitude greater than $k_{R1(O1-H1)}$, $k_{R2(C1-H3)}$ and $k_{R3(C1-H4)}$ in the whole temperature range, implying that R4 is the most favorable H-abstraction pathway. For example, $k_{R4(O3-H2)}$ is calculated to be $2.9 \times 10^{-11}$ $cm^3$ molecule$^{-1}$ s$^{-1}$ at 298 K, which is higher than $k_{R1(O1-H1)}$ $(1.8 \times 10^{-13})$, $k_{R2(C1-H3)}$ $(9.9 \times 10^{-13})$ and $k_{R3(C1-H4)}$ $(2.0 \times 10^{-12})$ by 161, 29 and 15 times, respectively.

From Table S3, it can be seen that the total rate coefficients $k'_{tot}$ of $HOCH(CH_3)OOH$ reaction with OH radical decrease in the range of $4.5 \times 10^{-11}$ (273 K) to $8.1 \times 10^{-12}$ (400 K) $cm^3$ molecule$^{-1}$ s$^{-1}$ with increasing temperature, and they exhibit a slightly negative temperature dependence. $k_{R8(O3-H2)}$ are approximately identical to $k'_{tot}$ in the entire temperature range, which are 1-2 orders of magnitude greater than $k_{R5(O1-H1)}$, $k_{R6(C1-H3)}$, $k_{R7-1(C2-H4)}$, $k_{R7-2(C2-H5)}$ and $k_{R7-3(C2-H6)}$. The result again shows that H-abstraction from the -OOH group (R8) is preferable kinetically. It should be noted that although the barriers of R8 and R6 are comparable, $k_{R8(O3-H2)}$ is

about one order of magnitude higher than $k_{R6(C1-H3)}$ over the temperature range studied.
The most likely reason is due to the stability of pre-reactive complexes that IM8-a is
more stable than IM6-a in energy. A similar conclusion is derived from the results of
rate coefficients of $HOC(CH_3)_2OOH + OH$ reaction that H-abstraction from the -OOH
group (R12) is favorable kinetically (Table S4). The atmospheric lifetime of
$HOCH_2OOH$, $HOCH(CH_3)OOH$ and $HOC(CH_3)_2OOH$ reactivity toward OH radical
are estimated to be 0.58-1.74 h, 0.60-1.79 h and 1.23-3.69 h at room temperature
under typical OH radical concentrations of $5\text{-}15 \times 10^6$ molecules $cm^{-3}$ during daylight
(Long et al., 2017).

348        In summary, the dominant pathway is the H-abstraction from the -OOH group in

the initiation reactions of OH radical with $HOCH_2OOH$. H-abstraction from the -CH
group is competitive with that from the -OOH group in the reaction of OH radical
with $HOCH(CH_3)OOH$. Compared with the barriers of H-abstraction from the -OOH
and $-CH_2$ groups in the reaction of OH radical with $HOCH_2OOH$, it can be found
that the barrier of H-abstraction from the -CH group is reduced by 3.6 kcal $mol^{-1}$,
whereas the barrier of H-abstraction from the -OOH group is increased by 0.2
kcal $mol^{-1}$ when a methyl group substitution occurs at the C1-position of
$HOCH_2OOH$. The dominant pathway is the H-abstraction from the -OOH group in
the reaction of OH radical with $HOC(CH_3)_2OOH$, and the barrier height is increased
by 1.2 kcal $mol^{-1}$ compared to the OH + $HOCH_2OOH$ system. The barrier of
H-abstraction from the -OOH group is slightly increased as the number of methyl
group is increased. It is interesting to compare the rate coefficient of dominant
pathway in the OH + $HOCH_2OOH$ system with that for the analogous reactions in
the OH + $HOCH(CH_3)OOH$ and OH + $HOC(CH_3)_2OOH$ reactions. It can be found
that the rate coefficient is almost identical when a methyl group substitution occurs
at the $C_1$-position, whereas the rate coefficient reduces by a factor of 2-5 when two
methyl groups are introduced into the $C_1$-position.

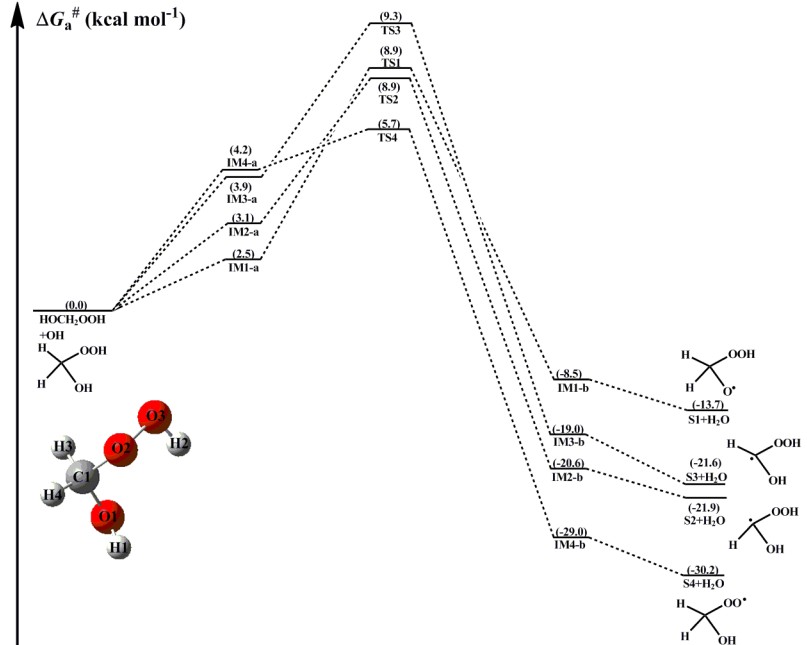

**Figure 2.** PES ($\Delta G_a^{\#}$) for the OH-initiated reactions of $HOCH_2OOH$ from the $CH_2OO + H_2O$
reaction predicted at the M06-2X/ma-TZVP//M06-2X/6-311+G(2df,2p) level of theory (a and b
represent the pre-reactive and post-reactive complexes)

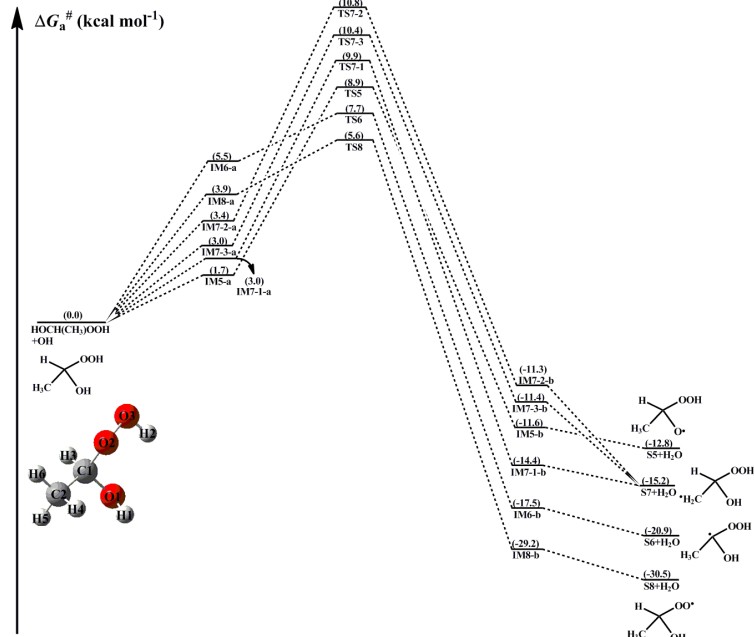

**Figure 3.** PES ($\Delta G_a^{\#}$) for the OH-initiated reactions of $HOCH(CH_3)OOH$ from the
*anti*-$CH_3CHOO + H_2O$ reaction predicted at the M06-2X/ma-TZVP//M06-2X/6-311+G(2df,2p)
level of theory (a and b represent the pre-reactive and post-reactive complexes)

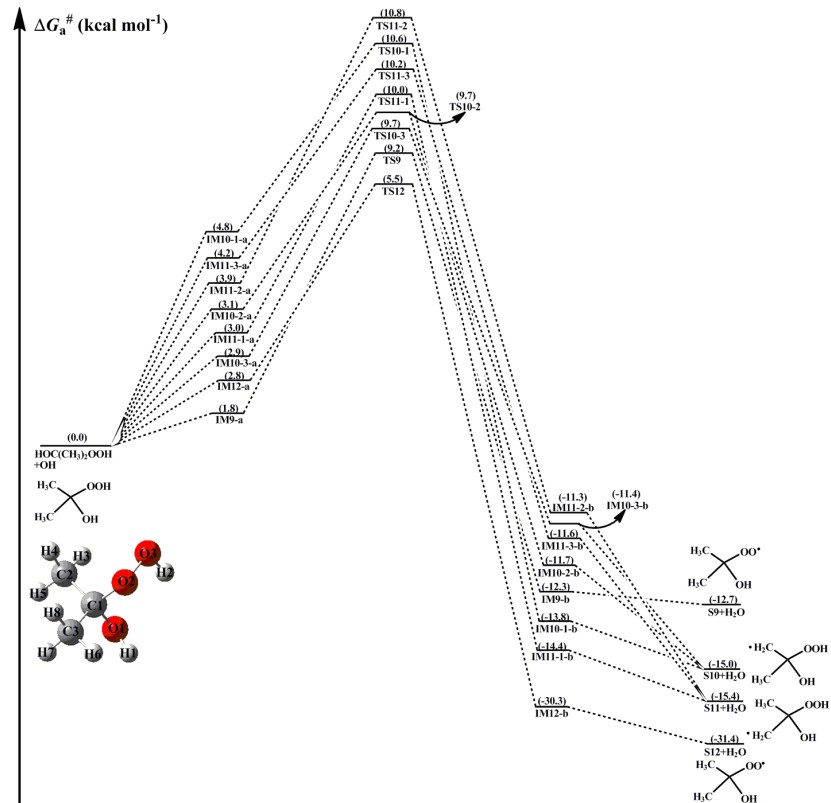


**Figure 4.** PES ($\Delta G_a^{\#}$) for the OH-initiated reactions of HOC(CH$_3$)$_2$OOH from the (CH$_3$)$_2$COO + H$_2$O reaction predicted at the M06-2X/ma-TZVP//M06-2X/6-311+G(2df,2p) level of theory (a and b represent the pre-reactive and post-reactive complexes)

## 3.2 Subsequent reactions of H-abstraction products RO$_2$ radicals in pristine environments

In principle, the H-abstraction products RO$_2$ radicals have three possible fates in pristine environments: (1) the self-reactions of RO$_2$ radicals can either produce RO + R'O + O$_2$ (propagation channel), or generate ROH + R'(-H, =O) + O$_2$ or produce ROOR + O$_2$ (termination channel) that has been recognized as an important SOA precursor (Berndt et al., 2018; Zhang et al., 2012); (2) RO$_2$ radicals react with HO$_2$ radical leading to the formation of hydroperoxide ROOH, alcohol, OH and other products (Winiberg et al., 2016; Chen et al., 2021); (3) RO$_2$ radicals autoxidation through intramolecular H-shift and alternating O$_2$ addition steps generate HOMs (Ehn et al., 2014; Bianchi et al., 2019; Nozière and Vereecken, 2019; Rissanen et al., 2014). The relevant details for these three kinds of reactions will be discussed in the following paragraph.

## 3.2.1 Reactions mechanism for the self-reaction of $RO_2$ radicals

The self-reaction is one of dominant removal pathways for $RO_2$ radicals when the concentration of NO is low and the concentration of $RO_2$ radicals is high. The self-reaction of $RO_2$ radicals usually follows the Russell mechanism (Russell, 1957), and mainly includes four possible pathways: (1) $2RO_2 \rightarrow 2RO + O_2$; (2) $2RO_2 \rightarrow ROH + R'CO + O_2$; (3) $2RO_2 \rightarrow ROOR + O_2$; (4) $2RO_2 \rightarrow ROOH + R'CHOO$ (Atkinson and Arey, 2003). The relative importance of different pathways is varied considerably depending on the nature of $RO_2$ radicals (Valiev et al., 2019; Lee et al., 2016). A schematic PES for the self-reaction of $HOCH_2OO$ radical is drawn in Figure 5. As can be seen in Figure 5a, the self-reaction of $HOCH_2OO$ radical starts with the formations of tetroxide complexes IM13-a and IM14-a in the entrance channel, with 2.9 and 3.4 kcal·$mol^{-1}$ stability. Then they fragment into dimer S13 + $^1O_2$ (R13) and $HOCH_2OOH + HOCHOO$ (R14) via transition states TS13 and TS14 with the barriers of 43.3 and 51.5 kcal·$mol^{-1}$. But the barriers of R13 and R14 are extremely high, making them irrelevant in the atmosphere.

From Figure 5b, it is seen that the self-reaction of $HOCH_2OO$ radical proceeds via oxygen-to-oxygen coupling leading to the formation of tetroxide intermediate S14 with the electronic energy and free energy barriers of 7.3 and 19.6 kcal·$mol^{-1}$. Kumar and Francisco reported that the electronic energy barrier of the gas phase decomposition of $HOCH_2OO$ radical is 14.0 kcal·$mol^{-1}$ and it could be a new source of $HO_2$ radical in the troposphere (Kumar and Francisco, 2015, 2016). Compared with the electronic energy barriers of unimolecular dissociation of $HOCH_2OO$ radical and its self-reaction, it can be found that the self-reaction of $HOCH_2OO$ radical resulting in formation of S14 is significantly feasible. The formed S14 can fragment into $HOCH_2O + HCOOH + HO_2$ via a concerted process of $O_2$-$O_3$ and $O_5$-$O_6$ bonds rupture and $O_3$-$H_6$ bond forming with the barrier of 29.8 kcal·$mol^{-1}$. Alternatively, S14 can convert into the caged tetroxide intermediate S16 through the asymmetric two step $O_2$-$O_3$ and $O_5$-$O_6$ bonds scission with the barriers of 19.1 and 3.1 kcal·$mol^{-1}$, respectively. The result shows that the latter pathway is more preferable than the

former channel owing to its lower barrier. The overall spin multiplicity of S16 is singlet, in which the $O_2$ moiety maintains the triplet ground state (spin up) and is very loosely bound. In order to preserve the overall singlet multiplicity, the two $HOCH_2O$ radical pairs ($^3(HOCH_2O \cdot\cdot HOCH_2O)$) must have the triplet multiplicity (spin down). S16 could be regarded as the ground state $^3O_2$ moving away from the two $HOCH_2O$ radical pairs that keep interacting. Due to the difficulty in performing the constrained optimization for the dissociation of S16, the $^3O_2$ moiety is considered as a leaving moiety away from two $HOCH_2O$ radical pairs, and merely the dissociation of $^3(HOCH_2O \cdot\cdot HOCH_2O)$ is taken into consideration in the present study. It has three types of pathways: (1) it yields $HOCH_2OH$ and excited-state $^3HCOOH$ through the alpha hydrogen transfer with the barrier of 14.0 kcal $mol^{-1}$ and 10.2 kcal $mol^{-1}$ exothermicity, followed by the excited $^3HCOOH$ to go back to the ground-state $^1HCOOH$; (2) it generates two $HOCH_2O$ radicals via the barrierless process with the exoergicity of 16.9 kcal $mol^{-1}$; (3) it produces dimer S17 via an intersystem crossing (ISC) step with the exoergicity of 32.1 kcal $mol^{-1}$. Based on the calculated reaction barriers, it can be found that the rate-limiting step is the cleavage of $O_2$-$O_3$ bond (R17) in the unimolecular decay processes of S14. This conclusion coincides with the previous result obtained from the dissociation of di-t-butyl tetroxide that the rate-controlling step is the rupture of single O-O bond (Lee et al., 2016). Valiev et al. (2019) proposed that the ISC rate of ROOR dimer formed from the different (RO $\cdot\cdot$ R'O) systems is extremely rapid ($> 10^8$ $s^{-1}$) and exhibits a strong stereoselectivity.

Figure 6 depicts a schematic PES for the self-reaction of $HOCH(CH_3)OO$ radical. As shown in Figure 6a, the self-reaction of $HOCH(CH_3)OO$ radical can either produce dimer S18 along with $^1O_2$ via transition state TS20 with the barrier of 44.4 kcal $mol^{-1}$, or generate $HOCH(CH_3)OOH$ and $HOC(CH_3)OO$ though transition state TS21 with the barrier of 54.3 kcal $mol^{-1}$. But the barriers of R20 and R21 are significantly high, making them of less importance in the atmosphere. Alternatively, the self-reaction of $HOCH(CH_3)OO$ radical proceeds via an oxygen-to-oxygen coupling resulting in formation of tetroxide intermediate S19 with the barrier of 19.9

kcal mol$^{-1}$ (Figure 6b). The formed S19 proceeds through the asymmetric two step
$O_2$-$O_3$ and $O_5$-$O_6$ bonds scission to produce a caged tetroxide intermediate S21 of
overall singlet multiplicity comprising two same-spin alkoxyl radicals (spin down)
and triplet oxygen (spin up). These two processes overcome the barriers of 21.4 and
1.3 kcal mol$^{-1}$, respectively. Then, S21 decomposes into the propagation
($2HOCH(CH_3)O$ + $^3O_2$) and termination products ($HOCH(CH_3)OH$ + $^3CH_3OOH$ +
$^3O_2$ and dimer S22 + $^3O_2$) with the exoergicity of 12.5, 11.7 and 33.0 kcal mol$^{-1}$. The
rate-determining step is the rupture of $O_2$-$O_3$ bond (R24) in the dissociation processes
of S19.

459        As shown in Figure 7, the dominant pathway for the self-reaction of

$HO(CH_3)_2COO$ radical begins with the formation of tetroxide intermediate S24 via an
oxygen-to-oxygen coupling transition state TS28 with the barrier of 20.4 kcal mol$^{-1}$;
then it transforms into the caged tetroxide intermediate S26 of overall singlet spin
multiplicity through the asymmetric two-step O-O bond cleavage with the barriers of
22.0 and 3.4 kcal mol$^{-1}$; finally, S26 can either produce two $HO(CH_3)_2CO$ radicals
with the exoergicity of 10.3 kcal mol$^{-1}$, or generate dimer S27 with the exothermicity
of 31.5 kcal mol$^{-1}$. Compared with the self-reactions of $HOCH_2OO$ and
$HOCH(CH_3)OO$ radicals, it can be found that the termination product of the
self-reaction of $HOC(CH_3)_2OO$ radical is exclusively dimer S27. The reason is due to
the absence of alpha hydrogen atom in $HOC(CH_3)_2OO$ radical. Compared with the
barrier of rate-determining route R17 in the self-reaction of $HOCH_2OO$ radical, it can
be found that the barrier of rate-limiting step R29 is increased by about 3.0 kcal mol$^{-1}$
when two methyl substitutions are introduced into the C1-position of $HOCH_2OO$
radical. The reason might be attributed to the cage escape of alkoxyl radicals. It is
therefore that the tertiary $RO_2$ radicals have great opportunity to react with $HO_2$
radical or undergo autoxidation in pristine environments.

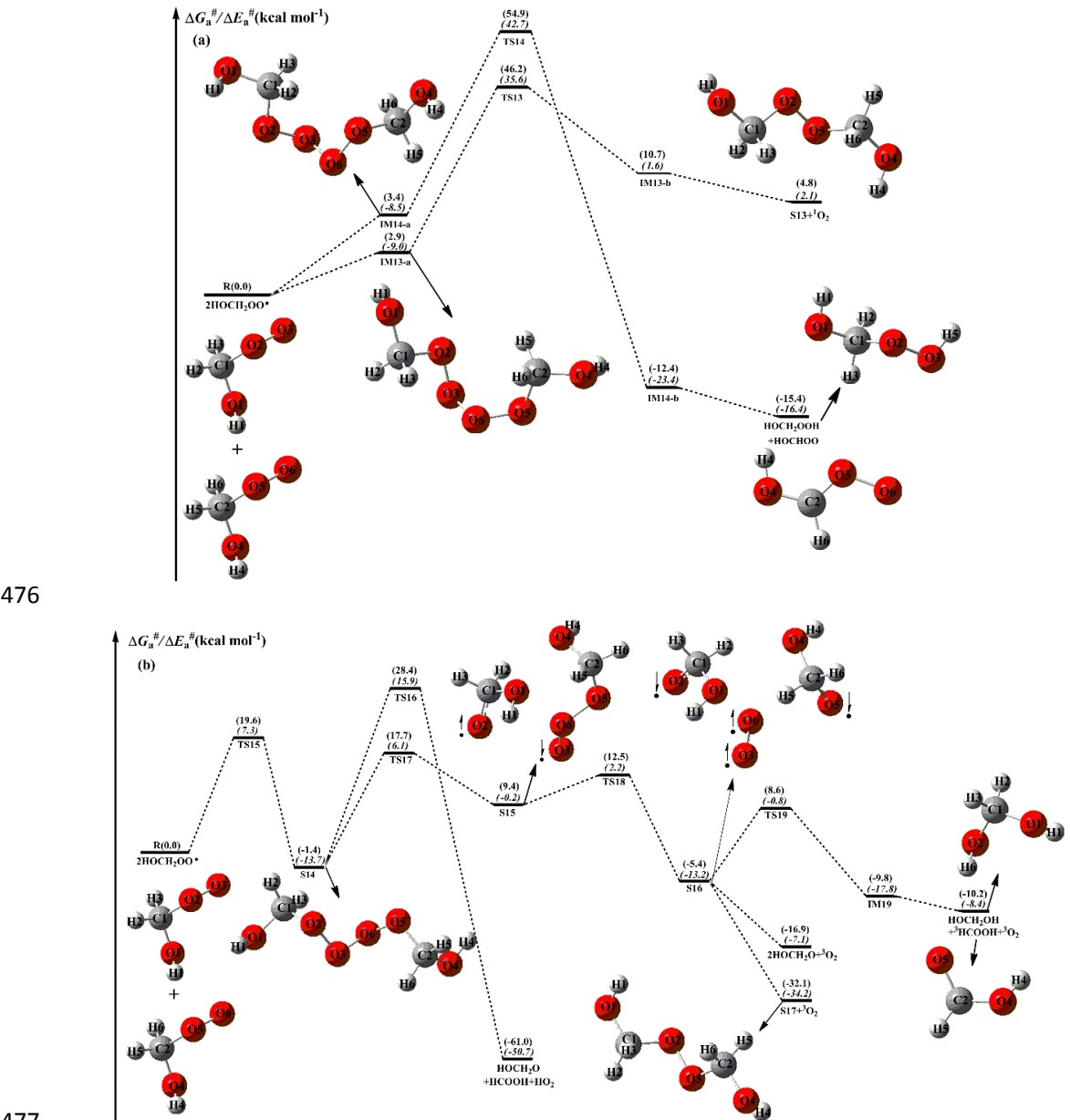



**Figure 5.** PES ($\Delta G_a^{\#}$ and $\Delta E_a^{\#}$, in italics) for the self-reaction of HOCH$_2$OO radicals predicted at the M06-2X/ma-TZVP//M06-2X/6-311+G(2df,2p) level of theory

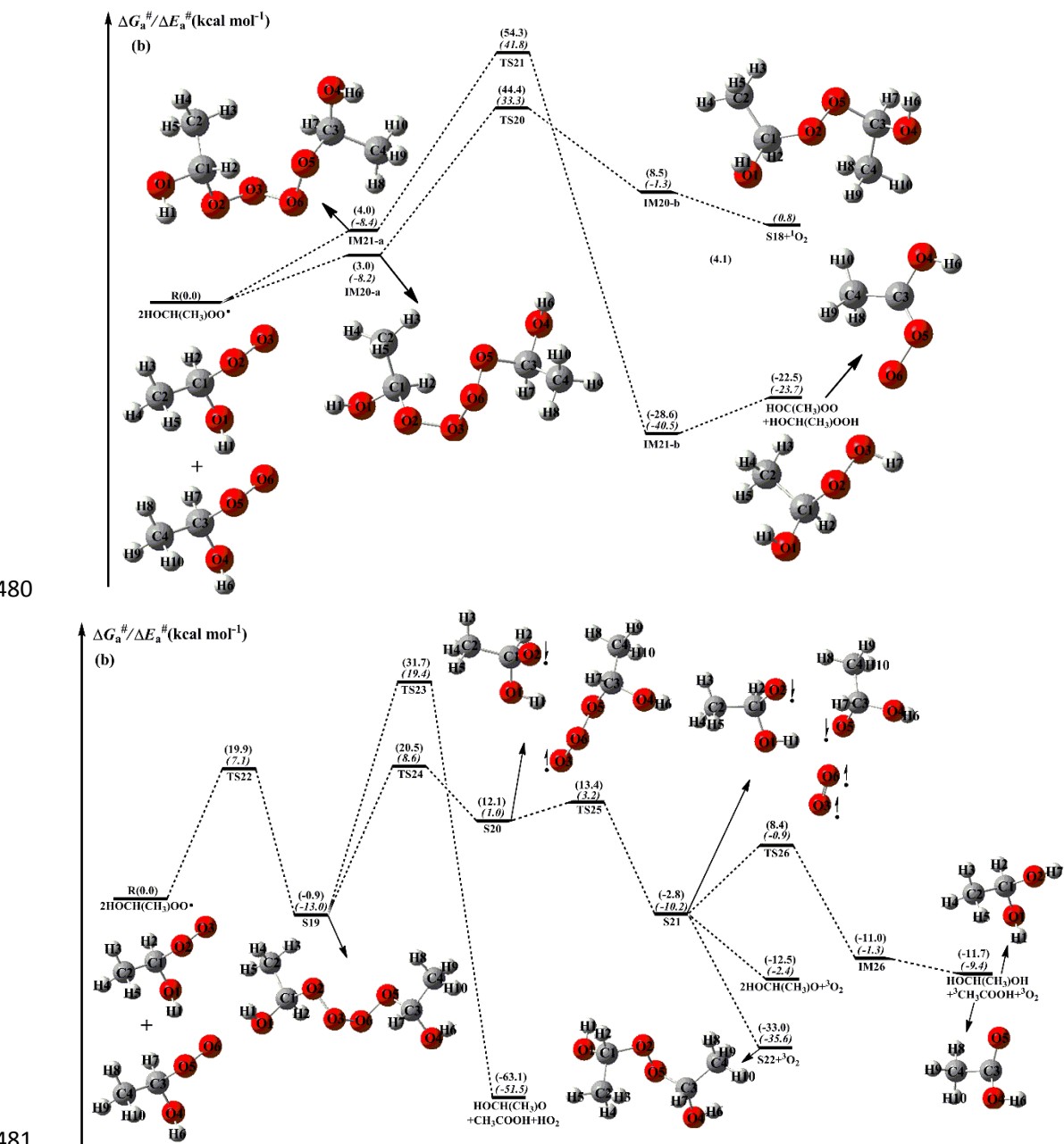



**Figure 6.** PES ($\Delta G_a^{\#}$ and $\Delta E_a^{\#}$, in italics) for the self-reaction of HOCH(CH$_3$)OO radicals predicted at the M06-2X/ma-TZVP//M06-2X/6-311+G(2df,2p) level of theory

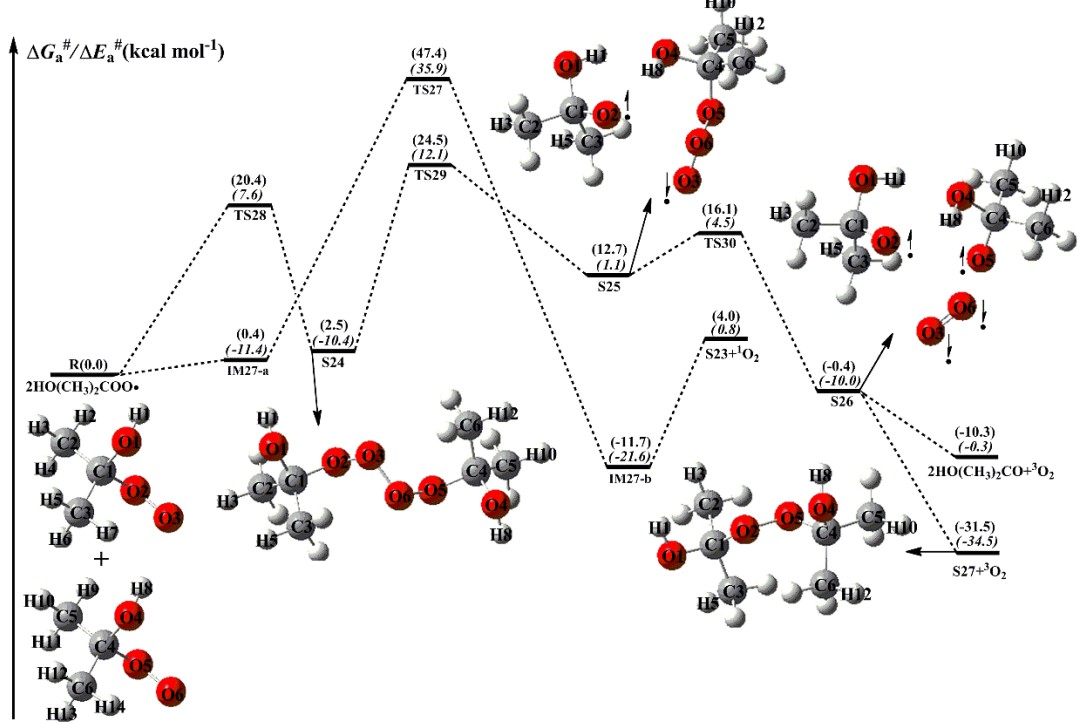

**Figure 7.** PES ($\Delta G_a^{\#}$ and $\Delta E_a^{\#}$, in italics) for the self-reaction of HO(CH$_3$)$_2$COO radicals predicted at the M06-2X/ma-TZVP//M06-2X/6-311+G(2df,2p) level of theory

### 3.2.2 Reactions mechanism for the reaction of RO$_2$ radicals with HO$_2$ radical

When NO is present in low concentration, the bimolecular reaction of RO$_2$ radicals with HO$_2$ radical is generally expected to be the dominant pathway as the main product hydroperoxide ROOH. The primary sources of HO$_2$ radical involve the photo-oxidation of oxygenated volatile organic compounds (OVOCs) and the ozonolysis reaction, as well as secondary sources include the reactions of OH radical with CO, ozone and volatile organic compounds (VOCs), the reaction of alkoxy radical RO with O$_2$ and the red-light-induced decomposition of α-hydroxy methylperoxy radical OHCH$_2$OO (Kumar and Francisco, 2015; Stone et al., 2012; Hofzumahaus et al., 2009). The atmospheric concentration of HO$_2$ radical is 1.5-10 $\times$ 10$^8$ molecules cm$^{-3}$ at ground level in polluted urban environments (Stone et al., 2012). A schematic PES for the reactions of distinct RO$_2$ radicals with HO$_2$ radical is presented in Figure 8. As shown in Figure 8, all the reactions are strongly exothermic and spontaneous, indicating that they are feasible thermodynamically in the atmosphere. The reaction for HOCH$_2$OO with HO$_2$ (R31) starts with the formation of

a pre-reactive complex IM31-a in the entrance channel, which is more stable than the
separate reactants by 3.8 kcal·mol$^{-1}$ in energy. Then it converts into HOCH$_2$OOH and
O$_2$ via a hydrogen atom transfer from the HO$_2$ radical to the terminal oxygen atom of
HOCH$_2$OO radical with the barrier of 2.0 kcal·mol$^{-1}$. The mechanism of
HOCH(CH$_3$)OO + HO$_2$ (R32) and HO(CH$_3$)$_2$COO + HO$_2$ (R33) reactions is quite
similar to that of HOCH$_2$OO + HO$_2$ system. In order to avoid redundancy, we will not
discuss them in detail. It deserves mentioning that the barrier height is only reduced
by 0.1 kcal·mol$^{-1}$ when one or two methyl substitutions occur at the C1-position of
HOCH$_2$OO radical, compared to the barrier of HOCH$_2$OO + HO$_2$ reaction. This result
implies that the barrier height is not seem to be influenced by the number of methyl
substitution. The rate coefficients of distinct RO$_2$ radical reactions with HO$_2$ radical
are summarized in Table S5 and Figure S12. As shown in Table S5, the rate
coefficients $k_{R31}$ of HOCH$_2$OO + HO$_2$ reaction vary from 3.1 $\times 10^{-11}$ (273 K) to 2.1 $\times$
$10^{-12}$ cm$^3$ molecule$^{-1}$ s$^{-1}$ (400 K), and they exhibit a negative temperature dependence.
Similar conclusion is also obtained from the rate coefficients $k_{R32}$ and $k_{R33}$ that they
decrease with the temperature increasing. It should be noted that the rate coefficient is
slightly increased as the number of methyl group is increased. At ambient temperature,
$k_{R31}$ is estimated to be 1.7 $\times 10^{-11}$ cm$^3$ molecule$^{-1}$ s$^{-1}$, which is in good agreement with
the value of ~2 $\times 10^{-11}$ cm$^3$ molecule$^{-1}$ s$^{-1}$ for the reaction of acyl peroxy radicals with
HO$_2$ radical (Wennberg et al., 2018). The typical atmospheric concentrations of HO$_2$
radical are 5, 20 and 50 pptv in the urban, rural and forest environments (Bianchi et
al., 2019), translating into the pseudo-first-order rate constants $k'_{HO2} = k_{HO2}$[HO$_2$] of
1.1 $\times 10^{-3}$, 4.2 $\times 10^{-3}$ and 1.1 $\times 10^{-2}$ s$^{-1}$, respectively. The pseudo-first-order rate
constants of R32 and R33 are predicted to be 3.0 $\times 10^{-3}$ and 4.8 $\times 10^{-3}$ (urban), 1.1 $\times$
$10^{-2}$ and 1.8 $\times 10^{-2}$ (rural), 3.0 $\times 10^{-2}$ and 4.8 $\times 10^{-2}$ s$^{-1}$ (forest) at room temperature.

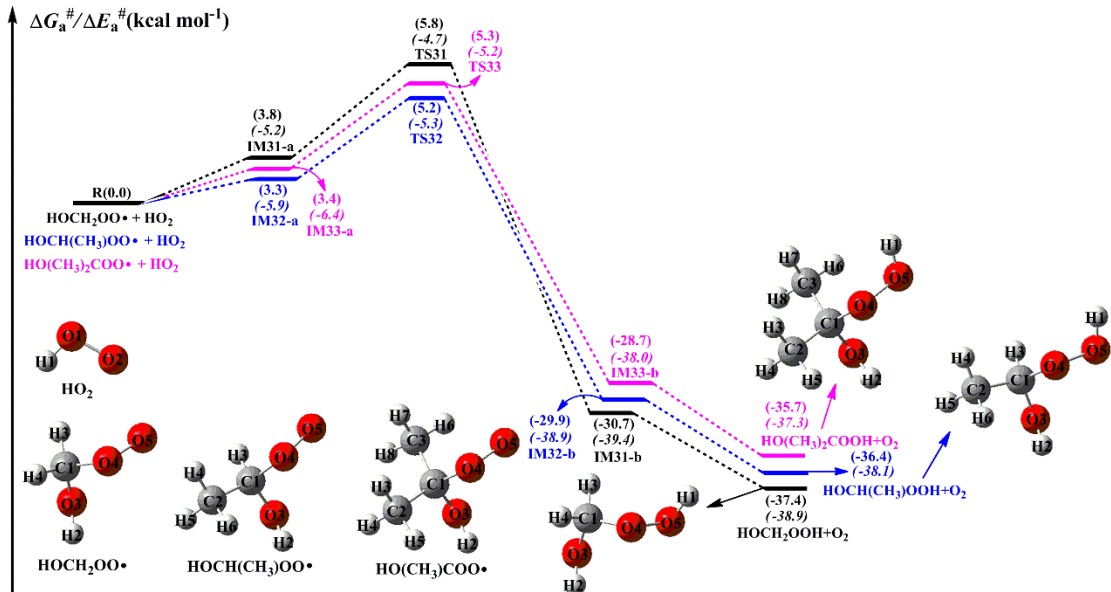


**Figure 8.** PES ($\Delta G_a^{\#}$ and $\Delta E_a^{\#}$, in italics) for the reactions of $HO_2$ radical with distinct $RO_2$ radicals predicted at the M06-2X/ma-TZVP//M06-2X/6-311+G(2df,2p) level of theory

### 3.2.3 Reactions mechanism for the isomerization of $RO_2$ radicals

Autoxidation of $RO_2$ radicals is known to play an important role in the (re)generation of $HO_x$ radicals and in the formation of HOMs (Xu et al., 2014; Bianchi et al., 2019; Rissanen et al., 2014; Ehn et al., 2017). The autoxidation mechanism includes an intramolecular H-shift from the -$CH_3$ or -$CH_2$- groups to the -OO site, leading to the formation of a hydroperoxyalkyl radical QOOH, followed by $O_2$ addition to form a new peroxy radical ($HOOQO_2$), one after the other, resulting in formation of HOMs (Rissanen et al., 2014; Berndt et al., 2015). For the H-shift reactions of $RO_2$ radicals, reactants, transition states and products have multiple conformers due to the effect of degree of freedom for internal rotation. Based on the calculated results, it can be found that $HOCH_2OO$ radical has four energetically similar conformers ($HOCH_2OO$-a, $HOCH_2OO$-b, $HOCH_2OO$-c and $HOCH_2OO$-d). The relative free energy and Boltzmann population ($w_i$) of individual conformer are listed in Table S6. As shown in Table S6, the Boltzmann populations of these four conformers are 46.39, 46.31, 2.99 and 4.32%, respectively.

A schematic PES for the H-shift reaction of $HOCH_2OO$ radical is displayed in Figure 9. As can be seen in Figure 9, the lowest-energy conformer $HOCH_2OO$-a can proceed via a 1,3-H shift from the -$CH_2$ group to the terminal oxygen leading to the

formation of S28-a (HOCHOOH) with the barrier of 41.6 kcal mol$^{-1}$. HOCH$_2$OO-b

can isomerize to S28-b1 and S28-b2 via the four-membered ring transition states

TS34-b1 and TS34-b2 (1,3-H shifts) with the barriers of 41.6 and 45.0 kcal mol$^{-1}$. But

these three 1,3-H shift reactions have comparatively high barriers, making them

irrelevant in the atmosphere. Despite many attempts, the transition states of H-shift

reactions of HOCH$_2$OO-c and HOCH$_2$OO-d are not located. The result implies that

the H-shift reactions of these two conformers are inhibited, which is consistent with

the previous study that not all reactants will be in a conformation with a path across

the barrier to reaction in the H-shift reactions of RO$_2$ radicals (Møller et al., 2016).

Equivalent to the case of HOCH$_2$OO radical, the isomerization of HOCH(CH$_3$)OO

radical proceeds via the 1,3- and 1,4-H shifts from the -CH or -CH$_3$ groups to the

terminal oxygen resulting in formation of hydroperoxyalkyl radicals (Figure S13).

These 1,3- and 1,4-H shift reactions accompany with the extremely high barriers (>

37.9 kcal mol$^{-1}$), implying that they are of less importance in the atmosphere. Similar

conclusion is also derived from the isomerization of HO(CH$_3$)$_2$COO radical that 1,4-H

shift reactions are unfavourable kinetically (Figure S14). The high barriers of 1,3- and

1,4-H shifts can be interpreted as the result of the large ring strain energy (RSE) in the

cyclic transition state geometries. As a consequence, the isomerization reactions of

HOCH$_2$OO, HOCH(CH$_3$)OO and HO(CH$_3$)$_2$COO radicals are unlikely to proceed in

the atmosphere. This conclusion is further supported by the previous studies that the

intramolecular H-shift isomerizations are important only for RO$_2$ radicals with larger

carbon structures (Crounse et al., 2013; Jokinen et al., 2014; Rissanen et al., 2014).

The single-conformer rate coefficients ($k_{\text{IRC-TST}}$) and multi-conformer rate

coefficients ($k_{\text{MC-TST}}$) of the isomerization of HOCH$_2$OO, HOCH(CH$_3$)OO and

HOC(CH$_3$)$_2$OO radicals are calculated over the temperature range of 273-400 K as

listed in Table S9-S11. As can be seen in Table S9, $k_{\text{IRC-TST}}$ of each conformer exhibits

a marked positive temperature dependence over the temperature range studied.

$k_{\text{MC-TST}}$ is significantly increased with rising temperature, implying that the

temperature increasing is beneficial to the occurrence of HOCH$_2$OO radical

isomerization. Similar conclusion is also obtained from the isomerization of

HOCH(CH$_3$)OO and HOC(CH$_3$)$_2$OO radicals (Table S10-S11). It is worth mentioning
that $k_{MC\text{-}TST}$ is rapidly increased as the number of methyl group is increased. For
example, the room temperature $k_{MC\text{-}TST}$ of HOCH$_2$OO radical isomerization is
calculated to be $4.4 \times 10^{-16}$ s$^{-1}$, which is lower than those of the HOCH(CH$_3$)OO (2.9
$\times 10^{-13}$ s$^{-1}$) and HO(CH$_3$)$_2$COO ($3.0 \times 10^{-12}$ s$^{-1}$) radicals isomerization by 660 and
6820 times, respectively.

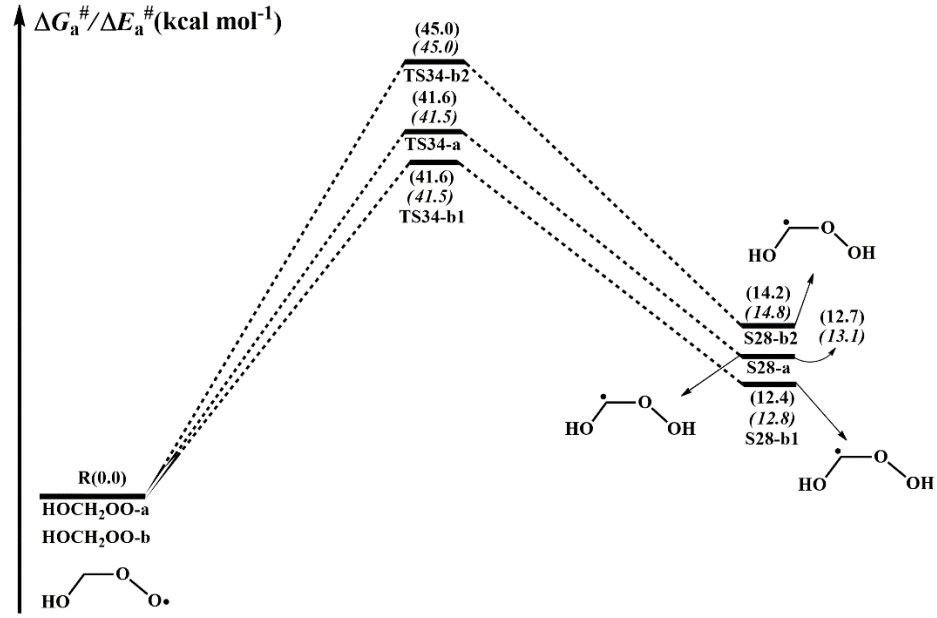

**Figure 9.** PES ($\Delta G_a^{\#}$ and $\Delta E_a^{\#}$, in italics) for the isomerization of HOCH$_2$OO radical predicted at
the M06-2X/ma-TZVP//M06-2X/6-311+G(2df,2p) level of theory

## 3.3 Subsequent reactions of H-abstraction products RO$_2$ radicals in urban environments

NO$_x$ is present in high concentration in urban environments, reaction with NO is
the dominant chemical sink for RO$_2$ radicals (Atkinson and Arey, 2003; Orlando and
Tyndall, 2012; Perring et al., 2013). The main pathways in this type of reaction lead to
the formations of NO$_2$, RO radicals, organic nitrites, and organic nitrates, and their
formation yields are highly dependent on the nature of R group (Orlando and Tyndall,
2012). The formation of NO$_2$ through subsequent photolysis ($\lambda < 420$ nm) produces
ozone and NO, increasing the concentrations of near-surface ozone and propagating
NO$_x$ chain (Orlando and Tyndall, 2012). The schematic PES for the reactions of
distinct RO$_2$ radicals with NO are displayed in Figures 10-12. As shown in Figure 10,
the bimolecular reaction of HOCH$_2$OO radical with NO initially leads to nitrite adduct
S31 via the barrierless addition of NO to terminal oxygen atom $O_3$ of $HOCH_2OO$
radical. The formed S31 exists two isomers: S31-*cis* refers to the $O_2$ and $O_4$ on the
same side ($DO_2O_3N_1O_4$ = 2.3°), whereas S31-*trans* refers to the $O_2$ and $O_4$ on the
opposite side ($DO_2O_3N_1O_4$ = -179.8°) with respect to the $O_3$-$N_1$ bond. The
calculations show that S31-*cis* is more stable than S31-*trans* by 1.1 kcal·mol$^{-1}$ in
energy. The tautomerization between S31-*cis* and S31-*trans* proceeds through the
rotating of $O_3$-$N_1$ bond with the barrier of 14.4 kcal·mol$^{-1}$, implying that they can be
regarded as the separate atmospheric species. According to the Boltzmann-weighted
distribution, at room temperature, the predicted percentages of S31-*cis* and S31-*trans*
are 86.5% and 13.5%, respectively. The result implies that the dominant product of
$HOCH_2OO$ radical reaction with NO is S31-*cis*, and it is selected as a model
compound to insight into the mechanism of secondary reactions in the following
sections.
S31-*cis* can either isomerize to organic nitrate S32 (R38) via a concerted
process of $O_2$-$O_3$ bond breaking and $O_2$-$N_1$ bond forming with the barrier of 47.8
kcal·mol$^{-1}$, or decompose into $HOCH_2O$ radical and $NO_2$ (R39) via the cleavage of
$O_2$-$O_3$ bond with the barrier of 11.3 kcal·mol$^{-1}$. The result shows that the latter
pathway is more favourable than the former channel. Similar conclusion is also
obtained from the reactions of NO with $HOCH(CH_3)OO$ and $HO(CH_3)_2COO$ radicals
that the formation of organic nitrate is of minor importance in the atmosphere. This
result is further supported by the prior studies that the direct formation of organic
nitrate from peroxy nitrites is a minor channel in the reactions of isoprene-derived
$RO_2$ radicals with NO (Piletic et al., 2017; Zhang et al., 2002). It should be noted that
the transition state TS39 is not located using M06-2X functional, but it is located at
the MP2/6-311+G(2df,2p) level of theory and is verified using IRC calculations. The
formed $HOCH_2O$ radical has two possible pathways: (1) it directly decomposes into
$CH_2O$ and OH radical (R40) via $\beta$-site $C_1$-$O_1$ bond scission with the barrier of 52.4
kcal·mol$^{-1}$; (2) it converts into HCOOH and $HO_2$ radical (R41) through H-abstraction
by $O_2$ with the barrier of 26.4 kcal·mol$^{-1}$. This result reveals that R41 is the most
feasible channel in the fragmentation of $HOCH_2O$ radical.
From Figure 11, it can be seen that the addition NO to $HOCH(CH_3)OO$ radical
leading to the formation of S33-*cis* is barrierless. Then, it decomposes into
$HOCH(CH_3)O$ radical and $NO_2$ (R44) via the cleavage of $O_2$-$O_3$ bond with the barrier
of 11.5 kcal mol$^{-1}$. The resulting $HOCH(CH_3)O$ radical has three possible pathways.
The first one is $\beta$-site $C_1$-$C_2$ bond scission leading to the formation of $HCOOH + CH_3$
(R45) with the barrier of 8.3 kcal mol$^{-1}$. The second one is $\beta$-site $C_1$-$O_1$ bond cleavage
resulting in formation of $CH_3COH + OH$ (R46) with the barrier of 26.7 kcal mol$^{-1}$.
The third one is H-abstraction by $O_2$ leading to $CH_3COOH + HO_2$ (R47) with the
barrier of 26.2 kcal mol$^{-1}$. Based on the calculated reaction barriers, it can be found
that $\beta$-site $C_1$-$C_2$ bond scission is the dominant pathway in the fragmentation of
$HOCH(CH_3)O$ radical. This conclusion is further supported by the previous
experimental result that $\beta$-hydroxy intermediates primarily proceed decomposition
rather than react with $O_2$ in the presence of NO (Aschmann et al., 2000). Equivalent to
the $HOCH(CH_3)OO + NO$ reaction, the bimolecular reaction of $HO(CH_3)_2COO$
radical with NO has similar transformation pathways (Figure 12). The reaction for
$HO(CH_3)_2COO$ with NO initially proceeds via a barrierless addition leading to
S35-*cis* with the binding energy of 12.6 kcal mol$^{-1}$. Then, S35-*cis* fragments into
$HO(CH_3)_2CO$ radical along with $NO_2$ (R50) via the cleavage of $O_2$-$O_3$ bond with the
barrier of 11.4 kcal mol$^{-1}$. The formed $HO(CH_3)_2CO$ radical can either dissociate to
$CH_3COOH + CH_3$ (R51) via the $C_1$-$C_3$ bond scission with the barrier of 8.2 kcal mol$^{-1}$,
or decompose into $CH_3COCH_3 + OH$ (R52) through the $C_1$-$O_1$ bond breaking with the
barrier of 24.3 kcal mol$^{-1}$. The result again shows that the $\beta$-site C-C bond scission is
the dominant pathway.
The typical atmospheric concentrations of NO are about 10 ppbv, 1 ppbv and 20
pptv in the urban, rural and forest environments (Bianchi et al., 2019). The rate
coefficient of $HOCH_2OO$ radical reaction with NO is calculated to be $4.3 \times 10^{-12}$ cm$^3$
molecule$^{-1}$ s$^{-1}$ at room temperature, resulting in the pseudo-first-order rate constants
$k'_{NO} = k_{NO}[NO]$ of $6.5 \times 10^{-1}$, $6.5 \times 10^{-2}$, and $1.3 \times 10^{-3}$, respectively, in the urban,
rural and forest environments. It is of interest to assess the relative importance for the
H-shift reaction of $HOCH_2OO$ radical and bimolecular reactions with $HO_2$ radical and
NO based on the calculated $k_{\text{MC-TST}}$, $k'_{\text{HO2}}$ and $k'_{\text{NO}}$. It can be found that the H-shift
reaction is of less importance, the $HO_2$ radical reaction is favorable in the forest
environments, while the NO reaction is predominant in the urban and rural regions.
Similar conclusion is also obtained from the cases of $HOCH(CH_3)OO$ and
$HO(CH_3)_2CHOO$ radicals.

666        The rate coefficients of the dominant pathways of $HOCH_2O$, $HOCH(CH_3)O$ and

$HO(CH_3)_2CHO$ radicals fragmentation are summarized in Table S12. As can be seen
in Table S12, $k_{\text{R41}}$ is slightly increased with the temperature increasing, and the
discrepancy is about a factor of 12 at the two extremes of temperature. At ground
level with $[O_2] = \sim 5.0 \times 10^{18}$ molecule $cm^{-3}$, the pseudo-first-order rate constant $k'_{\text{O2}}$
$= k_{\text{R41}}[O_2]$ is estimated to be 38.0 $s^{-1}$ at room temperature. $k_{\text{R45}}$ vary significantly from
$2.0 \times 10^6$ (273 K) to $3.1 \times 10^8$ (400 K) $s^{-1}$, and they exhibit a marked positive
temperature dependence. Similar phenomenon is also observed from $k_{\text{R51}}$ that $k_{\text{R51}}$ is
significantly increased with increasing temperature. $k_{\text{R51}}$ is a factor of ~1.3 greater
than $k_{\text{R45}}$ in the temperature range studied, implying that the rate coefficient of $\beta$-site
C-C bond scission is slightly increased as the number of methyl group is increased.

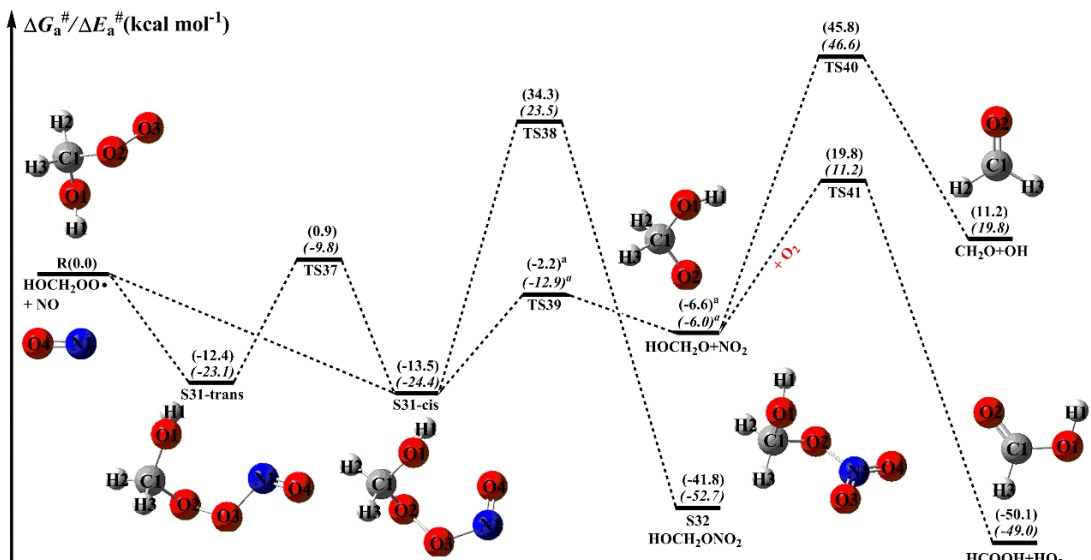


**Figure 10.** PES ($\Delta G_a^{\#}$ and $\Delta E_a^{\#}$, in italics) for the reaction of $HOCH_2OO$ radical with NO
predicted at the M06-2X/ma-TZVP//M06-2X/6-311+G(2df,2p) level of theory (the superscript a is
calculated at the MP2/ma-TZVP//MP2/6-311+G(2df,2p) level)

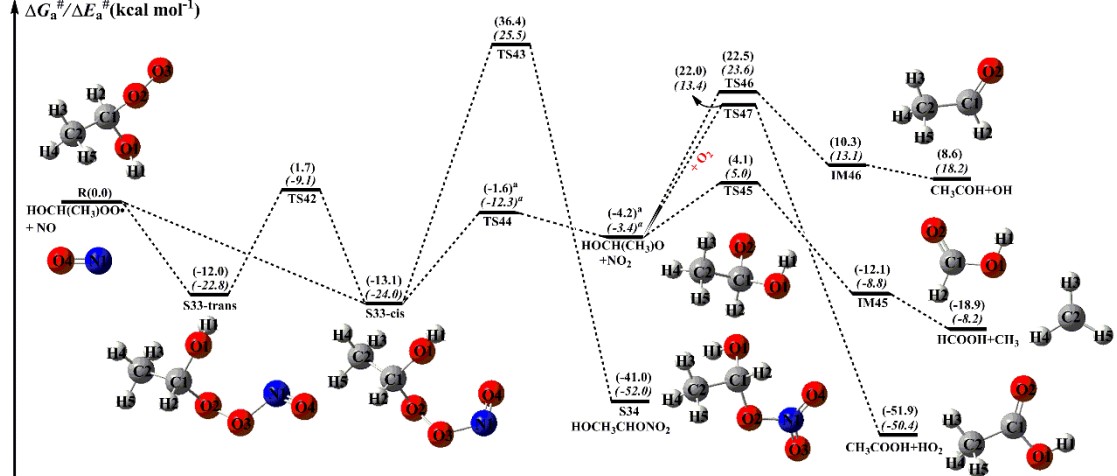

**Figure 11.** PES ($\Delta G_a^{\#}$ and $\Delta E_a^{\#}$, in italics) for the reaction of HOCH(CH₃)OO radical with NO predicted at the M06-2X/ma-TZVP//M06-2X/6-311+G(2df,2p) level of theory (the superscript a is calculated at the MP2/ma-TZVP//MP2/6-311+G(2df,2p) level)

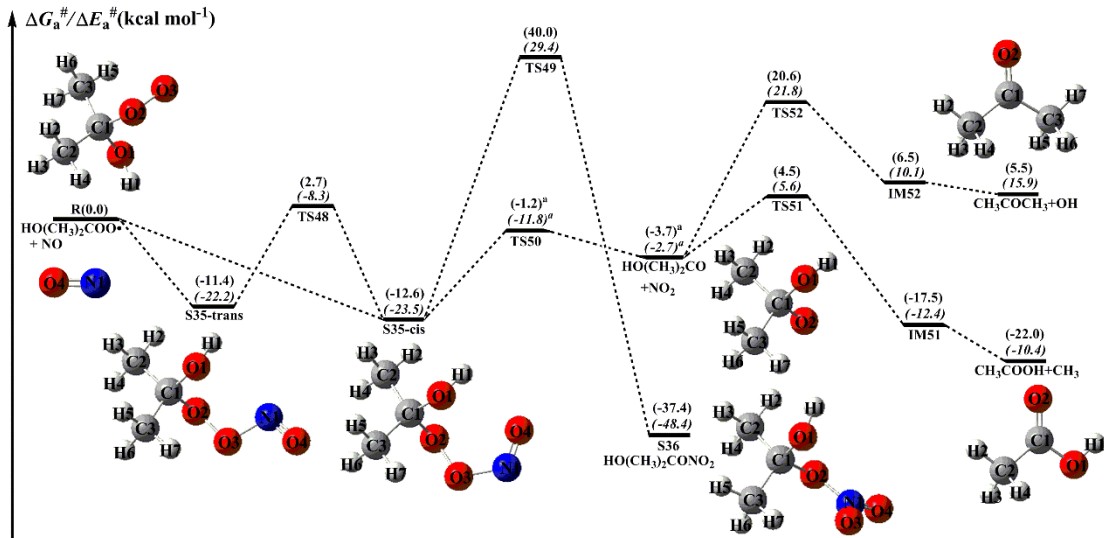

**Figure 12.** PES ($\Delta G_a^{\#}$ and $\Delta E_a^{\#}$, in italics) for the reaction of HO(CH₃)₂COO radical with NO predicted at the M06-2X/ma-TZVP//M06-2X/6-311+G(2df,2p) level of theory (the superscript a is calculated at the MP2/ma-TZVP//MP2/6-311+G(2df,2p) level)

## 4. Conclusions

The detailed mechanisms and kinetic properties of OH-initiated oxidation of distinct HHPs and subsequent transformation of resulting H-abstraction products are investigated using quantum chemical and kinetics modeling methods. The main conclusions are summarized as follows:

(a) The dominant pathway is the H-abstraction from the -OOH group in the initiation reactions of OH radical with HOCH₂OOH and HOC(CH₃)₂OOH. H-abstraction from the -CH group is competitive with that from the -OOH group in

the reaction of OH radical with $HOCH(CH_3)OOH$. The barrier of H-abstraction from the -OOH group is slightly increased as the number of methyl group is increased. Compared with the rate coefficient of dominant pathway in the parent system, it is almost identical when a methyl group substitution occurs at the $C_1$-position, whereas it reduces by a factor of 2-5 when two methyl groups are introduced into the $C_1$-position. The atmospheric lifetime of $HOCH_2OOH$, $HOCH(CH_3)OOH$ and $HOC(CH_3)_2OOH$ reactivity toward OH radical are estimated to be 0.58-1.74 h, 0.60-1.79 h and 1.23-3.69 h at room temperature under the typical OH radical concentrations of $5\text{-}15 \times 10^6$ molecules $cm^{-3}$ during daylight.

(b) The self-reaction of H-abstraction product $RO_2$ radical initially produces tetroxide intermediate via an oxygen-to-oxygen coupling, then it decomposes into propagation and termination products through the asymmetric two-step O-O bond scission. The rate-limiting step is the first O-O bond cleavage, and the barrier is increased with increasing the number of methyl group. This finding is meaningful to understand the self-reaction of complex $RO_2$ radicals.

(c) The bimolecular reactions of distinct $RO_2$ radicals with $HO_2$ radical lead to the formation of hydroperoxide ROOH as the main product, and the barrier height is independent on the number of methyl substitution. When compared to the rate coefficient for $HOCH_2OO + HO_2$ reaction, the rate coefficients increase by a factor of 2-5 when one or two methyl groups are introduced into the C1-position. Using a $HO_2$ radical concentration of ~50 pptv in the forest environments, the pseudo-first-order rate constants $k'_{HO2}$ of distinct $RO_2$ radical reactions with $HO_2$ radical vary from 1 to 5 $\times 10^{-2}$ $s^{-1}$.

(d) The isomerization reactions of $HOCH_2OO$, $HOCH(CH_3)OO$ and $HO(CH_3)_2COO$ radicals are unlikely to proceed in the atmosphere because the intramolecular H-shift steps have dramatically high barriers and strongly endergonic. The result implies that the isomerization of $RO_2$ radicals with smaller carbon structures is of less importance in the atmosphere.

(e) Reaction with $O_2$ forming formic acid and $HO_2$ radical is the dominant removal pathway for $HOCH_2O$ radical formed from the reaction of $HOCH_2OO$

radical with NO. The $\beta$-site C-C bond scission is the dominant pathway in the
dissociation of $HOCH(CH_3)O$ and $HOC(CH_3)_2O$ radicals formed from the reactions
of NO with $HOCH(CH_3)OO$ and $HOC(CH_3)_2OO$ radicals. The result implies that the
methyl-substituted alkoxyl radicals primarily proceed via $\beta$-site C-C bond scission to
produce aldehyde rather than react with $O_2$.

## Data availability

The data are accessible by contacting the corresponding author
(huangyu@ieecas.cn).

## Supplement

The following information is provided in the Supplement: Y//X (Y = M06-2X,
CCSD(T), X = 6-311+G(2df,2p), ma-TZVP) calculated energy barrier ($\Delta E_a^{\#}$, $\Delta G_a^{\#}$)
for OH + HHPs reactions; Rate coefficients of every elementary pathway involved in
the initial reactions of OH radical with $HOCH_2OOH$, $HOCH(CH_3)OOH$ and
$HO(CH_3)_2COOH$; Rate coefficients of $HO_2$ radical reactions with $HOCH_2OO$,
$HOCH(CH_3)OO$ and $HO(CH_3)_2COO$ radicals; The relative free energy and
Boltzmann populations ($w_i$) of the conformer of $HOCH_2OO$, $HOCH(CH_3)OO$ and
$HO(CH_3)_2COO$ radicals; The single-conformer rate coefficients ($k_{IRC-TST}$) and
multi-conformer rate coefficients ($k_{MC-TST}$) of $HOCH_2OO$, $HOCH(CH_3)OO$ and
$HO(CH_3)_2COO$ radicals; Rate coefficients of dominant pathways in the $HOCH_2OO \cdot +$
NO, $HOCH(CH_3)OO \cdot + NO$ and $HO(CH_3)_2CHOO \cdot + NO$ reactions; PESs ($\Delta E_a^{\#}$) for
the OH-initiated reactions of $HOCH_2OOH$, $HOCH(CH_3)OOH$, $HOC(CH_3)_2OOH$;
Geometries of all the stationary points; Plots of the rate coefficients of every
elementary pathway versus temperature; PESs ($\Delta G_a^{\#}$ and $\Delta E_a^{\#}$, in italics) for the
isomerization of $HOCH(CH_3)OO$ and $HO(CH_3)_2COO$ radicals.

## Author contribution

LC designed the study. LC and YH wrote the paper. LC performed theoretical
calculation. YX, ZJ, and WW analyzed the data. All authors reviewed and commented
on the paper.

## Competing interests

The authors declare that they have no conflict of interest.

## Acknowledgments

This work was supported by the National Natural Science Foundation of China
(grant Nos. 42175134, 41805107, and 22002080). It was also partially supported as
Key Projects of Chinese Academy of Sciences, China (grant No. ZDRW-ZS-2017-6),
Strategic Priority Research Program of the Chinese Academy of Sciences, China
(grant Nos. XDA23010300 and XDA23010000), Key Project of International
Cooperation of the Chinese Academy of Sciences, China (grant No. GJHZ1543),
Research Grants Council of Hong Kong, China (grant No. PolyU 152083/14E), CAS
"Light of West China" Program (XAB2019B01) and the General Project of Shaanxi
Province (2020JQ-432).

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
