# Peer review of "atmospheric degradation OH-Initiated of hydroxyalkyl 1 hydroperoxides: mechanism, kinetics, and structure-activity 2 relationship 3 Long Chen, 1,2 Yu Huang, \*, 1,2 Yonggang Xue, 1,2 Zhihui Jia, 3 Wenliang Wang 4 4 1 State Key Lab of Loess and Quaternary Geology (SKLLQG), Institute of Earth 5 Environment, Chinese Academy of Sciences (CAS), Xi'an, 710061, China 6 2 CAS Center for Excellence in Quat"

_Atmospheric Chemistry and Physics, 2021_

## Author Comment (AC1)

Prof. Yu Huang
State Key Lab of Loess and Quaternary Geology
Institute of Earth Environment, Chinese Academy
of Sciences, Xi'an, 710061, China
Tel./Fax: (86) 29-62336261
E-mail: huangyu@ieecas.cn

Jan. 27, 2022

Dear reviewer,

**Revision for Manuscript acp-2021-890**

We thank you very much for giving us the opportunity to revise our manuscript. We highly appreciate the reviewer for their comments and suggestions on the manuscript entitled "**OH-initiated atmospheric degradation of hydroxyalkyl hydroperoxides: mechanism, kinetics, and structure-activity relationship**". We have made revisions of our manuscript carefully according to the comments and suggestions of reviewer. The revised contents are marked in blue color. The response letter to reviewer is attached at the end of this cover letter.

We hope that the revised manuscript can meet the requirement of Atmospheric Chemistry & Physics. Any further modifications or revisions, please do not hesitate to contact us.

Look forward to hearing from you as soon as possible.

Best regards,

Yu Huang

**Comments of reviewer #1**

1. The investigated HHPs are generated from the reactions of $CH_2OO$, anti-$CH_3CHOO$ and $(CH_3)_2COO$ with water vapor, not considering the HHP from the bimolecular reaction of syn-$CH_3CHOO$ with water. This should be stated.

   **Response:** Based on the Reviewer's suggestion, OH-initiated oxidation of hydroxyalkyl hydroperoxide (HHP), generated from the bimolecular reaction of *syn*-$CH_3CHOO$ with water, has been added in the revised manuscript. The corresponding free-energy and electronic-energy potential energy surface (PES) are displayed in Figures S4 and S5, respectively. As shown in Figure S4, the H-abstraction by OH radical from $HOCH(CH_3)OOH$ has six kinds of pathways. For each pathway, a per-reactive complex is formed prior to the corresponding transition state, and then it overcomes modest barrier to reaction. The $\Delta G_a^{\#}$ of R6' and R8' are 2.3 and 1.8 kcal·mol$^{-1}$, respectively, which are ~ 5 kcal·mol$^{-1}$ lower than those of R5' and R7'. This result shows that H-abstraction from the -CH (R6') and -OOH (R8') groups are preferable kinetically. Same conclusion is also derived from the energy barriers $\Delta E_a^{\#}$ that R6' and R8' the most favourable H-abstraction pathways (Figure S5). It should be noted that although the barriers of R6' and R8' are comparable, the exoergicity of the former case is significantly lower than that of the latter case. The above-mentioned conclusions are consistent with the results derived from the OH-initiated oxidation of $HOCH(CH_3)OOH$ from the *anti*-$CH_3CHOO$ + $H_2O$ reaction. Zhou et al. has demonstrated that the bimolecular reaction of *syn*-$CH_3CHOO$ with water leading to the formation of $HOCH(CH_3)OOH$ is of less importance in the atmosphere, while the unimolecular decay to OH radical is the major loss process of *syn*-$CH_3CHOO$ (Zhou et al., 2019). Therefore, in the present study, we mainly focus on the subsequent mechanism of intermediate generated from OH-initiated oxidation of $HOCH(CH_3)OOH$ from the *anti*-$CH_3CHOO$ + $H_2O$ reaction.

[Figure]

**Figure S4.** PES ($\Delta G_a^{\#}$) for the OH-initiated reactions of HOCH(CH$_3$)OOH from the *syn*-CH$_3$CHOO + H$_2$O reaction predicted at the M06-2X/ma-TZVP//M06-2X/6-311+G(2df,2p) level of theory (a and b represent the pre-reactive and post-reactive complexes)

[Figure]

**Figure S5.** PES ($\Delta E_a^{\#}$) for the OH-initiated reactions of HOCH(CH$_3$)OOH from the *syn*-CH$_3$CHOO + H$_2$O reaction predicted at the M06-2X/ma-TZVP//M06-2X/6-311+G(2df,2p) level of theory (a and b represent the pre-reactive and post-reactive complexes)

Corresponding descriptions have been added in the page 11 line 289-308 of the revised manuscript:

*For the OH-initiated oxidation of HOCH(CH₃)OOH from the syn-CH₃CHOO + H₂O reaction, the corresponding free-energy and electronic-energy PESs are displayed in Figures S4 and S5, respectively. From Figure S4, it can be seen the H-abstraction by OH radical from HOCH(CH₃)OOH has six kinds of pathways. For each pathway, a per-reactive complex is formed prior to the corresponding transition state, and then it overcomes modest barrier to reaction. The $\Delta G_a^{\#}$ of R6' and R8' are 2.3 and 1.8 kcal mol$^{-1}$, respectively, which are about 5 kcal mol$^{-1}$ lower than those of R5' and R7'. This result shows that H-abstraction from the -CH (R6') and -OOH (R8') groups are preferable kinetically. Same conclusion is also derived from the energy barriers $\Delta E_a^{\#}$ that the R6' and R8' the most favourable H-abstraction pathways (Figure S5). It should be noted that although the barriers of R6' and R8' are comparable, the exoergicity of the former case is significantly lower than that of the latter case. The above-mentioned conclusions are consistent with the results derived from the OH-initiated oxidation of HOCH(CH₃)OOH from the anti-CH₃CHOO + H₂O reaction. Zhou et al. has demonstrated that the bimolecular reaction of syn-CH₃CHOO with water leading to the formation of HOCH(CH₃)OOH is of less importance in the atmosphere, while the unimolecular decay to OH radical is the major loss process of syn-CH₃CHOO (Zhou et al., 2019). Therefore, in the present study, we mainly focus on the subsequent mechanism of intermediate generated from OH-initiated oxidation of HOCH(CH₃)OOH from the anti-CH₃CHOO + H₂O reaction.*

2. Line 226-228, the reaction barriers are reduced in the order of 6.4 (R1) > 5.8 (R3) ≈ 5.4 (R2) > 1.5 (R4) kcal mol$^{-1}$, indicating that H-abstraction from the -OOH group is the most favorable. The authors should explain the order of TS1, TS3, TS2, and TS4 in the initial H-abstraction reactions.

**Response:** Based on the Reviewer's suggestion, the corresponding explanations on the order of barrier heights of H-abstraction reactions have been added in the revised manuscript. A schematic PES for the initiation reactions of OH radical with HOCH₂OOH is drawn in Figure 2. As can be seen in Figure 2, the reaction for HOCH₂OOH with OH radical proceeds via four distinct pathways: H-abstraction from the -O₁H₁ (R1), -C₁H₃ (R2), -C₁H₄ (R3) and -O₂O₃H₂ groups (R4). For each pathway, a pre-reactive complex with a six- or seven-membered ring

structure is formed in the entrance channel, which is stabilized by hydrogen bond interactions between the oxygen atom of OH radical and the abstraction hydrogen atom of HOCH$_2$OOH, and the remnant hydrogen atom of OH radical and one of oxygen atoms of HOCH$_2$OOH. Then, it surmounts modest barrier that is higher in energy than the reactants to reaction. The reaction barrier $\Delta G_a^{\#}$ are reduced in the order of 6.4 (R1) > 5.8 (R2) ≈ 5.4 (R3) > 1.5 (R4) kcal mol$^{-1}$, indicating that H-abstraction from the -O$_2$O$_3$H$_2$ group (R4) is more preferable than those from the -O$_1$H$_1$, -C$_1$H$_3$ and -C$_1$H$_4$ groups (R1-R3). Same conclusion is also derived from the energy barriers $\Delta E_a^{\#}$ that R4 is the most favorable H-abstraction pathway (Figure S1). The difference of barrier heights can be attributed to the bond dissociation energy (BDE) of different types of bonds in HOCH$_2$OOH molecule. The BDE are decreased in the order of 103.7 (O$_1$-H$_1$) > 98.2 (C$_1$-H$_3$) ≈ 97.4 (C$_1$-H$_4$) > 87.2 (O$_3$-H$_2$) kcal mol$^{-1}$, which are in good agreement with the order of barrier heights of H-abstraction reactions.

[Figure]

**Figure 2.** PES ($\Delta G_a^{\#}$) for the OH-initiated reactions of HOCH$_2$OOH from the CH$_2$OO + H$_2$O reaction predicted at the M06-2X/ma-TZVP//M06-2X/6-311+G(2df,2p) level of theory (a and b represent the pre-reactive and post-reactive complexes)

[Figure]

**Figure S1.** PES ($\Delta E_a^{\#}$) for the OH-initiated reactions of HOCH$_2$OOH from the CH$_2$OO + H$_2$O reaction predicted at the M06-2X/ma-TZVP//M06-2X/6-311+G(2df,2p) level of theory (a and b represent the pre-reactive and post-reactive complexes)

Corresponding descriptions have been added in the page 10 line 253-269 of the revised manuscript:

*As can be seen in Figure 2, the reaction for HOCH$_2$OOH with OH radical proceeds via four distinct pathways: H-abstraction from the -O$_1$H$_1$ (R1), -C$_1$H$_3$ (R2), -C$_1$H$_4$ (R3) and -O$_2$O$_3$H$_2$ groups (R4). For each pathway, a pre-reactive complex with a six- or seven-membered ring structure is formed in the entrance channel, which is stabilized by hydrogen bond interactions between the oxygen atom of OH radical and the abstraction hydrogen atom of HOCH$_2$OOH, and the remnant hydrogen atom of OH radical and one of oxygen atoms of HOCH$_2$OOH. Then, it surmounts modest barrier that is higher in energy than the reactants to reaction. The reaction barrier $\Delta G_a^{\#}$ are reduced in the order of 6.4 (R1) > 5.8 (R2) ≈ 5.4 (R3) > 1.5 (R4) kcal mol$^{-1}$, indicating that H-abstraction from the -O$_2$O$_3$H$_2$ group (R4) is more preferable than those from the -O$_1$H$_1$, -C$_1$H$_3$ and -C$_1$H$_4$ groups (R1-R3). Same conclusion is also derived from the energy barriers $\Delta E_a^{\#}$ that R4 is the most favorable H-abstraction pathway (Figure S1). The difference of barrier heights can be attributed to the bond dissociation energy (BDE) of different types of bonds in HOCH$_2$OOH molecule. The BDE are decreased in the order of 103.7 (O$_1$-H$_1$) > 98.2 (C$_1$-H$_3$) ≈ 97.4 (C$_1$-H$_4$) > 87.2 (O$_3$-H$_2$) kcal mol$^{-1}$, which are in good agreement with the order of barrier heights of H-abstraction reactions.*

3. The authors discuss the mechanism of $RO_2$ reactions with $HO_2$. But there is no information provided on $HO_2$. They must describe how $HO_2$ is formed in the atmosphere, what is its concentration, and where this reaction could be relevant.

**Response:** Based on the Reviewer's suggestion, the relevant information on the production of $HO_2$ radical has been added in the revised manuscript. The main sources of $HO_2$ radcial involve the photo-oxidation of oxygenated volatile organic compounds (OVOCs) and the ozonolysis reaction, as well as senondary sources include the reactions of OH radical with CO, ozone and volatile organic compounds (VOCs), the reaction of alkoxy radcial RO with $O_2$ and the red-light-induced decomposition of α-hydroxy methylperoxy radical $OHCH_2OO$ (Stone et al., 2012; Hofzumahaus et al., 2009; Kumar et al., 2015). The atmospheric concentration of $HO_2$ radical is 1.5-10 $\times 10^8$ molecules $cm^{-3}$ at ground level in polluted urban environments (Stone et al., 2012).

Corresponding descriptions have been added in the page 21 line 496-503 of the revised manuscript:

*The main sources of $HO_2$ radical involve the photo-oxidation of oxygenated volatile organic compounds (OVOCs) and the ozonolysis reaction, as well as senondary sources include the reactions of OH radical with CO, ozone and volatile organic compounds (VOCs), the reaction of alkoxy radical RO with $O_2$ and the red-light-induced decomposition of α-hydroxy methylperoxy radical $OHCH_2OO$ (Kumar et al., 2015; Stone et al., 2012; Hofzumahaus et al., 2009). The atmospheric concentration of $HO_2$ radical is 1.5-10 $\times 10^8$ molecules $cm^{-3}$ at ground level in polluted urban environments (Stone et al., 2012).*

4. Authors should compare $k_{MC\text{-}TST}$ and the pseudo first-order rates ($k'_{HO2}$ and $k'_{NO}$) for the bimolecular processes ($HO_2$ reaction and NO reaction) as a function of concentration. See the recent review of autoxidation by Bianchi et al. (Chem. Rev. 2019, 119, 6, 3472-3509).

**Response:** As the Reviewer's said, the relative importance of different transformation pathways (unimolecular, $HO_2 \cdot$ and NO reactions) of peroxy radicals $RO_2$ is significantly dependent on the rate coefficients and coreactant concentrations. For the H-shift reaction of $RO_2$ radicals, the multi-conformer rate coefficient $k_{MC\text{-}TST}$ can be calculated by the weighted sum of the single-conformer rate coefficient $k_{IRC\text{-}TST}$. At room temperature, $k_{MC\text{-}TST}$ of first H-shift reaction of

HOCH$_2$OO radical is calculated to be 4.4 $\times$ 10$^{-16}$ s$^{-1}$. The room temperature rate coefficient of HOCH$_2$OO radical reaction with HO$_2$ radical is estimated to be 1.7 $\times$ 10$^{-11}$ cm$^3$ molecule$^{-1}$ s$^{-1}$. The typical atmospheric concentrations of HO$_2$ radical are 5, 20 and 50 pptv in the urban, rural and forest environments (Bianchi et al., 2019), translating into the pseudo-first-order rate constants $k'_{HO2} = k_{HO2}[HO_2]$ of 1.1 $\times$ 10$^{-3}$, 4.2 $\times$ 10$^{-3}$ and 1.1 $\times$ 10$^{-2}$ s$^{-1}$, respectively.

The typical atmospheric concentrations of NO are about 10 ppbv, 1 ppbv and 20 pptv in the urban, rural and forest environments (Bianchi et al., 2019). The rate coefficient of HOCH$_2$OO radical reaction with NO is calculated to be 4.3 $\times$ 10$^{-12}$ cm$^3$ molecule$^{-1}$ s$^{-1}$ at room temperature, resulting in the pseudo-first-order rate constants $k'_{NO} = k_{NO}[NO]$ of 6.5 $\times$ 10$^{-1}$, 6.5 $\times$ 10$^{-2}$, and 1.3 $\times$ 10$^{-3}$, respectively, in the urban, rural and forest environments. It is of interest to assess the relative importance for the H-shift reaction of HOCH$_2$OO radical and bimolecular reactions with HO$_2$ ·and NO based on the calculated $k_{MC\text{-}TST}$, $k'_{HO2}$ and $k'_{NO}$. It can be found that the H-shift reaction is of less importance, the HO$_2$ radical reaction is favourable in the forest environment, the NO reaction is predominant in the urban and rural regions. Similar conclusion is also obtained from the cases of HOCH(CH$_3$)OO and HO(CH$_3$)$_2$CHOO radicals.

Corresponding descriptions have been added in the page 22 line 525-531 and page 27 line 653-664 of the revised manuscript:

*At ambient temperature, $k_{R31}$ is estimated to be 1.7 $\times$ 10$^{-11}$ cm$^3$ molecule$^{-1}$ s$^{-1}$, which is in good agreement with the value of ~2 $\times$ 10$^{-11}$ cm$^3$ molecule$^{-1}$ s$^{-1}$ for the reaction of acyl peroxy radicals with HO$_2$ radical (Wennberg et al., 2018). The typical atmospheric concentrations of HO$_2$ radical are about 5, 20 and 50 pptv in the urban, rural and forest environments (Bianchi et al., 2019), translating into the pseudo-first-order rate constants $k'_{HO2} = k_{HO2}[HO_2]$ of 1.1 $\times$ 10$^{-3}$, 4.2 $\times$ 10$^{-3}$ and 1.1 $\times$ 10$^{-2}$ s$^{-1}$, respectively.*

*The typical atmospheric concentrations of NO are about 10 ppbv, 1 ppbv and 20 pptv in the urban, rural and forest environments (Bianchi et al., 2019). The rate coefficient of HOCH$_2$OO · reaction with NO is calculated to be 4.3 $\times$ 10$^{-12}$ cm$^3$ molecule$^{-1}$ s$^{-1}$ at room temperature, resulting in the pseudo-first-order rate constants $k'_{NO} = k_{NO}[NO]$ of 6.5 $\times$ 10$^{-1}$, 6.5 $\times$ 10$^{-2}$, and 1.3 $\times$ 10$^{-3}$, respectively, in the urban, rural and forest environments. It is of interest to assess the relative importance for the H-shift reaction of HOCH$_2$OO radical and bimolecular reactions with HO$_2$ ·and NO based on the calculated $k_{MC\text{-}TST}$, $k'_{HO2}$ and $k'_{NO}$. It can be found that*

*the H-shift reaction is of less importance, the HO$_2$ radical reaction is favorable in the forest environment, while the NO reaction is predominant in the urban and rural regions. Similar conclusion is also obtained from the cases of HOCH(CH$_3$)OO and HO(CH$_3$)$_2$CHOO radicals.*

5. For the alkoxy radical fragmentation, the author should calculate rate constants in the temperature range studied.

**Response:** Based on the Reviewer's suggestion, the rate coefficients of the dominant pathways of alkoxyl radical fragmentation have been calculated over the temperature range of 273-400 K. The corresponding results are listed in Table S12 of the revised manuscript. For the fragmentation of HOCH$_2$O radical, the dominant pathway is H-abstraction by O$_2$ from HOCH$_2$O radical resulting in formation of HCOOH and HO$_2$ radical (R41). For the fragmentation of HOCH(CH$_3$)O and HOC(CH$_3$)$_2$O radicals, the dominant pathways are $\beta$-site C-C bond scission leading to the formation of HCOOH + CH$_3 \cdot$(R45) and CH$_3$COOH + CH$_3 \cdot$(R51). As can be seen in Table S12, $k_{R41}$ is slightly increased with the temperature increasing, and the discrepancy is about a factor of 12 at the two extremes of temperature. At ground level with [O$_2$] = ~ $5.0 \times 10^{18}$ molecule cm$^{-3}$, the pseudo-first-order rate constant $k'_{O2} = k_{R41}$[O$_2$] is estimated to be 38.0 s$^{-1}$ at room temperature. $k_{R45}$ vary significantly from $2.0 \times 10^6$ (273 K) to $3.1 \times 10^8$ (400 K) s$^{-1}$, and they exhibit a marked positive temperature dependence. Similar phenomenon is also observed from $k_{R51}$ that $k_{R51}$ is significantly increased with the temperature increasing. $k_{R51}$ is a factor of ~ 1.3 greater than $k_{R45}$ in the temperature range studied, implying that the rate coefficient of $\beta$-site C-C bond scission is slightly increased as the number of methyl group is increased.

**Table S12** Rate coefficients of the dominant pathways of the fragmentation of HOCH$_2$O $\cdot$(R41), HOCH(CH$_3$)O $\cdot$(R45) and HO(CH$_3$)$_2$CO $\cdot$(R51) computed at different temperatures

| T/K | $k_{R41}$(cm$^3$ molecule$^{-1}$ s$^{-1}$) | $k_{R45}$(s$^{-1}$) | $k_{R51}$(s$^{-1}$) |
|---|---|---|---|
| 273 | $4.3 \times 10^{-18}$ | $2.0 \times 10^6$ | $2.6 \times 10^6$ |
| 280 | $5.0 \times 10^{-18}$ | $2.9 \times 10^6$ | $3.8 \times 10^6$ |
| 298 | $7.6 \times 10^{-18}$ | $7.3 \times 10^6$ | $9.5 \times 10^6$ |
| 300 | $7.9 \times 10^{-18}$ | $8.1 \times 10^6$ | $1.0 \times 10^7$ |
| 320 | $1.2 \times 10^{-17}$ | $1.9 \times 10^7$ | $2.5 \times 10^7$ |
| 340 | $1.8 \times 10^{-17}$ | $4.4 \times 10^7$ | $5.6 \times 10^7$ |
| 360 | $2.6 \times 10^{-17}$ | $9.0 \times 10^7$ | $1.1 \times 10^8$ |

| | | | |
|---|---|---|---|
| 380 | $3.7 \times 10^{-17}$ | $1.7 \times 10^8$ | $2.2 \times 10^8$ |
| 400 | $5.1 \times 10^{-17}$ | $3.1 \times 10^8$ | $3.8 \times 10^8$ |

Corresponding descriptions have been revised in the page 26 line 625-629, page 27 line 633-640, page 27 line 648-652 and page 28 line 655-675 of the revised manuscript:

*The formed $HOCH_2O$ radical has two kinds of pathways: (1) it directly decomposes into $CH_2O$ and OH radical (R40) via β-site $C_1$-$O_1$ bond scission with the barrier of 52.4 kcal $mol^{-1}$; (2) it converts into HCOOH and $HO_2$ radical (R41) through H-abstraction by $O_2$ with the barrier of 26.4 kcal $mol^{-1}$. This result reveals that R41 is the most feasible channel in the fragmentation of $HOCH_2O$ radical.*

*The resulting $HOCH(CH_3)O$ radical has three types of pathways. The first one is β-site $C_1$-$C_2$ bond scission leading to the formation of HCOOH + $CH_3$ (R45) with the barrier of 8.3 kcal $mol^{-1}$. The second one is β-site $C_1$-$O_1$ bond cleavage resulting in formation of $CH_3COH$ + OH (R46) with the barrier of 26.7 kcal $mol^{-1}$. The third one is H-abstraction by $O_2$ leading to $CH_3COOH$ + $HO_2$ ·(R47) with the barrier of 26.2 kcal $mol^{-1}$. Based on the calculated reaction barriers, it can be found that β-site $C_1$-$C_2$ bond scission is the dominant pathway in the fragmentation of $HOCH(CH_3)O$ radical.*

*The formed $HO(CH_3)_2CO$ radical can either dissociate to $CH_3COOH$ + $CH_3$ ·(R51) via the $C_1$-$C_3$ bond scission with the barrier of 8.2 kcal $mol^{-1}$, or decompose into $CH_3COCH_3$ + OH (R52) through the $C_1$-$O_1$ bond breaking with the barrier of 24.3 kcal $mol^{-1}$. The result again shows that the β-site C-C bond scission is the dominate pathway.*

*The rate coefficients of the dominant pathways of $HOCH_2O$, $HOCH(CH_3)O$ and $HO(CH_3)_2CHO$ radicals fragmentation are summarized in Table S12. As can be seen in Table S12, $k_{R41}$ is slightly increased with the temperature increasing, and the discrepancy is about a factor of 12 at the two extremes of temperature. At ground level with $[O_2]$ = ~ $5.0 \times 10^{18}$ molecule $cm^{-3}$, the pseudo-first-order rate constant $k'_{O2}$ = $k_{R41}[O_2]$ is estimated to be 38.0 $s^{-1}$ at room temperature. $k_{R45}$ vary significantly from $2.0 \times 10^6$ (273 K) to $3.1 \times 10^8$ (400 K) $s^{-1}$, and they exhibit a marked positive temperature dependence. Similar phenomenon is also observed from $k_{R51}$ that $k_{R51}$ is significantly increased with increasing temperature. $k_{R51}$ is a factor of ~ 1.3 greater than $k_{R45}$ in the temperature range studied, implying that the rate coefficient of β-site C-C bond scission is slightly increased as the number of methyl group is increased.*

6. The prefix 'anti' should be italicized throughout the manuscript.

    **Response:** Based on the Reviewer's suggestion, the prefix 'anti' is italicized throughout the manuscript.

7. The italics/non-italics energies in Fig. S13 and S14 in the supplement are not always in the same vertical order.

    **Response:** Based on the Reviewer's suggestion, the non-italics energies have been placed in the upper number in Figures S13 and S14.

[Figure]

**Figure S13.** PES ($\Delta G_a^{\#}$ and $\Delta E_a^{\#}$, in italics) for the autoxidation of HOCH$_3$CHOO radical predicted at the M06-2X/ma-TZVP//M06-2X/6-311+G(2df,2p) level of theory

[Figure]

**Figure S14.** PES ($\Delta G_a^{\#}$ and $\Delta E_a^{\#}$, in italics) for the autoxidation of HO(CH$_3$)$_2$COO radical predicted at the M06-2X/ma-TZVP//M06-2X/6-311+G(2df,2p) level of theory

8. There are some grammatical and logical errors in this manuscript. I suggest revising the grammatical errors accordingly.

**Response:** Based on the Reviewer's suggestion, the sentences/phrases, missing words, and the grammatically confusing sentences have been corrected carefully in the revised manuscript.

References

1. Zhou, X., Liu, Y., Dong, W., and Yang, X.: Unimolecular reaction rate measurement of *syn*-CH$_3$CHOO, J. Phys. Chem. Lett., 10, 4817-4821, https://doi.org/10.1021/acs.jpclett.9b01740, 2019.

2. Stone, D., Whalley, L. K., and Heard, D. E.: Tropospheric OH and HO$_2$ radicals: field measurements and model comparisons, Chem. Soc. Rev., 41, 6348-6404, https://doi.org/10.1039/c2cs35140d, 2012.

3. Hofzumahaus, A., Rohrer, F., Lu, K., Bohn, B., Brauers, T., Chang, C. C., Fuchs, H., Holland, F., Kita, K., Kondo, Y., Li, X., Lou, S., Shao, M., Zeng, L., Wahner, A., and Zhang, Y.: Amplified trace gas removal in the troposphere, Science, 324, 1702-1704, https://doi.org/10.1126/science.1164566, 2009.

4. Kumar, M., and Francisco, J. S.: Red-light-induced decomposition of an organic peroxy radical: a new source of the HO$_2$ radical, Angew. Chem. Int. Ed., 54, 15711-15714, https://doi.org/10.1002/anie.201509311, 2015.

5. Bianchi, F., Kurten, T., Riva, M., Mohr, C., Rissanen, M. P., Roldin, P., Berndt, T., Crounse, J. D., Wennberg, P. O., Mentel, T. F., Wildt, J., Junninen, H., Jokinen, T., Kulmala, M., Worsnop, D. R., Thornton, J. A., Donahue, N., Kjaergaard, H. G., and Ehn, M.: Highly oxygenated organic molecules (HOM) from gas-phase autoxidation involving peroxy radicals: a key contributor to atmospheric aerosol, Chem. Rev., 119, 3472-3509, https://doi.org/10.1021/acs.chemrev.8b00395, 2019.

---

## Author Comment (AC2)

Prof. Yu Huang
State Key Lab of Loess and Quaternary Geology
Institute of Earth Environment, Chinese Academy
of Sciences, Xi'an, 710061, China
Tel./Fax: (86) 29-62336261
E-mail: huangyu@ieecas.cn

Jan. 27, 2022

Dear reviewer,

**Revision for Manuscript acp-2021-890**

We thank you very much for giving us the opportunity to revise our manuscript. We highly appreciate the reviewer for their comments and suggestions on the manuscript entitled "**OH-initiated atmospheric degradation of hydroxyalkyl hydroperoxides: mechanism, kinetics, and structure-activity relationship**". We have made revisions of our manuscript carefully according to the comments and suggestions of reviewer. The revised contents are marked in blue color. The response letter to reviewer is attached at the end of this cover letter.

We hope that the revised manuscript can meet the requirement of Atmospheric Chemistry & Physics. Any further modifications or revisions, please do not hesitate to contact us.

Look forward to hearing from you as soon as possible.

Best regards,

Yu Huang

**Comments of reviewer #2**

1. Authors discuss the transformation mechanism of HHPs. But there is no information on the concentration of HHPs (in forested regions?).

**Response:** Based on the Reviewer's suggestion, the concentrations of hydroxyalkyl hydroperoxides (HHPs) have been added in the revised manuscript. Hydroxymethyl hydroperoxide (HMHP, $HOCH_2OOH$), the simplest HHPs from the ozonolysis of ethene in the presence of water, is observed in significant abundance in the atmosphere (Allen et al., 2018). The measured concentration of HMHP is varied considerably depending on the location, season and altitude, and its concentration is measured to be up to 5 ppbv in forested regions (Allen et al., 2018; Francisco and Eisfeld, 2009). Recently, the concentration of HMHP was measured during the summer 2013 in the southeastern United States, and found that the average mixing ratio of HMHP is 0.25 ppbv with a maximum of 4.0 ppbv in the boundary layer (Allen et al., 2018).

Corresponding descriptions have been added in the page 4 line 98-106 of the revised manuscript:

*Hydroxymethyl hydroperoxide (HMHP, $HOCH_2OOH$), the simplest HHPs from the ozonolysis of ethene in the presence of water, is observed in significant abundance in the atmosphere (Allen et al., 2018). The measured concentration of HMHP is varied considerably depending on the location, season and altitude, and its concentration is measured to be up to 5 ppbv in forested regions (Allen et al., 2018; Francisco and Eisfeld, 2009). Recently, the concentration of HMHP was measured during the summer 2013 in the southeastern United States, and found that the average mixing ratio of HMHP is 0.25 ppbv with a maximum of 4.0 ppbv in the boundary layer(Allen et al., 2018).*

2. The lifetimes of distinct HHPs with respect to OH should be estimated under atmospheric conditions.

**Response:** Based on the Reviewer's suggestion, the lifetime of distinct HHPs reactivity toward OH radical has been added in the revised manuscript. The atmospheric lifetime is expressed as eqn (1): (Long et al., 2017)

$$\tau = \frac{1}{k[X]} \tag{1}$$

where $k$ is the rate coefficient of distinct HHPs reactions with OH radical. [X] is the concentration of OH radical, which varies from 5 to $15 \times 10^6$ molecules cm$^{-3}$ during daylight (Tan et al., 2017). At ambient temperature, the total rate coefficients of OH radical reactions with HOCH$_2$OOH, HOCH(CH$_3$)OOH and HOC(CH$_3$)$_2$OOH are $3.3 \times 10^{-11}$, $3.0 \times 10^{-11}$ and $1.6 \times 10^{-11}$ cm$^3$ molecule$^{-1}$ s$^{-1}$, respectively. The atmospheric lifetime of HOCH$_2$OOH, HOCH(CH$_3$)OOH and HOC(CH$_3$)$_2$OOH reactivity toward OH radical are estimated to be 0.58-1.74 h, 0.60-1.79 h and 1.23-3.69 h at room temperature.

Corresponding descriptions have been added in the page 13 line 348-352 of the revised manuscript:

*The concentrations of OH radical vary from 5 to $15 \times 10^6$ molecules cm$^{-3}$ during daylight (Long et al., 2017), resulting in the atmospheric lifetime of HOCH$_2$OOH, HOCH(CH$_3$)OOH and HOC(CH$_3$)$_2$OOH reactivity toward OH radical are estimated to be 0.58-1.74 h, 0.60-1.79 h and 1.23-3.69 h at room temperature.*

3. $\Delta E_a^{\#}$, $\Delta G_a^{\#}$ and $\Delta G$ are employed in the manuscript, the author should explain the meaning of each item in detail.

**Response:** Based on the Reviewer's suggestion, the explanations on the meaning of $\Delta E_a^{\#}$, $\Delta G_a^{\#}$ and $\Delta G$ have been added in the revised manuscript. The Gibbs free energy ($G$) for each species is obtained by combining the single-point energy with the Gibbs correction ($G = G_{corr} + E$). The electronic energy ($\Delta E_a^{\#}$) and free energy ($\Delta G_a^{\#}$) barriers are defined as the difference in energy between transition state and pre-reactive complex ($\Delta E_a^{\#} = E_{TS} - E_{RC}$ and $\Delta G_a^{\#} = G_{TS} - G_{RC}$). The reaction free energy ($\Delta G$) is referred to the difference in energy between product and reactant ($\Delta G = G_P - G_R$).

Corresponding descriptions have been added in the page 7 line 178-183 of the revised manuscript:

*Herein, the Gibbs free energy (G) for each species is obtained by combining the single-point energy with the Gibbs correction ($G = G_{corr} + E$). The electronic energy ($\Delta E_a^{\#}$) and free energy ($\Delta G_a^{\#}$) barriers are defined as the difference in energy between transition state and pre-reactive complex ($\Delta E_a^{\#} = E_{TS} - E_{RC}$ and $\Delta G_a^{\#} = G_{TS} - G_{RC}$). The reaction free energy ($\Delta G$) is referred to the difference in energy between product and reactant ($\Delta G = G_P - G_R$).*

4. Author should compare the barriers of the gas phase decomposition of HOCH$_2$OO radical with the barrier of self-reaction of HOCH$_2$OO radical. Kumar and Francisco reported the unimolecular decay of HOCH$_2$OO radical could be a new source of HO2 radical (Angew. Chem. Int. Ed. 2015, 54, 15711-15714; J. Phys. Chem. A 2016, 120, 2677-2683).

**Response:** Based on the Reviewer's suggestion, the comparation on the barriers of the gas phase decomposition of HOCH$_2$OO radical and its self-reaction has been added in the revised manuscript. As can be seen in Figure 5, the self-reaction of HOCH$_2$OO radical proceeds via oxygen-to-oxygen coupling leading to the formation of tetroxide intermediate S14 with the electronic energy and free energy barriers of 7.3 and 19.6 kcal·mol$^{-1}$. Kumar and Francisco reported that the electronic energy barrier of the gas phase decomposition of HOCH$_2$OO radical is 14.0 kcal·mol$^{-1}$ and it could be a new source of HO$_2$ radical in the troposphere (Kumar and Francisco, 2015, 2016). Compared with the electronic energy barriers of unimolecular decomposition of HOCH$_2$OO radical and its self-reaction, it can be found that the self-reaction of HOCH$_2$OO radical resulting in formation of tetroxide intermediate S14 is significantly feasible. The related reference has been cited in the revised manuscript.

Corresponding descriptions have been added in the page 16 line 412-420 of the revised manuscript:

*The self-reaction of HOCH$_2$OO radical proceeds via oxygen-to-oxygen coupling leading to the formation of tetroxide intermediate S14 with the electronic energy and free energy barriers of 7.3 and 19.6 kcal·mol$^{-1}$. Kumar and Francisco reported that the electronic energy barrier of the gas phase decomposition of HOCH$_2$OO radical is 14.0 kcal·mol$^{-1}$ and it could be a new source of HO$_2$ radical in the troposphere (Kumar and Francisco, 2015, 2016). Compared with the electronic energy barriers of unimolecular dissociation of HOCH$_2$OO radical and its self-reaction, it can be found that the self-reaction of HOCH$_2$OO radical resulting in formation of S14 is significantly feasible.*

5. Author should provide the pseudo first order rates for the reactions of distinct RO$_2$ radicals with HO$_2$ and NO under the urban, rural and forest environments.

**Response:** Based on the Reviewer's suggestion, the pseudo-first-order rate constants of

distinct $RO_2$ radicals reactions with $HO_2$ radical and NO have been added in the revised manuscript. The typical atmospheric concentrations of $HO_2$ radical are 5, 20 and 50 pptv in the urban, rural and forest environments (Bianchi et al., 2019). At ambient temperature, the rate coefficient $k_{R31}$ of $HOCH_2OO \cdot + HO_2 \cdot$ reaction (R31) is estimated to be $1.7 \times 10^{-11}$ cm$^3$ molecule$^{-1}$ s$^{-1}$, translating into the pseudo-first-order rate constants $k'_{HO2} = k_{HO2}[HO_2]$ of $1.1 \times 10^{-3}$, $4.2 \times 10^{-3}$ and $1.1 \times 10^{-2}$ s$^{-1}$, respectively, in the urban, rural and forest environments. The pseudo-first-order rate constants of $HOCH(CH_3)OO \cdot + HO_2 \cdot$ (R32) and $HOC(CH_3)_2OO \cdot + HO_2 \cdot$ (R33) reactions are predicted to be $3.0 \times 10^{-3}$ and $4.8 \times 10^{-3}$ (urban), $1.1 \times 10^{-2}$ and $1.8 \times 10^{-2}$ (rural), $3.0 \times 10^{-2}$ and $4.8 \times 10^{-2}$ s$^{-1}$ (forest) at room temperature.

The typical atmospheric concentrations of NO are about 10 ppbv, 1 ppbv and 20 pptv in the urban, rural and forest environments (Bianchi et al., 2019). The rate coefficient $k_{R39}$ of $HOCH_2OO$ radical reaction with NO is estimated to be $4.3 \times 10^{-12}$ cm$^3$ molecule$^{-1}$ s$^{-1}$ at ambient temperature, resulting in the pseudo-first-order rate constants $k'_{NO} = k_{NO}[NO]$ of $6.5 \times 10^{-1}$, $6.5 \times 10^{-2}$, and $1.3 \times 10^{-3}$, respectively, in the urban, rural and forest environments. For the bimolecular reaction of $HOCH(CH_3)OO$ radical with NO, the predicted pseudo-first-order rate constants are $6.7 \times 10^{-1}$, $6.7 \times 10^{-2}$, and $1.3 \times 10^{-3}$, respectively, in the urban, rural and forest environments. The pseudo-first-order rate constants of $HOC(CH_3)_2OO$ radical reaction with NO are $7.3 \times 10^{-1}$, $7.3 \times 10^{-2}$, and $1.5 \times 10^{-3}$, respectively, in the urban, rural and forest environments.

Corresponding descriptions have been added in the page 22 line 525-533 and page 27 line 653-658 of the revised manuscript:

*At ambient temperature, $k_{R31}$ is estimated to be $1.7 \times 10^{-11}$ cm$^3$ molecule$^{-1}$ s$^{-1}$, which is in good agreement with the value of $\sim 2 \times 10^{-11}$ cm$^3$ molecule$^{-1}$ s$^{-1}$ for the reaction of acyl peroxy radicals with $HO_2$ radical (Wennberg et al., 2018). The typical atmospheric concentrations of $HO_2$ radical are about 5, 20 and 50 pptv in the urban, rural and forest environments (Bianchi et al., 2019), translating into the pseudo-first-order rate constants $k'_{HO2} = k_{HO2}[HO_2]$ of $1.1 \times 10^{-3}$, $4.2 \times 10^{-3}$ and $1.1 \times 10^{-2}$ s$^{-1}$, respectively. The pseudo-first-order rate constants of R32 and R33 are predicted to be $3.0 \times 10^{-3}$ and $4.8 \times 10^{-3}$ (urban), $1.1 \times 10^{-2}$ and $1.8 \times 10^{-2}$ (rural), $3.0 \times 10^{-2}$ and $4.8 \times 10^{-2}$ s$^{-1}$ (forest) at room temperature.*

*The typical atmospheric concentrations of NO are about 10 ppbv, 1 ppbv and 20 pptv in the urban, rural and forest environments (Bianchi et al., 2019). The rate coefficient of*

*HOCH₂OO·* reaction with NO is calculated to be $4.3 \times 10^{-12}$ cm³ molecule⁻¹ s⁻¹ at room temperature, resulting in the pseudo-first-order rate constants $k'_{NO} = k_{NO}[NO]$ of $6.5 \times 10^{-1}$, $6.5 \times 10^{-2}$, and $1.3 \times 10^{-3}$, respectively, in the urban, rural and forest environments. For the bimolecular reaction of HOCH(CH₃)OO radical with NO, the predicted pseudo-first-order rate constants are $6.7 \times 10^{-1}$, $6.7 \times 10^{-2}$, and $1.3 \times 10^{-3}$, respectively, in the urban, rural and forest environments. The pseudo-first-order rate constants of HOC(CH₃)₂OO radical reaction with NO are $7.3 \times 10^{-1}$, $7.3 \times 10^{-2}$, and $1.5 \times 10^{-3}$, respectively, in the urban, rural and forest environments.

6. In Fig. 2, the text (mentioned structural parameters) overlaps with the structures.

**Response:** Based on the Reviewer's suggestion, the Figure 2 has been redrawn in the revised manuscript. For clarity, the 2D drawings of some important species are labeled in Figure 2. The optimized geometries of all the stationary points involved in the initial reactions of OH radical with HOCH₂OOH are displayed Figure S6.

[Figure]

**Figure 2.** PES ($\Delta G_a^{\#}$) for the OH-initiated reactions of HOCH₂OOH from the CH₂OO + H₂O reaction predicted at the M06-2X/ma-TZVP//M06-2X/6-311+G(2df,2p) level of theory (a and b represent the pre-reactive and post-reactive complexes)

[Figure]

**Figure S6.** Geometries of all the stationary points for the initial reaction of HOCH₂OOH with OH radical optimized at the M06-2X/6-311+G(2df,2p) level of theory

Reference

Allen, H. M., Crounse, J. D., Bates, K. H., Teng, A. P., Krawiec-Thayer, M. P., Rivera-Rios, J. C., Keutsch, F. N., Clair, J. M. S., Hanisco, T. F., Møller, K. H., Kjaergaard, H. G., and Wennberg, P. O.: Kinetics and product yields of the OH initiated oxidation of hydroxymethyl hydroperoxide, J. Phys. Chem. A, 122, 6292-6302, https://doi.org/10.1021/acs.jpca.8b04577, 2018.

Bianchi, F., Kurten, T., Riva, M., Mohr, C., Rissanen, M. P., Roldin, P., Berndt, T., Crounse, J. D., Wennberg, P. O., Mentel, T. F., Wildt, J., Junninen, H., Jokinen, T., Kulmala, M., Worsnop, D. R., Thornton, J. A., Donahue, N., Kjaergaard, H. G., and Ehn, M.: Highly oxygenated organic molecules (HOM) from gas-phase autoxidation involving peroxy radicals: a key contributor to atmospheric aerosol, Chem. Rev., 119, 3472-3509, https://doi.org/10.1021/acs.chemrev.8b00395, 2019.

Francisco, J. S., and Eisfeld, W.: Atmospheric oxidation mechanism of hydroxymethyl hydroperoxide, J. Phys. Chem. A, 113, 7593-7600, https://doi.org/10.1021/jp901735z, 2009.

Kumar, M., and Francisco, J. S.: Red-light initiated decomposition of $\alpha$-hydroxy methylperoxy radical in the presence of organic and inorganic acids: implications for the $HO_x$ formation in the lower stratosphere, J. Phys. Chem. A, 120, 2677-2683, https://doi.org/10.1021/acs.jpca.6b01515, 2016.

Kumar, M., and Francisco, J. S.: Red-light-induced decomposition of an organic peroxy radical: a new source of the $HO_2$ radical, Angew. Chem. Int. Ed., 54, 15711-15714, https://doi.org/10.1002/anie.201509311, 2015.

Long, B., Bao, J. L., and Truhlar, D. G.: Reaction of $SO_2$ with OH in the atmosphere, Phys. Chem. Chem. Phys., 19, 8091-8100, https://doi.org/10.1039/C7CP00497D, 2017.

Tan, Z., Fuchs, H., Lu, K., Hofzumahaus, A., Bohn, B., Broch, S., Dong, H., Gomm, S., Häseler, R., He, L., Holland, F., Li, X., Liu, Y., Lu, S., Rohrer, F., Shao, M., Wang, B., Wang, M., Wu, Y., Zeng, L., Zhang, Y., Wahner, A., and Zhang, Y.: Radical chemistry at a rural site (Wangdu) in the North China Plain: observation and model calculations of OH, HO2 and RO2 radicals, Atmos. Chem. Phys., 17, 663–690, https://doi.org/10.5194/acp-17-663-2017, 2017.

---

## Author Comment (AC3)

Prof. Yu Huang
State Key Lab of Loess and Quaternary Geology
Institute of Earth Environment, Chinese Academy
of Sciences, Xi'an, 710061, China
Tel./Fax: (86) 29-62336261
E-mail: huangyu@ieecas.cn

Jan. 27, 2022

Dear reviewer,

**Revision for Manuscript acp-2021-890**

We thank you very much for giving us the opportunity to revise our manuscript. We highly appreciate the reviewer for their comments and suggestions on the manuscript entitled "**OH-initiated atmospheric degradation of hydroxyalkyl hydroperoxides: mechanism, kinetics, and structure-activity relationship**". We have made revisions of our manuscript carefully according to the comments and suggestions of reviewer. The revised contents are marked in blue color. The response letter to reviewer is attached at the end of this cover letter.

We hope that the revised manuscript can meet the requirement of Atmospheric Chemistry & Physics. Any further modifications or revisions, please do not hesitate to contact us.

Look forward to hearing from you as soon as possible.

Best regards,

Yu Huang

**Comments of reviewer #3**

1. I see two major issues with the work. First, it is generally known that the internal H-shift isomerizations become important only at larger carbon structures than studied here. The rather extensive previous literature amply points out that the H-shifts (discussed in older literature often as H transfer) are not competitive from the same, or adjacent, C-bearing functional groups. Thus it is rather surprising this was even considered here, and I think the whole discussion about $RO_2$ autoxidation does not make any sense for these small systems. Additionally, the resulting accretion products ROOR are important for aerosol growth only at larger sizes. Thus, these small systems are not expected to play any practical role in atmospheric particulate matter formation (unless the second $RO_2$ forming the ROOR is a very big and polar molecule). Moreover, one H-shift reaction does not really constitute autoxidation sequence, but is rather just a single isomerization/ rearrangement reaction. Thus the whole word "autoxidation" should not be associated with the current work. All that said, for the sake of completeness, the current calculations could/should be left in, but it has to be made crystal clear that autoxidation is not expected here, and these are only common isomerization (by H-shift) reactions.

   **Response:** Extensive previous literatures have demonstrated the autoxidation of peroxy radical $RO_2$ plays an important role in the oxidation of volatile organic compounds (VOCs) with high molecular weight in the atmosphere (Bianchi et al., 2019; Ehn et al., 2017). The autoxidation of $RO_2$ radical includes sequential intramolecular H-shifts and $O_2$ additions, in which the first H-shift is strongly rate-limiting reaction (Nozière and Vereecken, 2019; Crounse et al., 2013). In the present study, we mainly focus on the mechanism of isomerization reactions of $HOCH_2OO$, $HOCH(CH_3)OO$, and $HO(CH_3)_2COO$ radicals. The corresponding potential energy surfaces (PES) are shown in Figures 9, S13 and S14, respectively. As can be seen in Figure 9, the lowest-energy conformer $HOCH_2OO$-a can proceed via a 1,3-H shift from the -$CH_2$ group to the terminal oxygen leading to S28-a (HO CHOOH) with the barrier of 41.6 kcal $mol^{-1}$. $HOCH_2OO$-b can isomerize to S28-b1 and S28-b2 via the four-membered ring transition states TS34-b1 and TS34-b2 (1,3-H shifts) with the barriers of 41.6 and 45.0 kcal $mol^{-1}$. But these three 1,3-H shift reactions have comparatively high barriers, making them irrelevant in the atmosphere. Equivalent to the case of $HOCH_2OO$ radical, the isomerization of $HOCH(CH_3)OO$ radical proceeds via the 1,3- and 1,4-H shifts from the -CH or -$CH_3$ groups to the terminal oxygen resulting in formation of

hydroperoxyalkyl radicals (Figure S13). These 1,3- and 1,4-H shift reactions accompany with the extremely high barriers (> 37.9 kcal mol$^{-1}$), implying that they are of less importance in the atmosphere. Similar conclusion is also derived from the isomerization of HO(CH$_3$)$_2$COO radical that 1,4-H shift reactions are unfavourable kinetically (Figure S14). The high barriers of 1,3- and 1,4-H shifts can be interpreted as the result of the large ring strain energy (RSE) in the cyclic transition state geometries. As a consequence, the isomerization reactions of HOCH$_2$OO, HOCH(CH$_3$)OO and HO(CH$_3$)$_2$COO radicals are not likely to proceed in the atmosphere.

The accretion products ROOR are of less importance in the self-reactions of small RO$_2$ radicals (e.g. CH$_3$OO, C$_2$H$_5$OO, CH$_3$C(O)OO radicals), while they are characterized as an effective source of secondary organic aerosol (SOA) in the self-reactions of large RO$_2$ radicals (Berndt et al., 2018; Zhang et al., 2012; Liang et al., 2011). Although the ROOR formed from the self-reactions of small RO$_2$ radicals is unimportant, the present mechanism investigation is meaningful to understand the self-reactions of complex RO$_2$ radicals. In the present study, the schematic PESs for the self-reactions of HOCH$_2$OO, HOCH(CH$_3$)OO and HO(CH$_3$)$_2$COO radicals are shown in Figures 5-7, respectively. As can be seen in Figure 5a, the self-reaction of HOCH$_2$OO radical starts with the formations of tetroxide complexes IM13-a and IM14-a in the entrance channel, with 2.9 and 3.4 kcal mol$^{-1}$ stability. Then they fragment into dimer S13 + $^1$O$_2$ (R13) and HOCH$_2$OOH + HOCHOO (R14) via transition states TS13 and TS14 with the barriers of 43.3 and 51.5 kcal mol$^{-1}$. But the barriers of R13 and R14 are extremely high, making them irrelevant in the atmosphere.

[revised manuscript text omitted]

2. Another major issue connects to Figure 2: Are you sure you get the RC's and PC's right in the mechanism shown? Seems strange that such analogous reactions with so similar reaction partners (i.e., substituting -H with -CH$_3$ in adjacent sp3 C-atom is not expect have a profound influence) would have so different pre-reaction complexes. How was this specifically verified. I mean, is it possible you were doing a too constrained original conformer search/optimization and missed certain RC's? Did you try what the energetics would be if the RC spatial structure would be close to identical in every system? Especially RC3 really stands out, but others differ too. The whole issue might be better visualized, if you would show these with the actual reagent structures, and not just with grey symbols. After all there is only 3 systems, so this would not increase the space demands much.

**Response:** Based on the Reviewer's suggestion, the pre-reactive complexes (RCs) and post-reactive complexes (PCs) considered in the OH-initiated oxidation of the three smallest hydroxyalkyl hydroperoxides (HHPs) have been rechecked in the revised manuscript. Due to the representation way of presented RCs in Figure 2 of the original manuscript, the structure of RCs is misleading. In the revised manuscript, the original Figure 2 is divided into three figures. The free-energy and electronic-energy PESs for the initiation reactions of OH radical with HOCH$_2$OOH, HOCH(CH$_3$)OOH and HOC(CH$_3$)$_2$OOH are presented in Figures 2-4 and S1-S3 of the revised manuscript. And the optimized geometries of all the stationary points, including reactants, RCs, transition states, PCs, and products, are displayed in Figures S6-S8.

In the present study, the structures of distinct HHPs are displayed in Figure 1. The equilibrium geometries of all the stationary points on the PESs are fully optimized at the M06-2X/6-311+G(2df,2p) level of theory, rather than the constrained optimization. For each pre-reactive complex, a conformer search is employed to search the stable conformers of HHPs with OH radical. The structures obtained from the conformer search are initially optimized at the

B3LYP/6-31G(d) level of theory, since the B3LYP functional has been shown to yield reliable relative energies between conformers (Møller et al., 2016). Then, all unique conformers with electronic energies within 5.0 kcal·mol$^{-1}$ with respect to the lowest-energy conformer are further optimized at the M06-2X/6-311+G(2df,2p) level of theory. Based on the obtained conformers, the transition states of various possible H-abstraction reactions between OH radical and distinct HHPs are located at the M06-2X/6-311+G(2df,2p) level of theory. The intrinsic reaction coordinate (IRC) calculations are performed in both directions at the M06-2X/6-311+G(2df,2p) level of theory, and then the forward and reverse IRC endpoints are optimized at the same level. For improved energies, the single-point calculations are performed at the M06-2X/ma-TZVP level of theory

A schematic PES for the reaction of OH radical with HOCH$_2$OOH is shown in Figure 2. As can be seen in Figure 2, the reaction for HOCH$_2$OOH with OH radical proceeds via four distinct pathways: H-abstraction from the -O$_1$H$_1$ (R1), -C$_1$H$_3$ (R2), -C$_1$H$_4$ (R3) and -O$_2$O$_3$H$_2$ groups (R4). For each pathway, a pre-reactive complex with a six- or seven-membered ring structure is formed in the entrance channel, which is stabilized by hydrogen bond interactions between the oxygen atom of OH radical and the abstraction hydrogen atom of HOCH$_2$OOH, and the remnant hydrogen atom of OH radical and one of oxygen atoms of HOCH$_2$OOH (Figures S6). Then, it surmounts modest barrier that is higher in energy than the reactants to reaction. The reaction barrier $\Delta G_a^{\#}$ are reduced in the order of 6.4 (R1) > 5.8 (R2) ≈ 5.4 (R3) > 1.5 (R4) kcal·mol$^{-1}$, indicating that H-abstraction from the -O$_2$O$_3$H$_2$ group (R4) is more preferable than those from the -O$_1$H$_1$, -C$_1$H$_3$ and -C$_1$H$_4$ groups (R1-R3). Same conclusion is also derived from the energy barriers $\Delta E_a^{\#}$ that R4 is the most favorable H-abstraction pathway (Figure S1). The difference of barrier heights can be attributed to the bond dissociation energy (BDE) of different types of bonds in HOCH$_2$OOH molecule. The BDE are decreased in the order of 103.7 (O$_1$-H$_1$) > 98.2 (C$_1$-H$_3$) ≈ 97.4 (C$_1$-H$_4$) > 87.2 (O$_3$-H$_2$) kcal·mol$^{-1}$, which are in good agreement with the order of barrier heights of H-abstraction reactions. As indicated by their reaction free energy values, it can be found that the exothermicity of R4 is the largest among these four H-abstraction reactions. Based on the above discussions, it is concluded that H-abstraction from the -O$_2$O$_3$H$_2$ group resulting in formation of HOCH$_2$OO radical (R4) is feasible on both thermodynamically and kinetically. If we assume that the spatial structure of RC is closed to identical in each H-abstraction reaction, it can be found that H-abstraction from -CH$_2$ group is competitive with that from the -OH group on HOCH$_2$OOH. The

result is contrary to the above-mentioned conclusion, implying that our hypothesis is incorrect.

[Figure]

**Figure 1.** The structures of distinct HHPs

[Figure]

**Figure 2.** PES ($\Delta G_a^\#$) for the OH-initiated reactions of HOCH$_2$OOH from the CH$_2$OO + H$_2$O reaction predicted at the M06-2X/ma-TZVP//M06-2X/6-311+G(2df,2p) level of theory (a and b represent the pre-reactive and post-reactive complexes)

[Figure]

**Figure S1.** PES ($\Delta E_a^{\#}$) for the OH-initiated reactions of HOCH$_2$OOH from the CH$_2$OO + H$_2$O reaction predicted at the M06-2X/ma-TZVP//M06-2X/6-311+G(2df,2p) level of theory (a and b represent the pre-reactive and post-reactive complexes)

[Figure]

**Figure S6.** Geometries of all the stationary points for the initial reaction of HOCH$_2$OOH with OH radical optimized at the M06-2X/6-311+G(2df,2p) level of theory

Corresponding descriptions have been added in the page 10 line 253-273 of the revised manuscript:

*As can be seen in Figure 2, the reaction for HOCH$_2$OOH with OH radical proceeds via four distinct pathways: H-abstraction from the -O$_1$H$_1$ (R1), -C$_1$H$_3$ (R2), -C$_1$H$_4$ (R3) and -O$_2$O$_3$H$_2$ groups (R4). For each pathway, a pre-reactive complex with a six- or seven-membered ring structure is formed in the entrance channel, which is stabilized by hydrogen bond interactions between the oxygen atom of OH radical and the abstraction hydrogen atom of HOCH$_2$OOH, and the remnant hydrogen atom of OH radical and one of oxygen atoms of HOCH$_2$OOH (Figure S6).*

*Then, it surmounts modest barrier that is higher in energy than the reactants to reaction. The reaction barrier $\Delta G_a^{\#}$ are reduced in the order of 6.4 (R1) > 5.8 (R2) ≈ 5.4 (R3) > 1.5 (R4) kcal mol$^{-1}$, indicating that H-abstraction from the $-O_2O_3H_2$ group (R4) is more preferable than those from the $-O_1H_1$, $-C_1H_3$ and $-C_1H_4$ groups (R1-R3). Same conclusion is also derived from the energy barriers $\Delta E_a^{\#}$ that R4 is the most favorable H-abstraction pathway (Figure S1). The difference of barrier heights can be attributed to the bond dissociation energy (BDE) of different types of bonds in HOCH$_2$OOH molecule. The BDE are decreased in the order of 103.7 ($O_1$-$H_1$) > 98.2 ($C_1$-$H_3$) ≈ 97.4 ($C_1$-$H_4$) > 87.2 ($O_3$-$H_2$) kcal mol$^{-1}$, which are in good agreement with the order of barrier heights of H-abstraction reactions. As indicated by their reaction free energy values, it can be found that the exothermicity of R4 is the largest among these four H-abstraction reactions. Based on the above discussions, it is concluded that H-abstraction from the $-O_2O_3H_2$ group resulting in formation of HOCH$_2$OO radical (R4) is feasible on both thermodynamically and kinetically.*

3. Furthermore, it feels a bit strange that in the methylated radicals, the C-H abstraction does not play a bigger role, as seen in some older work on OH + alcohols and OH + amines. From the same previous work it seems somewhat strange that abstraction from -OH is the next likely pathway, and still no C-H abstraction. Moreover, the conclusion that one methyl group substitution does not really matter, but two groups do, seem evenly strange as the methyl groups seem to be rather in the by-stander position, and are likely to influence little on what is occurring at the C-O-OH functionality. What type of sensitivity tests were made to ensure you have found the correct pathways? The IRC computation only ensures you are connecting the right reactants with the correct products, but it does not tell if you have found the most likely pathway or not.

**Response:** Based on the Reviewer's suggestion, all the H-abstraction pathways included in the initiation reactions of OH radical with HOCH$_2$OOH, HOCH(CH$_3$)OOH and HOC(CH$_3$)$_2$OOH have been recalculated in the revised manuscript. Owing to our carelessness, the conclusion that two methyl groups substitutions have a significant influence on the barrier of H-abstraction from the -OOH group is incorrect in the original manuscript. In the revised manuscript, a conformer search is employed to search the stable conformers of HHPs with OH radical. Based on the structures resulting from the conformer search, the transition states of various possible

H-abstraction reactions between OH radical and distinct HHPs are located at the M06-2X/6-311+G(2df,2p) level of theory. The intrinsic reaction coordinate (IRC) calculations are performed in both directions at the M06-2X/6-311+G(2df,2p) level of theory, and then the forward and reverse IRC endpoints are optimized at the same level. The single-point calculations are performed at the M06-2X/ma-TZVP level of theory based on the M06-2X/6-311+G(2df,2p) optimized geometries. By comparing the stability of pre-reactive complex and the barrier height of transition state, the most likely manner of each H-abstraction pathway is found. The corresponding free-energy and electronic-energy PESs for the OH-initiated oxidation of $HOCH_2OOH$, $HOCH(CH_3)OOH$ and $HOC(CH_3)_2OOH$ are presented in Figures 2-4 and S1-S3 of the revised manuscript. And the geometrical structures of all the stationary points are displayed in Figures S6-S8.

[revised manuscript text omitted]

Corresponding descriptions have been added in the page 10 line 274-288, page 11 line 309-320 and page 13 line 353-365 of the revised manuscript:

*Considering the different reaction sites of hydrogen atoms, the atmospheric transformation of $HOCH(CH_3)OOH$ from the anti-$CH_3CHOO$ + $H_2O$ reaction should have six types of*

*H-abstraction pathways as presented in Figure 3. As shown in Figure 3, each H-abstraction reaction begins with the formation of a weakly bound hydrogen bonded pre-reactive complex with a six- or seven-membered ring structure in the entrance channel (Figure S7). Then it immediately transforms into the respective product via the corresponding transition state. The $\Delta G_a^{\#}$ of H-abstraction from the $-C_1H_3$ (R6) and $-O_2O_3H_2$ (R8) groups are 2.2 and 1.7 $kcal\,mol^{-1}$, respectively, which are ~ 4-5 $kcal\,mol^{-1}$ lower than those from the $-O_1H_1$ (R5) and $-CH_3$ groups (R7). This result shows that R6 and R8 have nearly identical importance in the atmosphere. Compared with the barriers of H-abstraction at the $C_\alpha$ (R6) and $C_\beta$ (R7) positions, it can be found that the former case is more favourable than the latter case. This conclusion is further supported by Jara-Toro's study for the reactions of OH radical with linear saturated alcohols (methanol, ethanol and n-propanol) that H-abstraction at the $C_\alpha$ position is predominant (Jara-Toro, R. A et al., 2017, 2018).*

*From Figure 4, it can be seen that H-abstraction from $HOC(CH_3)_2OOH$ includes eight possible H-abstraction pathways. All the H-abstraction reactions are strongly exothermic and spontaneous, signifying that they are thermodynamically feasible under atmospheric conditions. It deserves mentioning that the release of energy of R12 is significantly greater than those of R9-R11. For each H-abstraction pathway, a RC with a six- or seven-membered ring structure is formed prior to the corresponding TS, which is more stable than the separate reactants due to the hydrogen bond interactions between $HOC(CH_3)_2OOH$ and OH radical. Then, the RC overcomes modest barrier to reaction. The $\Delta G_a^{\#}$ of H-abstraction from the $-O_2O_3H_2$ group (R12) is 2.7 $kcal\,mol^{-1}$, which is the lowest among these eight H-abstraction reactions. This result again shows that the H-abstraction from the $-O_2O_3H_2$ group is the dominant pathway.*

*In summary, the dominant pathway is the H-abstraction from the -OOH group in the initiation reactions of OH radical with $HOCH_2OOH$. H-abstraction from -CH group is competitive with that from the -OOH group in the reaction of OH radical with $HOCH(CH_3)OOH$. Compared the barriers of H-abstraction from the -OOH and $-CH_2$ groups in the $OH + HOCH_2OOH$ system with that for the analogous reactions in the $OH + HOCH(CH_3)OOH$ system. It can be found that the barrier of H-abstraction from the -CH group is reduced by 3.6 $kcal\,mol^{-1}$, whereas the barrier of H-abstraction from the -OOH group is increased by 0.2 $kcal\,mol^{-1}$ when a methyl group substitution occurs at the C1-position of $HOCH_2OOH$. The dominant pathway is the*

*H-abstraction from the -OOH group in the reaction of OH radical with HOC(CH₃)₂OOH, and the barrier height is increased by 1.2 kcal mol⁻¹ compared to the OH + HOCH₂OOH system. The barrier of H-abstraction from the -OOH group is slightly increased as the number of methyl group is increased.*

4. Figures: Symbol fonts should be increased in all figures showing potential energy surfaces. Currently they are in many places unreadable. Moreover, the molecular figures are too small to follow the mechanism from the figures with this symbolism. To make matters worse, it is difficult to see what peroxy radicals are reacting to make the tetroxides in the figures, and there is no help from the captions. There must be a better way to make these readable. The easiest way is to split the figures in parts (i.e., Figure 3 becomes, for example, Figures 3a to 3c) and at the same time considerably increase the amount of text in the captions.

**Response:** Based on the Reviewer's suggestion, the symbol fonts and molecular structures in all figures have been redrawn in the revised manuscript. For clarity, the 2D drawings of some important species are labeled in the PESs of the initiation reactions of OH radical with HOCH₂OOH, HOCH(CH₃)OOH and HOC(CH₃)₂OOH (Figures 2-4 and S1-S3). And the optimized geometries of all the stationary points are displayed in Figures S6-S8. The PESs of the self-reactions of HOCH₂OO and HOCH(CH₃)OO radicals are divided into two parts (Figures 5a and 5b, Figures 6a and 6b). And the captions of all figures are reworded in the revised manuscript.

5. Figure 2 caption: Far more details of this figure should be included. Currently I am having very hard time understanding it based on the manuscript text. Why, for example, the RC is uphill in energy although you go from separated reactants into a pre-reactive (=favorable binding) complex? Currently the caption does not help.

**Response:** Based on the Reviewer's suggestion, the original Figure 2 has been divided into three figures in the revised manuscript. The free energy and electronic energy PESs for the initiation reactions of OH radical with HOCH₂OOH, HOCH(CH₃)OOH and HOC(CH₃)₂OOH are presented in Figures 2-4 and S1-S3, respectively. The energy of pre-reactive complex is higher than that of the separate reactants when the free energies are used to construct the PES. The energy of pre-reactive complex is lower than that of the separate reactants when the electronic energies

are employed to build the PES. The change in the position of pre-reactive complex relative to the initial reactants is due to the contribution of entropy effect in the free energy.

[Figure]

**Figure 2.** PES ($\Delta G_a^{\#}$) for the OH-initiated reactions of HOCH$_2$OOH from the CH$_2$OO + H$_2$O reaction predicted at the M06-2X/ma-TZVP//M06-2X/6-311+G(2df,2p) level of theory (a and b represent the pre-reactive and post-reactive complexes)

[Figure]

**Figure 3.** PES ($\Delta G_a^{\#}$) for the OH-initiated reactions of HOCH(CH$_3$)OOH from the *anti*-CH$_3$CHOO + H$_2$O reaction predicted at the M06-2X/ma-TZVP//M06-2X/6-311+G(2df,2p) level of theory (a and b represent the pre-reactive and post-reactive complexes)

[Figure]

**Figure 4.** PES ($\Delta G_a^{\#}$) for the OH-initiated reactions of HOC(CH$_3$)$_2$OOH from the (CH$_3$)$_2$COO + H$_2$O reaction predicted at the M06-2X/ma-TZVP//M06-2X/6-311+G(2df,2p) level of theory (a and b represent the pre-reactive and post-reactive complexes)

[Figure]

**Figure S1.** PES ($\Delta E_a^{\#}$) for the OH-initiated reactions of HOCH$_2$OOH from the CH$_2$OO + H$_2$O reaction predicted at the M06-2X/ma-TZVP//M06-2X/6-311+G(2df,2p) level of theory (a and b represent the pre-reactive and post-reactive complexes)

[Figure]

**Figure S2.** PES ($\Delta E_a^{\#}$) for the OH-initiated reactions of HOCH(CH$_3$)OOH from the *anti*-CH$_3$CHOO + H$_2$O reaction predicted at the M06-2X/ma-TZVP//M06-2X/6-311+G(2df,2p) level of theory (a and b represent the pre-reactive and post-reactive complexes)

[Figure]

**Figure S3.** PES ($\Delta E_a^{\#}$) for the OH-initiated reactions of HOC(CH$_3$)$_2$OOH from the (CH$_3$)$_2$COO + H$_2$O reaction predicted at the M06-2X/ma-TZVP//M06-2X/6-311+G(2df,2p) level of theory (a and b represent the pre-reactive and post-reactive complexes)

6 Although the message comes "mostly clear" throughout the text, the text should be language edited. Again, I do understand it quite well being a non-native speaker, but I assume native speakers will not agree with me.

**Response:** Based on the Reviewer's suggestion, the revised manuscript has been corrected carefully, and the sentences and grammar have been proofread detailedly by some native English speakers.

7 Hydroxyalkyl hydroperoxides (HHPs) are formed in several other reactions too, especially in OH addition initiation with subsequent $HO_2$ termination. This could be mentioned in the intro too.

**Response:** Based on the Reviewer's suggestion, the HHPs formed from the initiation OH-addition with subsequent $HO_2$-termination reactions have been added in the Introduction of the revised manuscript. Hydroxyalkyl hydroperoxides (HHPs), formed in the reactions of Criegee intermediates (CIs) with water vapour and in the initiation OH-addition with subsequent $HO_2$-termination reactions, play important roles in the formation of secondary organic aerosol (SOA).

Corresponding descriptions have been revised in the page 3 line 58-61 of the revised manuscript:

*Hydroxyalkyl hydroperoxides (HHPs), formed in the reactions of Criegee intermediates (CIs) with water vapour and in the initiation OH-addition with subsequent $HO_2$-termination reactions, play important roles in the formation of secondary organic aerosol (SOA) (Qiu et al., 2019; Kumar et al., 2014).*

8 Line 69: What is the difference between vapor pressure and volatility?

**Response:** In general, at a given temperature, the higher the saturated vapor pressure of compound, the stronger the volatility. At different temperatures, the saturated vapor pressure of the same compound is different. HHPs, due to the presence of both hydroxyl and perhydroxy moieties, have relatively low volatility contributing substantially to the formation of SOA.

Corresponding descriptions have been revised in the page 3 line 76-78 of the revised manuscript:

*HHPs, due to the presence of both hydroxyl and perhydroxy moieties, have relatively low*

*volatility contributing substantially to the formation of SOA (Qiu et al., 2019).*

9 In the Abstract, please reword the following sentence: "In urban environments, the rate-limiting step is the hydrogen abstraction by $O_2$ in the processes of $HOCH_2OO$ radical reaction with NO, while it becomes the O-O bond scission when one or two methyl substitutions occur at the C1-position of $HOCH_2OO$ radical." I think I know what this means, but I can't be sure.

**Response:** Based on the Reviewer's suggestion, the mentioned sentence has been reworded as "In urban environments, reaction with $O_2$ forming formic acid and $HO_2$ radical is the dominant removal pathway for $HOCH_2O$ radical formed from the reaction of $HOCH_2OO$ radical with NO. The $\beta$-site C-C bond scission is the dominate pathway in the dissociation of $HOCH(CH_3)O$ and $HOC(CH_3)_2O$ radicals formed from the $HOCH(CH_3)OO \cdot + NO$ and $HOC(CH_3)_2OO \cdot + NO$ reactions."

Corresponding descriptions have been revised in the page 2 line 48-536 of the revised manuscript:

*In urban environments, reaction with $O_2$ forming formic acid and $HO_2$ radical is the dominant removal pathway for $HOCH_2O$ radical formed from the reaction of $HOCH_2OO$ radical with NO. The $\beta$-site C-C bond scission is the dominate pathway in the dissociation of $HOCH(CH_3)O$ and $HOC(CH_3)_2O$ radicals formed from the $HOCH(CH_3)OO \cdot + NO$ and $HOC(CH_3)_2OO \cdot + NO$ reactions.*

10 The sentence: "Previous literatures have been confirmed that the energies obtained from unrestricted DFT are comparable to the multi-reference CASSCF method (Lee et al., 2016; Bach et al., 2005)." seems to indicate that "unrestricted DFT" gives similar results to "CASSCF". Is this really the case, and why it is so? Does not the choice of active space factor in?

**Response:** The tetroxide intermediate formed from the self-reaction of $RO_2$ radical proceeds through the asymmetric two step O-O bond scission to produce a caged tetroxide intermediate of overall singlet multiplicity comprising two same-spin alkoxyl radicals (spin down) and triplet oxygen (spin up). This type of reaction mechanism can be described by the broken symmetry unrestricted DFT (UDFT) and multi-reference CASSCF methods (Lee et al., 2016; Bach et al., 2005). Previous studies have demonstrated that the UDFT method is suitable to identify the

metastable singlet caged radical complex minimum and is successfully located the transition states of O-O bond homolysis, and the energies are comparable to the more accurate and expensive CASSCF method (Lee et al., 2016; Bach et al., 2005). In the present study, the UDFT method is selected to study the asymmetric O-O bond scission and represents a compromise between the computational accuracy and efficiency. The broken symmetry UM06-2X method is applied to generate the initial guesses of the tetroxide intermediate and transition state geometries with mixed HOMO and LUMO ($S^2 \approx 1$) by using the guess = mix keyword. The single-point energies are refined at the UM06-2X/ma-TZVP level of theory.

Corresponding descriptions have been revised in the page 6 line 155-170 of the revised manuscript:

*The tetroxide intermediate formed from the self-reaction of $RO_2$ radical proceeds through the asymmetric two step O-O bond scission to produce a caged tetroxide intermediate of overall singlet multiplicity comprising two same-spin alkoxyl radicals (spin down) and triplet oxygen (spin up). This type of reaction mechanism can be described by the broken symmetry unrestricted DFT (UDFT) and multi-reference CASSCF methods (Lee et al., 2016; Bach et al., 2005). Previous studies have demonstrated that the UDFT method is suitable to identify the metastable singlet caged radical complex minimum and is successfully located the transition states of O-O bond homolysis, and the energies are comparable to the more accurate and expensive CASSCF method (Lee et al., 2016; Bach et al., 2005). In the present study, the UDFT method is selected to study the asymmetric O-O bond scission and represents a compromise between the computational accuracy and efficiency. The broken symmetry UM06-2X/6-311+G(2df,2p) method is applied to generate the initial guesses of the tetroxide intermediate and transition state geometries with mixed HOMO and LUMO ($S^2 \approx 1$) by using the guess = mix keyword. The single-point energies are refined at the UM06-2X/ma-TZVP level of theory.*

11 Please embed figures into text. It helps no one if they are positioned after the text.

**Response:** Based on the Reviewer's suggestion, all figures have been embedded into the revised manuscript.

12 Why is there four reactions in Figure 2, although there are 3 title reactions handled in the paper?

In fact, the fourth option is hinted in the text "Considering the different chemical environments of hydrogen atoms, the atmospheric transformation of HHPs initiated by OH radical should have four types of H-abstraction pathways as presented in Figure 2." but I did not observe an explanation what is meant by it. In any case it would be good to break the Figure 2 into several separate figures - one for each reaction.

**Response:** Based on the Reviewer's suggestion, the Figure 2 has been divided into three separate figures in the revised manuscript. The free-energy PESs for the initiation reactions of OH radical with $HOCH_2OOH$, $HOCH(CH_3)OOH$ and $HOC(CH_3)_2OOH$ are presented in Figures 2-4, respectively. As can be seen in Figure 2, the reaction for $HOCH_2OOH$ with OH radical proceeds via four distinct pathways: H-abstraction from the $-O_1H_1$ (R1), $-C_1H_3$ (R2), $-C_1H_4$ (R3) and $-O_2O_3H_2$ groups (R4). For each pathway, a pre-reactive complex with a six- or seven-membered ring structure is formed in the entrance channel, which is stabilized by hydrogen bond interactions between the oxygen atom of OH radical and the abstraction hydrogen atom of $HOCH_2OOH$, and the remnant hydrogen atom of OH radical and one of oxygen atoms of $HOCH_2OOH$. Then, it surmounts modest barrier that is higher in energy than the reactants to reaction. The reaction barrier $\Delta G_a^{\#}$ are reduced in the order of 6.4 (R1) > 5.8 (R2) $\approx$ 5.4 (R3) > 1.5 (R4) kcal mol$^{-1}$, indicating that H-abstraction from the $-O_2O_3H_2$ group (R4) is more preferable than those from the $-O_1H_1$, $-C_1H_3$ and $-C_1H_4$ groups (R1-R3). The difference of barrier heights can be attributed to the bond dissociation energy (BDE) of different types of bonds in $HOCH_2OOH$ molecule. The BDE are decreased in the order of 103.7 ($O_1$-$H_1$) > 98.2 ($C_1$-$H_3$) $\approx$ 97.4 ($C_1$-$H_4$) > 87.2 ($O_3$-$H_2$) kcal mol$^{-1}$, which are in good agreement with the order of barrier heights of distinct H-abstraction reactions. As indicated by their reaction free energy values, it can be found that the exothermicity of R4 is the largest among these four H-abstraction reactions. Based on the above discussions, it is concluded that H-abstraction from the $-O_2O_3H_2$ group resulting in formation of $HOCH_2OO$ radical (R4) is feasible on both thermodynamically and kinetically.

[revised manuscript text omitted]

13 The following statement:" pseudo-first-order rate constant $k'_{HO2}$ of $\sim 10^{-2}$ s$^{-1}$ in the forest environments" is completely condition dependent and cannot be represented by a single value. A range of values would be equally ambiguous, yet still better.

**Response:** Based on the Reviewer's suggestion, the range of pseudo-first-order rate constant $k'_{HO2}$ is given in the revised manuscript. The rate coefficients of distinct $RO_2$ radicals reactions with $HO_2$ radical exhibit a weakly negative temperature dependence, translating into the pseudo-first-order rate constant $k'_{HO2}$ of 1-5 $\times 10^{-2}$ s$^{-1}$ in the forest environments.

Corresponding descriptions have been revised in the page 30 line 710-712 of the revised manuscript:

*The calculated rate coefficients exhibit a weakly negative temperature dependence, translating into the pseudo-first-order rate constant $k'_{HO2}$ of 1-5 $\times$ 10$^{-2}$ s$^{-1}$ in the forest environments.*

14 I am not sure if I can follow what is meant by this: "(e) The rate-limiting step is the hydrogen abstraction by $O_2$ in the processes of $HOCH_2OO$ radical reaction with NO, while it becomes the C-C bond scission when one or two methyl substitutions occur at the C1-position of $HOCH_2OO$

radical." Please clarify and reword the statement.

**Response:** Based on the Reviewer's suggestion, the statement on the mentioned sentence has been reworded in the revised manuscript. The $HOCH_2O$ radical formed from the reaction of $HOCH_2OO$ radical with NO has two kinds of pathways (Figure 10): (1) it directly decomposes into $CH_2O$ and OH radical (R40) via $\beta$-site $C_1$-$O_1$ bond scission with the barrier of 52.4 kcal mol$^{-1}$; (2) it converts into HCOOH and $HO_2$ radical (R41) through H-abstraction by $O_2$ with the barrier of 26.4 kcal mol$^{-1}$. This result reveals that R41 is the most feasible channel in the fragmentation of $HOCH_2O$ radical. The resulting $HOCH(CH_3)O$ radical from the $HOCH(CH_3)OO \cdot +$ NO reaction has three types of pathways (Figure 11). The first one is $\beta$-site $C_1$-$C_2$ bond scission leading to the formation of HCOOH + $CH_3$ (R45) with the barrier of 8.3 kcal mol$^{-1}$. The second one is $\beta$-site $C_1$-$O_1$ bond cleavage resulting in formation of $CH_3COH +$ OH (R46) with the barrier of 26.7 kcal mol$^{-1}$. The third one is H-abstraction by $O_2$ leading to $CH_3COOH + HO_2 \cdot$ (R47) with the barrier of 26.2 kcal mol$^{-1}$. Based on the calculated reaction barriers, it can be found that $\beta$-site $C_1$-$C_2$ bond scission is the dominant pathway in the fragmentation of $HOCH(CH_3)O$ radical. The $HO(CH_3)_2CO$ radical formed from the $HOC(CH_3)_2OO \cdot +$ NO reaction can either dissociate to $CH_3COOH + CH_3 \cdot$ (R51) via the $C_1$-$C_3$ bond scission with the barrier of 8.2 kcal mol$^{-1}$, or decompose into $CH_3COCH_3 +$ OH (R52) through the $C_1$-$O_1$ bond breaking with the barrier of 24.3 kcal mol$^{-1}$ (Figure 12). The result again shows that the $\beta$-site C-C bond scission is the dominate pathway.

In summary, reaction with $O_2$ forming formic acid and $HO_2$ radical is the dominant removal pathway for $HOCH_2O$ radical formed from the reaction of $HOCH_2OO$ radical with NO. The $\beta$-site C-C bond scission is the dominate pathway in the dissociation of $HOCH(CH_3)O$ and $HOC(CH_3)_2O$ radicals formed from the $HOCH(CH_3)OO \cdot +$ NO and $HOC(CH_3)_2OO \cdot +$ NO reactions.

[Figure]

**Figure 10.** PES ($\Delta G_a^{\#}$ and $\Delta E_a^{\#}$, in italics) for the reaction of $HOCH_2OO$ radical with NO predicted at the M06-2X/ma-TZVP//M06-2X/6-311+G(2df,2p) level of theory (the superscript a is calculated at the MP2/ma-TZVP//MP2/6-311+G(2df,2p) level)

[Figure]

**Figure 11.** PES ($\Delta G_a^{\#}$ and $\Delta E_a^{\#}$, in italics) for the reaction of $HOCH(CH_3)OO$ radical with NO predicted at the M06-2X/ma-TZVP//M06-2X/6-311+G(2df,2p) level of theory (the superscript a is calculated at the MP2/ma-TZVP//MP2/6-311+G(2df,2p) level)

[Figure]

**Figure 12.** PES ($\Delta G_a^{\#}$ and $\Delta E_a^{\#}$, in italics) for the reaction of HO(CH$_3$)$_2$COO radical with NO predicted at the M06-2X/ma-TZVP//M06-2X/6-311+G(2df,2p) level of theory (the superscript a is calculated at the MP2/ma-TZVP//MP2/6-311+G(2df,2p) level)

Corresponding descriptions have been revised in the page 26 line 624-629, page 27 line 633-640, page 27 line 648-652 and page 30 line 713-717 of the revised manuscript:

*The formed HOCH$_2$O radical has two kinds of pathways: (1) it directly decomposes into CH$_2$O and OH radical (R40) via β-site C$_1$-O$_1$ bond scission with the barrier of 52.4 kcal mol$^{-1}$; (2) it converts into HCOOH and HO$_2$ radical (R41) through H-abstraction by O$_2$ with the barrier of 26.4 kcal mol$^{-1}$. This result reveals that R41 is the most feasible channel in the fragmentation of HOCH$_2$O radical.*

*The resulting HOCH(CH$_3$)O radical has three types of pathways. The first one is β-site C$_1$-C$_2$ bond scission leading to the formation of HCOOH + CH$_3$ (R45) with the barrier of 8.3 kcal mol$^{-1}$. The second one is β-site C$_1$-O$_1$ bond cleavage resulting in formation of CH$_3$COH + OH (R46) with the barrier of 26.7 kcal mol$^{-1}$. The third one is H-abstraction by O$_2$ leading to CH$_3$COOH + HO$_2$ ·(R47) with the barrier of 26.2 kcal mol$^{-1}$. Based on the calculated reaction barriers, it can be found that β-site C$_1$-C$_2$ bond scission is the dominant pathway in the fragmentation of HOCH(CH$_3$)O radical.*

*The formed HO(CH$_3$)$_2$CO radical can either dissociate to CH$_3$COOH + CH$_3$ ·(R51) via the C$_1$-C$_3$ bond scission with the barrier of 8.2 kcal mol$^{-1}$, or decompose into CH$_3$COCH$_3$ + OH (R52) through the C$_1$-O$_1$ bond breaking with the barrier of 24.3 kcal mol$^{-1}$. The result again shows that the β-site C-C bond scission is the dominate pathway.*

*Reaction with $O_2$ forming formic acid and $HO_2$ radical is the dominant removal pathway for HOCH$_2$O radical formed from the reaction of HOCH$_2$OO radical with NO. The $\beta$-site C-C bond scission is the dominate pathway in the dissociation of HOCH(CH$_3$)O and HOC(CH$_3$)$_2$O radicals formed from the HOCH(CH$_3$)OO · + NO and HOC(CH$_3$)$_2$OO · + NO reactions.*

15 It is unclear what is meant by the following statement "One reason for the barrier difference could lie in the fact that the bond dissociation energies (BDE) of different types of bonds are significantly different in the HOCH$_2$OOH molecule."

**Response:** The free-energy and electronic-energy PESs for the initiation reactions of OH radical with HOCH$_2$OOH are displayed in Figure 2 and S1, respectively. As can be seen in Figure 2, the reaction for HOCH$_2$OOH with OH radical proceeds via four distinct pathways: H-abstraction from the -O$_1$H$_1$ (R1), -C$_1$H$_3$ (R2), -C$_1$H$_4$ (R3) and -O$_2$O$_3$H$_2$ groups (R4). The reaction barrier $\Delta G_a^{\#}$ are reduced in the order of 6.4 (R1) > 5.8 (R2) $\approx$ 5.4 (R3) > 1.5 (R4) kcal mol$^{-1}$, indicating that H-abstraction from the -O$_2$O$_3$H$_2$ group (R4) is more preferable than those from the -O$_1$H$_1$, -C$_1$H$_3$ and -C$_1$H$_4$ groups (R1-R3). Same conclusion is also derived from the energy barriers $\Delta E_a^{\#}$ that R4 is the most favorable H-abstraction pathway (Figure S1). The difference of barrier heights can be attributed to the bond dissociation energy (BDE) of different types of bonds in HOCH$_2$OOH molecule. The BDE are decreased in the order of 103.7 (O$_1$-H$_1$) > 98.2 (C$_1$-H$_3$) $\approx$ 97.4 (C$_1$-H$_4$) > 87.2 (O$_3$-H$_2$) kcal mol$^{-1}$, which are in good agreement with the order of barrier heights of H-abstraction reactions.

Corresponding descriptions have been revised in the page 10 line 253-269 of the revised manuscript:

*As can be seen in Figure 2, the reaction for HOCH$_2$OOH with OH radical proceeds via four distinct pathways: H-abstraction from the -O$_1$H$_1$ (R1), -C$_1$H$_3$ (R2), -C$_1$H$_4$ (R3) and -O$_2$O$_3$H$_2$ groups (R4). The reaction barrier $\Delta G_a^{\#}$ are reduced in the order of 6.4 (R1) > 5.8 (R2) $\approx$ 5.4 (R3) > 1.5 (R4) kcal mol$^{-1}$, indicating that H-abstraction from the -O$_2$O$_3$H$_2$ group (R4) is more preferable than those from the -O$_1$H$_1$, -C$_1$H$_3$ and -C$_1$H$_4$ groups (R1-R3). Same conclusion is also derived from the energy barriers $\Delta E_a^{\#}$ that R4 is the most favorable H-abstraction pathway (Figure S1). The difference of barrier heights can be attributed to the bond dissociation energy (BDE) of different types of bonds in HOCH$_2$OOH molecule. The BDE are decreased in the order of 103.7*

*($O_1$-$H_1$) > 98.2 ($C_1$-$H_3$) ≈ 97.4 ($C_1$-$H_4$) > 87.2 ($O_3$-$H_2$) kcal mol$^{-1}$, which are in good agreement with the order of barrier heights of H-abstraction reactions.*

16 I would like to see the rates obtained (k vs T) also in Figures in relation to each other, and not just as Tables. I think this could be very useful to the reader, as the tabular format is more difficult to compare.

**Response:** Based on the Reviewer's suggestion, the rate coefficients versus temperature have been plotted in figures of the revised manuscript.

17 I find it a bit odd to state that a single channel of hydroxymethylperoxy radical oxidation giving $HO_2$ radical is "a new source of $HO_2$ radical in the troposphere". I mean, can this specific radical have even a minute influence on the tropospheric $HO_2$ burden?

**Response:** Kumar and Francisco investigated the gas phase decomposition of α-hydroxymethylperoxy radical $HOCH_2OO$ by using quantum chemical method (Kumar et al., 2015). It was found that the $HOCH_2OO$ radical decomposition represents a new source of $HO_2$ radical in the troposphere. This finding may help in understanding the discrepancy between the modeled and measured concentrations of $HO_2$ radical in the troposphere. However, to the best of our knowledge, the contribution of the $HOCH_2OO$ radical decomposition to the tropospheric $HO_2$ radical burden is still unknown. In the future work, we will adopt the combination of quantum chemistry and numerical simulation to estimate the contribution of the $HOCH_2OO$ radical decomposition to the tropospheric $HO_2$ radical burden.

18 In the beginning of chapter 3.2., you are missing the second RO produced in the reaction.

**Response:** Based on the Reviewer's suggestion, the second RO radical formed from the self-reaction of $RO_2$ radicals has been added in the revised manuscript. The self-reactions of $RO_2$ radicals can either produce RO ·+ R'O ·+ $O_2$ (propagation channel), or generate ROH + R'(-H, =O) + $O_2$ or produce ROOR + $O_2$ (termination channel) that has been recognized as an important SOA precursor.

Corresponding descriptions have been revised in the page 15 line 387-390 of the revised manuscript:

*The self-reactions of RO₂ radicals can either produce RO · + R'O · + O₂ (propagation channel), or generate ROH + R'(-H, =O) + O₂ or produce ROOR + O₂ (termination channel) that has been recognized as an important SOA precursor (Berndt et al., 2018; Zhang et al., 2012)*

19 Chapter 3.2.1: Mark all radicals the same way (i.e., with similar dot).

**Response:** Based on the Reviewer's suggestion, the dot is applied to mark all radicals in the revised manuscript.

20 Chapter 3.2.1: It is rather disappointing to hear that "It is worth noting that the termination products are not found in the $HO(CH_3)_2COO$ radical reaction system owing to the absence of alpha hydrogen atom." when it was just in previous sentence stated that:" is not discussed in detail to avoid redundancy." Please explain further these currently missing channels.

**Response:** Based on the Reviewer's suggestion, the missing pathways for the self-reaction of $HO(CH_3)_2COO$ radical have been added in the revised manuscript. Figure 7 depicts a schematic PES for the self-reaction of $HOC(CH_3)_2OO$ radical. As shown in Figure 7, the dominant pathway for the self-reaction of $HO(CH_3)_2COO$ radical begins with the formation of tetroxide intermediate S24 via an oxygen-to-oxygen coupling transition state TS28 with the barrier of 20.4 kcal mol$^{-1}$; then it transforms into the caged tetroxide intermediate S26 of overall singlet spin multiplicity through the asymmetric two-step O-O bond cleavage with the barriers of 22.0 and 3.4 kcal mol$^{-1}$; finally, S26 can either produce two $HO(CH_3)_2CO$ radicals with the exoergicity of 10.3 kcal mol$^{-1}$, or generate dimer S27 with the exothermicity of 31.5 kcal mol$^{-1}$. Different the self-reactions of $HOCH_2OO$ and $HOCH(CH_3)OO$ radicals, the termination product of the self-reaction of $HOC(CH_3)_2OO$ radical is exclusively dimer S27. The reason is due to the absence of alpha hydrogen atom in $HOC(CH_3)_2OO$ radical.

[Figure]

**Figure 7.** PES ($\Delta G_a^{\#}$ and $\Delta E_a^{\#}$, in italics) for the self-reaction of HO(CH$_3$)$_2$COO radicals predicted at the M06-2X/ma-TZVP//M06-2X/6-311+G(2df,2p) level of theory

Corresponding descriptions have been revised in the page 18 line 465-475 of the revised manuscript:

*As shown in Figure 7, the dominant pathway for the self-reaction of HO(CH$_3$)$_2$COO radical begins with the formation of tetroxide intermediate S24 via an oxygen-to-oxygen coupling transition state TS28 with the barrier of 20.4 kcal mol$^{-1}$; then it transforms into the caged tetroxide intermediate S26 of overall singlet spin multiplicity through the asymmetric two-step O-O bond cleavage with the barriers of 22.0 and 3.4 kcal mol$^{-1}$; finally, S26 can either produce two HO(CH$_3$)$_2$CO radicals with the exoergicity of 10.3 kcal mol$^{-1}$, or generate dimer S27 with the exothermicity of 31.5 kcal mol$^{-1}$. Different the self-reactions of HOCH$_2$OO and HOCH(CH$_3$)OO radicals, the termination product of the self-reaction of HOC(CH$_3$)$_2$OO radical is exclusively dimer S27. The reason is due to the absence of alpha hydrogen atom in HOC(CH$_3$)$_2$OO radical.*

21 It is stated that "The main primary sources of HO$_2$ radical in the atmosphere are from the photolysis of CH$_2$O and OVOCs, and the ozonolysis reactions". I guess the authors meant "photo-oxidation" rather than "photolysis" here.

**Response:** Based on the Reviewer's suggestion, the word "photolysis" has been replaced by

"photo-oxidation" in the revised manuscript. The main sources of $HO_2$ radical involve the photo-oxidation of oxygenated volatile organic compounds (OVOCs) and the ozonolysis reaction.

Corresponding descriptions have been revised in the page 21 line 496-498 of the revised manuscript:

*The main sources of $HO_2$ radical involve the photo-oxidation of oxygenated volatile organic compounds (OVOCs) and the ozonolysis reaction.*

22 I wonder if you could find a better reference for [$HO_2$] (and other atmospheric concentrations) than Bianchi et al. 2019. To me it seems that those numbers are somewhat questionable, or perhaps better to say that it feels odd that you can give such a "common value" for a whole type-of-an-environment. Is single value really realistic?

**Response:** Based on the Reviewer's suggestion, the concentrations of $HO_2$ radical have been corrected in the revised manuscript. The atmospheric concentration of $HO_2$ radical is 1.5-10 $\times 10^8$ molecules $cm^{-3}$ at ground level in polluted urban environments (Stone et al., 2012).

Corresponding descriptions have been revised in the page 21 line 502-503 of the revised manuscript:

*The atmospheric concentration of $HO_2$ radical is 1.5-10 $\times 10^8$ molecules $cm^{-3}$ at ground level in polluted urban environments (Stone et al., 2012).*

23 According to "HOM review" by Bianchi et al, almost none of the compounds in the work of Noziere and Vereecken would be labelled HOMs, and thus I would strongly advice to change the referencing of the following sentence: "…one after the other, and the resulting finally HOMs (Nozière and Vereecken, 2019; Vereecken and Nozière, 2020)."

**Response:** Based on the Reviewer's suggestion, the references of the formation mechanism of HOMs have been changed in the revised manuscript. The autoxidation mechanism includes an intramolecular H-shift from the $-CH_3$ or $-CH_2-$ groups to the $-OO\cdot$ site, resulting in formation of a hydroperoxyalkyl radical QOOH, followed by $O_2$ addition to form a new peroxy radical ($HOOQO_2$), one after the other, and the resulting finally HOMs (Berndt et al., 2015; Rissanen et al., 2014).

Corresponding descriptions have been revised in the page 23 line 541-545 of the revised manuscript:

[revised manuscript text omitted]

25 Where is the SAR mentioned in the title?

**Response:** Based on the Reviewer's suggestion, the structure-activity relationship on the

initiation reactions of OH radicals with distinct HHPs has been added in the revised manuscript. The dominant pathway is the H-abstraction from the -OOH group in the initiation reactions of OH radical with $HOCH_2OOH$. H-abstraction from -CH group is competitive with that from the -OOH group in the reaction of OH radical with $HOCH(CH_3)OOH$. Compared the barriers of H-abstraction from the -OOH and $-CH_2$ groups in the OH + $HOCH_2OOH$ system with that for the analogous reactions in the OH + $HOCH(CH_3)OOH$ system. It can be found that the barrier of H-abstraction from the -CH group is reduced by 3.6 kcal $mol^{-1}$, whereas the barrier of H-abstraction from the -OOH group is increased by 0.2 kcal $mol^{-1}$ when a methyl group substitution occurs at the C1-position of $HOCH_2OOH$. The dominant pathway is the H-abstraction from the -OOH group in the reaction of OH radical with $HOC(CH_3)_2OOH$, and the barrier height is increased by 1.2 kcal $mol^{-1}$ compared to the OH + $HOCH_2OOH$ system. The barrier of H-abstraction from the -OOH group is slightly increased as the number of methyl group is increased. It is interesting to compare the rate coefficient of dominant pathway in the OH + $HOCH_2OOH$ system with that for the analogous reactions in the OH + $HOCH(CH_3)OOH$ and OH + $HOC(CH_3)_2OOH$ reactions. It can be found that the rate coefficient is almost identical when a methyl group substitution occurs at the $C_1$-position, whereas the rate coefficient reduces by a factor of 2-5 when two methyl groups introduce into the $C_1$-position.

Corresponding descriptions have been added in the page 13 line 353-371 of the revised manuscript:

[revised manuscript text omitted]

---

## Author Response (AR2)

Prof. Yu Huang State Key Lab of Loess and Quaternary Geology Institute of Earth Environment, Chinese Academy of Sciences, Xi'an, 710061, China Tel./Fax: (86) 29-62336261 E-mail: huangyu@ieecas.cn

Feb. 9, 2022

Dear Prof. Chan,

**Revision for Manuscript acp-2021-890**

We thank you very much for giving us the opportunity to revise our manuscript. We highly appreciate the reviewer for their comments and suggestions on the manuscript entitled "OH-initiated atmospheric degradation of hydroxyalkyl hydroperoxides: mechanism, kinetics, and structure-activity relationship". We have made revisions of our manuscript carefully according to the comments and suggestions of reviewer. The revised contents are marked in blue color. The response letter to reviewer is attached at the end of this cover letter.

We hope that the revised manuscript can meet the requirement of Atmospheric Chemistry & Physics. Any further modifications or revisions, please do not hesitate to contact us.

Look forward to hearing from you as soon as possible.

Best regards,

Yu Huang

**Comments of reviewer #3**

1. Section 3.2.3 and 3.3: Reviewer 3 raised a valid point about isomerization and molecular size. It is common knowledge in the field that intramolecular rearrangement, such as H-shifts and organic nitrate formation, is important for larger molecules only. It is therefore unsurprising that these detailed quantum chemical calculations show the same. This point should be acknowledged by citing the appropriate literature.

**Response:** Based on the Reviewer's suggestion, the related references have been cited in the revised manuscript. The isomerization of HOCH2OO radical proceeds via 1,3-H shifts from the -CH2 group to the terminal oxygen leading to the formation of hydroperoxyalkyl radical S28. But these isomerization reactions have comparatively high barriers (> 41 kcal mol-1), making them of less importance in the atmosphere. Similar conclusion is also obtained from the isomerization of HOCH(CH3)OO and HO(CH3)2COO radicals that 1,3- and 1,4-H shifts are unfavourable kinetically. The high barriers of 1,3- and 1,4-H shifts can be interpreted as the result of the large ring strain energy (RSE) in the cyclic transition state geometries. As a consequence, the isomerization reactions of HOCH2OO, HOCH(CH3)OO and HO(CH3)2COO radicals are unlikely to proceed in the atmosphere. This conclusion is further supported by the previous studies that the intramolecular H-shift isomerizations are important only for RO2 radicals with larger carbon structures (Rissanen et al., 2014; Jokinen et al., 2014; Crounse et al., 2013).

The bimolecular reaction of HOCH2OO radical with NO initially leads to nitrite adduct S31 via the barrierless addition of NO to terminal oxygen atom of HOCH2OO radical. The formed S31 can either isomerize to organic nitrate S32 via a concerted process of  $O_2$ - $O_3$  bond breaking and  $O_2$ - $N_1$  bond forming with the barrier of 47.8 kcal mol-1, or decompose into HOCH2O radical and NO2 via the cleavage of  $O_2$ - $O_3$  bond with the barrier of 11.3 kcal mol-1. The result shows that the latter pathway is more favourable than the former channel. Similar conclusion is also obtained from the reactions of NO with HOCH(CH3)OO and HO(CH3)2COO radicals that the formation of organic nitrate is of minor importance in the atmosphere. This result is further supported by the prior studies that the direct formation of organic nitrate from peroxy nitrites is a minor channel in the reactions of isoprene-derived RO2 radicals with NO (Piletic et al., 2017; Zhang et al., 2002).

Corresponding descriptions have been revised in the page 24 line 567-571 and page 26 line 614-623 of the revised manuscript:

As a consequence, the isomerization reactions of  $HOCH_2OO$ ,  $HOCH(CH_3)OO$  and  $HO(CH_3)_2COO$  radicals are unlikely to proceed in the atmosphere. This conclusion is further supported by the previous studies that the intramolecular H-shift isomerizations are important only for  $RO_2$  radicals with larger carbon structures (Crounse et al., 2013; Jokinen et al., 2014; Rissanen et al., 2014).

The bimolecular reaction of HOCH2OO radical with NO initially leads to nitrite adduct S31 via the barrierless addition of NO to terminal oxygen atom  $O_3$  of HOCH2OO radical. S31-cis can either isomerize to organic nitrate S32 (R38) via a concerted process of  $O_2$ - $O_3$  bond breaking and  $O_2$ - $N_1$  bond forming with the barrier of 47.8 kcal mol-1, or decompose into HOCH2O radical and  $NO_2$  (R39) via the cleavage of  $O_2$ - $O_3$  bond with the barrier of 11.3 kcal mol-1. The result shows that the latter pathway is more favourable than the former channel. Similar conclusion is also obtained from the reactions of NO with HOCH(CH3)OO and HO(CH3)2COO radicals that the formation of organic nitrate is of minor importance in the atmosphere. This result is further supported by the prior studies that the direct formation of organic nitrate from peroxy nitrites is a minor channel in the reactions of isoprene-derived RO2 radicals with NO (Piletic et al., 2017; Zhang et al., 2002).

2. Section 4: Conclusions should include broader implications, not just summary of results.

**Response:** Based on the Reviewer's suggestion, the relevant implications have been added in the Conclusion of the revised manuscript.

(a) The dominant pathway is the H-abstraction from the -OOH group in the initiation reactions of OH radical with HOCH2OOH and HOC(CH3)2OOH. H-abstraction from the -CH group is competitive with that from the -OOH group in the reaction of OH radical with HOCH(CH3)OOH. The barrier of H-abstraction from the -OOH group is slightly increased as the number of methyl group is increased. Compared with the rate coefficient of dominant pathway in the parent system, it is almost identical when a methyl group substitution occurs at the C1-position, whereas it reduces by a factor of 2-5 when two methyl groups are introduced into the C1-position. The atmospheric lifetime of HOCH2OOH, HOCH(CH3)OOH and HOC(CH3)2OOH reactivity toward OH radical are estimated to be 0.58-1.74 h, 0.60-1.79 h and 1.23-3.69 h at room temperature under the typical OH radical concentrations of 5-15  $\times 10^6$  molecules cm-3 during daylight.

(b) The self-reaction of H-abstraction product  $RO_2$  radical initially produces tetroxide intermediate via an oxygen-to-oxygen coupling, then it decomposes into propagation and termination products through the asymmetric two-step O-O bond scission. The rate-limiting step is the first O-O bond cleavage, and the barrier is increased with increasing the number of methyl group. This finding is meaningful to understand the self-reaction of complex  $RO_2$  radicals.

(c) The bimolecular reactions of distinct RO2 radicals with HO2 radical lead to the formation of hydroperoxide ROOH as the main product, and the barrier height is independent on the number of methyl substitution. When compared to the rate coefficient for HOCH2OO + HO2 reaction, the rate coefficients increase by a factor of 2-5 when one or two methyl groups are introduced into the C1-position. Using a HO2 radical concentration of ~50 pptv in the forest environments, the pseudo-first-order rate constants  $k'_{HO2}$  of distinct RO2 radical reactions with HO2 radical vary from 1 to 5 × 10-2 s-1.

(d) The isomerization reactions of HOCH2OO, HOCH(CH3)OO and HO(CH3)2COO radicals are unlikely to proceed in the atmosphere because the intramolecular H-shift steps have dramatically high barriers and strongly endergonic. The result implies that the isomerization of  $RO_2$  radicals with smaller carbon structures is of less importance in the atmosphere.

(e) Reaction with  $O_2$  forming formic acid and  $HO_2$  radical is the dominant removal pathway for HOCH2O radical formed from the reaction of HOCH2OO radical with NO. The  $\beta$ -site C-C bond scission is the dominant pathway in the dissociation of HOCH(CH3)O and HOC(CH3)2O radicals formed from the reactions of NO with HOCH(CH3)OO and HOC(CH3)2OO radicals. The result implies that the methyl-substituted alkoxyl radicals primarily proceed via  $\beta$ -site C-C bond scission to produce aldehyde rather than react with O2.

Corresponding descriptions have been revised in the page 30 line 695-707 and page 31 line 708-732 of the revised manuscript:

(a) The dominant pathway is the H-abstraction from the -OOH group in the initiation reactions of OH radical with  $HOCH_2OOH$  and  $HOC(CH_3)_2OOH$ . H-abstraction from the -CH group is competitive with that from the -OOH group in the reaction of OH radical with  $HOCH(CH_3)OOH$ . The barrier of H-abstraction from the -OOH group is slightly increased as the number of methyl group is increased. Compared with the rate coefficient of dominant pathway in the parent system, it is almost identical when a methyl group substitution occurs at the  $C_1$ -position, whereas it reduces by a factor of 2-5 when two methyl groups are introduced into the  $C_1$ -position. The atmospheric lifetime of HOCH2OOH, HOCH(CH3)OOH and HOC(CH3)2OOH reactivity toward OH radical are estimated to be 0.58-1.74 h, 0.60-1.79 h and 1.23-3.69 h at room temperature under the typical OH radical concentrations of 5-15 × 106 molecules cm-3 during daylight.

(b) The self-reaction of H-abstraction product  $RO_2$  radical initially produces tetroxide intermediate via an oxygen-to-oxygen coupling, then it decomposes into propagation and termination products through the asymmetric two-step O-O bond scission. The rate-limiting step is the first O-O bond cleavage, and the barrier is increased with increasing the number of methyl group. This finding is meaningful to understand the self-reaction of complex  $RO_2$  radicals.

(c) The bimolecular reactions of distinct  $RO_2$  radicals with  $HO_2$  radical lead to the formation of hydroperoxide ROOH as the main product, and the barrier height is independent on the number of methyl substitution. When compared to the rate coefficient for  $HOCH_2OO + HO_2$  reaction, the rate coefficients increase by a factor of 2-5 when one or two methyl groups are introduced into the C1-position. Using a  $HO_2$  radical concentration of ~50 pptv in the forest environments, the pseudo-first-order rate constants  $k'_{HO2}$  of distinct  $RO_2$  radical reactions with  $HO_2$  radical vary from 1 to  $5 \times 10^{-2} \text{ s}^{-1}$ .

(d) The isomerization reactions of  $HOCH_2OO$ ,  $HOCH(CH_3)OO$  and  $HO(CH_3)_2COO$  radicals are unlikely to proceed in the atmosphere because the intramolecular H-shift steps have dramatically high barriers and strongly endergonic. The result implies that the isomerization of  $RO_2$  radicals with smaller carbon structures is of less importance in the atmosphere.

(e) Reaction with  $O_2$  forming formic acid and  $HO_2$  radical is the dominant removal pathway for  $HOCH_2O$  radical formed from the reaction of  $HOCH_2OO$  radical with NO. The  $\beta$ -site C-C bond scission is the dominant pathway in the dissociation of  $HOCH(CH_3)O$  and  $HOC(CH_3)_2O$ radicals formed from the reactions of NO with  $HOCH(CH_3)OO$  and  $HOC(CH_3)_2OO$  radicals. The result implies that the methyl-substituted alkoxyl radicals primarily proceed via  $\beta$ -site C-C bond scission to produce aldehyde rather than react with  $O_2$ .

3. In general, the manuscript can be understood by a reader, but there are still many language

issues. The manuscript should be reviewed more closely again for clarity. I point out the following issues here, but there are many others I have not pointed out.

**Response:** Based on the Reviewer's suggestion, the language issues have been corrected carefully in the revised manuscript.

4. Line 32 "is remain" remove "is".

**Response:** Based on the Reviewer's suggestion, the word "is" has been removed in the revised manuscript.

Corresponding description has been revised in the page 2 line 27-29 of the revised manuscript:

However, the transformation mechanisms for OH-initiated oxidation of HHPs remain incompletely understood.

5. Line 46 "RO2 radicals reactions" should be "RO2 radical reactions".

**Response:** Based on the Reviewer's suggestion, the word "RO2 radicals reactions" has been replaced by "RO2 radical reactions" in the revised manuscript.

Corresponding description has been revised in the page 2 line 41-43 of the revised manuscript:

The barrier height of distinct  $RO_2$  radical reactions with  $HO_2$  radical is independent on the number of methyl substitution.

6. Line 49 "dominate" is a verb, not an adjective.

**Response:** Based on the Reviewer's suggestion, the word "dominate" has been replaced by "dominant" in the revised manuscript.

Corresponding description has been revised in the page 2 line 45-47 of the revised manuscript:

The  $\beta$ -site C-C bond scission is the dominant pathway in the dissociation of HOCH(CH3)O and HOC(CH3)2O radicals formed from the reactions of NO with HOCH(CH3)OO and HOC(CH3)2OO radicals.

7. Line 51: Dot symbol is used to denote radical species here, but not in other places. Should be consistent.

**Response:** Based on the Reviewer's suggestion, the dot symbol has been replaced by the word "radical" in the revised manuscript.

8. Line 99: wouldn't HHP come from ozonolysis of all terminal alkenes, not just ethene.

**Response:** Based on the Reviewer's suggestion, the mentioned sentence has been reworded as "Hydroxymethyl hydroperoxide (HMHP, HOCH2OOH), the simplest HHPs come from the ozonolysis of all terminal alkenes in the presence of water, is observed in significant abundance in the atmosphere."

Corresponding description has been revised in the page 4 line 92-94 of the revised manuscript:

Hydroxymethyl hydroperoxide (HMHP,  $HOCH_2OOH$ ), the simplest HHPs come from the ozonolysis of all terminal alkenes in the presence of water, is observed in significant abundance in the atmosphere (Allen et al., 2018).

9. Line 100: replace "varies" with "is varied".

**Response:** Based on the Reviewer's suggestion, the word "varies" has been replaced by "is varied" in the revised manuscript.

Corresponding description has been revised in the page 4 line 94-97 of the revised manuscript:

The measured concentration of HMHP is varied considerably depending on the location, season and altitude, and its concentration is measured to be up to 5 ppbv in forested regions (Allen et al., 2018; Francisco and Eisfeld, 2009).

10. Line 106: awkward phrasing in "A recent experimental study by Allen et al. (2018) conducted ....

**Response:** Based on the Reviewer's suggestion, the mentioned sentence has been reworded as "Allen et al. conducted the OH-initiated oxidation of HMHP in an environmental chamber and

simulated the impact of HMHP oxidation on the global formic acid concentrations using the chemical transport model GEOS-Chem."

Corresponding description has been revised in the page 4 line 100-103 of the revised manuscript:

Allen et al. (2018) conducted the OH-initiated oxidation of HMHP in an environmental chamber and simulated the impact of HMHP oxidation on the global formic acid concentration using the chemical transport model GEOS-Chem.

11. Line 122: "there is few studies" should be "there are few studies"

**Response:** Based on the Reviewer's suggestion, the sentence "there is few studies" has been replaced by "there are few studies" in the revised manuscript.

Corresponding description has been revised in the page 5 line 116-118 of the revised manuscript:

There are few studies on the subsequent transformations of the resulting H-abstraction products formed from the OH-initiated oxidation of larger HHPs.

**12. Line 124 do not start a sentence with "And"**

**Response:** Based on the Reviewer's suggestion, the word "And" has been removed in the revised manuscript.

Corresponding description has been revised in the page 5 line 118-119 of the revised manuscript:

The effect of the size and number of substituents on the rates and outcomes of SOA precursors (e.g. ROOR, HOMs) is uncertain up to now.

13. Line 162 awkward phrasing "is successfully located"

**Response:** Based on the Reviewer's suggestion, the mentioned sentence has been reworded as "Previous studies have demonstrated that the UDFT method is suitable to identify the minimum of metastable singlet caged radical complex and the transition state of O-O bond homolysis."

Corresponding description has been revised in the page 6 line 154-156 of the revised manuscript:

Previous studies have demonstrated that the UDFT method is suitable to identify the minimum of metastable singlet caged radical complex and the transition state of O-O bond homolysis.

14. Line 196 and thereafter literature is not plural.

**Response:** Based on the Reviewer's suggestion, the mentioned sentence has been revised as "Previous literature has demonstrated that the reaction kinetics of multiconformers involvement are more precisely than that of the single conformer approximation"

Corresponding description has been revised in the page 7 line 190-192 of the revised manuscript:

Previous literature has demonstrated that the reaction kinetics of multiconformers involvement are more precisely than that of the single conformer approximation (Møller, et al., 2016, 2020)

15. Line 199 "multiconformers" should be "multiconformer".

**Response:** Based on the Reviewer's suggestion, the word "multiconformers" has been replaced by "multiconformer" in the revised manuscript.

Corresponding description has been revised in the page 7 line 192-193 of the revised manuscript:

Herein, the multiconformer treatment is performed to investigate the H-shift reactions of  $RO_2$  radicals.

16. Line 236 "a highly dependent" should be "strong dependence".

**Response:** Based on the Reviewer's suggestion, the word "a highly dependent" has been replaced by "strong dependence" in the revised manuscript.

Corresponding description has been revised in the page 9 line 229-231 of the revised manuscript:

Previous literatures have proposed that the lifetime of CI with respect to the reaction with water vapour exhibits strong dependence on the nature of CIs (Anglada and Sol é 2016; Taatjes, et al., 2013; Anglada, et al., 2011).

17. Line 263 Add "the" to "same conclusion...".

**Response:** Based on the Reviewer's suggestion, the word "the" has been added in the revised manuscript.

Corresponding description has been revised in the page 10 line 257-259 of the revised manuscript:

The same conclusion is also derived from the energy barriers  $\Delta E_a^{\#}$  that R4 is the most favorable H-abstraction pathway.

18. Line 263 Add "the" to "same conclusion...".

**Response:** Based on the Reviewer's suggestion, the word "the" has been added in the revised manuscript.

Corresponding description has been revised in the page 11 line 291-292 of the revised manuscript:

The same conclusion is also derived from the energy barriers  $\Delta E_a^{\#}$  that the R6' and R8' are the most favourable H-abstraction pathways.

19. Line 325 Should be "decrease slightly with increasing temperature".

**Response:** Based on the Reviewer's suggestion, the mentioned sentence has been revised as "the total rate coefficients  $k_{tot}$  of HOCH2OOH reaction with OH radical decrease slightly with increasing temperature".

Corresponding description has been revised in the page 12 line 319-320 of the revised manuscript:

The total rate coefficients  $k_{tot}$  of HOCH2OOH reaction with OH radical decrease slightly with increasing temperature.

20. Line 337 and in many other instances: replace "are decreased" with "decrease".

**Response:** Based on the Reviewer's suggestion, the word "are decreased" has been replaced

by "decrease" in the revised manuscript.

Corresponding descriptions have been revised in the page 12 line 331-334 and page 22 line 517-518 of the revised manuscript:

From Table S3, it can be seen that the total rate coefficients  $k'_{tot}$  of HOCH(CH3)OOH reaction with OH radical decrease in the range of  $4.5 \times 10^{-11}$  (273 K) to  $8.1 \times 10^{-12}$  (400 K) cm3 molecule-1 s-1 with increasing temperature, and they exhibit a slightly negative temperature dependence.

Similar conclusion is also obtained from the rate coefficients  $k_{R32}$  and  $k_{R33}$  that they decrease with the temperature increasing.

21. Line 345 replace "The reason is most likely due to..." with "The most likely reason is...".

**Response:** Based on the Reviewer's suggestion, the mentioned sentence has been revised as "The most likely reason is due to the stability of pre-reactive complexes that IM8-a is more stable than IM6-a in energy."

Corresponding description has been revised in the page 13 line 340-341 of the revised manuscript:

The most likely reason is due to the stability of pre-reactive complexes that IM8-a is more stable than IM6-a in energy.

22. Line 356-358 "Compared the barriers..." is an incomplete sentence.

**Response:** Based on the Reviewer's suggestion, the mentioned sentence has been reworded as "Compared with the barriers of H-abstraction from the -OOH and -CH2 groups in the reaction of OH radical with HOCH2OOH, it can be found that the barrier of H-abstraction from the -CH group is reduced by 3.6 kcal mol-1, whereas the barrier of H-abstraction from the -OOH group is increased by 0.2 kcal mol-1 when a methyl group substitution occurs at the C1-position of HOCH2OOH."

Corresponding description has been revised in the page 13 line 351-356 of the revised manuscript:

Compared with the barriers of H-abstraction from the -OOH and -CH2 groups in the reaction of OH radical with HOCH2OOH, it can be found that the barrier of H-abstraction from the -CH group is reduced by 3.6 kcal mol-1, whereas the barrier of H-abstraction from the -OOH group is increased by 0.2 kcal mol-1 when a methyl group substitution occurs at the C1-position of HOCH2OOH. 23. Line 370 replace "introduce" with "are introduced".

**Response:** Based on the Reviewer's suggestion, the word "introduce" has been replaced by "are introduced" in the revised manuscript.

Corresponding description has been revised in the page 14 line 362-365 of the revised manuscript:

It can be found that the rate coefficient is almost identical when a methyl group substitution occurs at the  $C_1$ -position, whereas the rate coefficient reduces by a factor of 2-5 when two methyl groups are introduced into the  $C_1$ -position.

24. Line 386-387 The wording in "have three types of channels" is a little awkward. I suggest "have three possible fates" or "react via three possible channels".

**Response:** Based on the Reviewer's suggestion, the mentioned sentence has been revised as "In principle, the H-abstraction products  $RO_2$  radicals have three possible fates in pristine environments."

Corresponding description has been revised in the page 15 line 380-381 of the revised manuscript:

In principle, the H-abstraction products RO2 radicals have three possible fates in pristine environments.

25. Line 398: would it also require RO2 concentrations to be high.

**Response:** Based on the Reviewer's suggestion, the mentioned sentence has been reworded as "The self-reaction is one of dominant removal pathways for  $RO_2$  radicals when the concentration of NO is low and the concentration of  $RO_2$  radicals is high."

Corresponding description has been revised in the page 16 line 392-393 of the revised manuscript:

The self-reaction is one of dominant removal pathways for  $RO_2$  radicals when the concentration of NO is low and the concentration of  $RO_2$  radicals is high.

26. Line 453: remove "are" from "are of less importance".

**Response:** Based on the Reviewer's suggestion, the word "are" has been removed in the revised manuscript.

Corresponding description has been revised in the page 17 line 446-447 of the revised manuscript:

But the barriers of R20 and R21 are significantly high, making them of less importance in the atmosphere.

27. Line 459-460: awkward phrasing in "accompany with".

**Response:** Based on the Reviewer's suggestion, the mentioned sentence has been reworded as "These two processes overcome the barriers of 21.4 and 1.3 kcal mol-1, respectively".

Corresponding description has been revised in the page 18 line 453-454 of the revised manuscript:

*These two processes overcome the barriers of 21.4 and 1.3 kcal mol*-1, *respectively.*

28. Line 460: "it" in "Then it decomposes..." is vague.

**Response:** Based on the Reviewer's suggestion, the mentioned sentence has been revised as "Then, S21 decomposes into the propagation (2HOCH(CH3)O +  ${}^{3}O_{2}$ ) and termination products (HOCH(CH3)OH +  ${}^{3}CH_{3}OOH + {}^{3}O_{2}$  and dimer S22 +  ${}^{3}O_{2}$ ) with the exoergicity of 12.5, 11.7 and 33.0 kcal mol-1."

Corresponding description has been revised in the page 18 line 454-456 of the revised manuscript:

Then, S21 decomposes into the propagation  $(2HOCH(CH_3)O + {}^{3}O_2)$  and termination products  $(HOCH(CH_3)OH + {}^{3}CH_3OOH + {}^{3}O_2$  and dimer S22 +  ${}^{3}O_2$ ) with the exoergicity of 12.5, 11.7 and 33.0 kcal mol-1.

29. Line 472-473: grammatical error in "Different the self-reactions of HOCH2OO and HOCH(CH3)OO radicals".

**Response:** Based on the Reviewer's suggestion, the mentioned sentence has been revised as "Compared with the self-reactions of HOCH2OO and HOCH(CH3)OO radicals, it can be found that the termination product of the self-reaction of  $HOC(CH_3)_2OO$  radical is exclusively dimer S27."

Corresponding description has been revised in the page 18 line 466-468 of the revised manuscript:

Compared with the self-reactions of  $HOCH_2OO$  and  $HOCH(CH_3)OO$  radicals, it can be found that the termination product of the self-reaction of  $HOC(CH_3)_2OO$  radical is exclusively dimer S27.

30. Line 480: "undergo autoxidation".

**Response:** Based on the Reviewer's suggestion, the word "undergo autoxidation" has been added in the revised manuscript.

Corresponding description has been revised in the page 18 line 473-475 of the revised manuscript:

It is therefore that the tertiary  $RO_2$  radicals have great opportunity to react with  $HO_2$  radical or undergo autoxidation in pristine environments.

31. Line 480: "pristine environments" or "a pristine environment".

**Response:** Based on the Reviewer's suggestion, the word "pristine environments" has been used in the revised manuscript.

Corresponding description has been revised in the page 18 line 473-475 of the revised manuscript:

It is therefore that the tertiary RO2 radicals have great opportunity to react with HO2 radical or undergo autoxidation in pristine environments.

32. Line 496: awkward phrasing in "The main sources of HO2 radical involve...".

**Response:** Based on the Reviewer's suggestion, the mentioned sentence has been reworded as "The primary sources of  $HO_2$  radical involve the photo-oxidation of oxygenated volatile organic compounds (OVOCs) and the ozonolysis reaction, as well as secondary sources include the reactions of OH radical with CO, ozone and volatile organic compounds (VOCs), the reaction of alkoxy radical RO with  $O_2$  and the red-light-induced decomposition of  $\alpha$ -hydroxy methylperoxy radical OHCH2OO."

Corresponding description has been revised in the page 21 line 491-497 of the revised manuscript:

The primary sources of  $HO_2$  radical involve the photo-oxidation of oxygenated volatile organic compounds (OVOCs) and the ozonolysis reaction, as well as secondary sources include the reactions of OH radical with CO, ozone and volatile organic compounds (VOCs), the reaction of alkoxy radical RO with  $O_2$  and the red-light-induced decomposition of  $\alpha$ -hydroxy methylperoxy radical OHCH2OO (Kumar and Francisco, 2015; Stone et al., 2012; Hofzumahaus et al., 2009).

33. Line 498: typo "senondary".

**Response:** Based on the Reviewer's suggestion, the word "senondary" has been corrected in the revised manuscript.

34. Line 545: awkward phrasing in "resulting finally HOMs...".

**Response:** Based on the Reviewer's suggestion, the mentioned sentence has been reworded as "The autoxidation mechanism includes an intramolecular H-shift from the  $-CH_3$  or  $-CH_2$ groups to the -OO site, leading to the formation of a hydroperoxyalkyl radical QOOH, followed by O2 addition to form a new peroxy radical (HOOQO2), one after the other, resulting in formation of HOMs"

Corresponding description has been revised in the page 21 line 535-539 of the revised manuscript:

The autoxidation mechanism includes an intramolecular H-shift from the  $-CH_3$  or  $-CH_2$ groups to the -OO site, leading to the formation of a hydroperoxyalkyl radical QOOH, followed by  $O_2$  addition to form a new peroxy radical (HOOQO2), one after the other, resulting in formation of HOMs (Rissanen et al., 2014; Berndt et al., 2015).

**References**

- Crounse, J. D., Nielsen, L. B., Jørgensen, S., Kjaergaard, H. G., and Wennberg, P. O.: Autoxidation of organic compounds in the atmosphere, J. Phys. Chem. Lett., 4, 3513-3520, https://doi.org/10.1021/jz4019207, 2013.
- Jokinen, T., Sipilä M., Richters, S., Kerminen, V. M., Paasonen, P., Stratmann, F., Worsnop, D., Kulmala, M., Ehn, M., Herrmann, H., and Berndt, T.: Rapid autoxidation forms highly oxidized RO2 radicals in the atmosphere, Angew. Chem. Int. Ed., 53, 14596-14600, https://doi.org/10.1002/anie.201408566, 2014.
- Piletic, I. R., Edney, E. O., and Bartolotti, L. J.: Barrierless reactions with loose transition states govern the yields and lifetimes of organic nitrates derived from isoprene, J. Phys. Chem. A, 121, 8306-8321, https://doi.org/10.1021/acs.jpca.7b08229, 2017.
- Rissanen, M. P., Kurtén, T., Sipilä, M., Thornton, J. A., Kangasluoma, J., Sarnela, N., Junninen, H., Jørgensen, S., Schallhart, S., Kajos, M. K., Taipale, R., Springer, M., Mentel, T. F., Ruuskanen, T., Petäjä, T., Worsnop, D. R., Kjaergaard, H. G., and Ehn, M.: The formation of highly oxidized multifunctional products in the ozonolysis of cyclohexene, J. Am. Chem. Soc., 136, 15596-15606, https://doi.org/10.1021/ja507146s, 2014.
- Zhang, D., Zhang, R., Park, J., and North, S. W.: Hydroxy peroxy nitrites and nitrates from OH initiated reactions of isoprene, J. Am. Chem. Soc., 124, 9600-9605, https://doi.org/10.1021/ja0255195, 2002.

---

## Author Response (AR3)

Prof. Yu Huang
State Key Lab of Loess and Quaternary Geology
Institute of Earth Environment, Chinese Academy
of Sciences, Xi'an, 710061, China
Tel./Fax: (86) 29-62336261
E-mail: huangyu@ieecas.cn

Feb. 21, 2022

Dear Prof. Chan,

**Revision for Manuscript acp-2021-890**

We thank you very much for giving us the opportunity to revise our manuscript. We highly appreciate the editor for their comments and suggestions on the manuscript entitled "**OH-initiated atmospheric degradation of hydroxyalkyl hydroperoxides: mechanism, kinetics, and structure-activity relationship**". We have made revisions of our manuscript carefully according to the comments and suggestions of editor. The language issues have been corrected carefully in the revised manuscript. The response letter to editor is attached at the end of this cover letter.

We hope that the revised manuscript can meet the requirement of Atmospheric Chemistry & Physics. Any further modifications or revisions, please do not hesitate to contact us.

Look forward to hearing from you as soon as possible.

Best regards,

Yu Huang

**Comments of editor**

1. Line 340: "The most likely reason is the stability...", not "the most likely reason is due to the stability"

   **Response:** Based on the Editor's suggestion, the mentioned sentence has been corrected as "The most likely reason is the stability of pre-reactive complexes that IM8-a is more stable than IM6-a in energy."

   Corresponding description has been revised in the page 13 line 341-342 of the revised manuscript:

   *The most likely reason is the stability of pre-reactive complexes that IM8-a is more stable than IM6-a in energy.*

2 Again, I suggest the authors go through and review the language issues of the whole manuscript thoroughly, in addition to the areas the reviewers and myself pointed out.

   **Response:** Based on the Editor's suggestion, the language issues have been corrected carefully in the revised manuscript, and the sentences and grammar have been proofread detailedly by some native English speakers.